# Repression of LSD1 potentiates homologous recombination-proficient ovarian cancer to PARP inhibitors through down-regulation of BRCA1/2 and RAD51

Lei Tao [1,10], Yue Zhou[1,10], Xiangyu Pan[1], Yuan Luo[1], Jiahao Qiu[1,2], Xia Zhou[1], Zhiqian Chen[3], Yan Li[4], Lian Xu[5], Yang Zhou[1], Zeping Zuo[1,6], Chunqi Liu[1], Liang Wang[1], Xiaocong Liu[1], Xinyu Tian[1], Na Su[7,8], Zhengnan Yang[1], Yu Zhang[9], Kun Gou[1], Na Sang[1], Huan Liu[2,7], Jiao Zou[1], Yuzhou Xiao[1], Xi Zhong[7], Jing Xu[1], Xinyu Yang[7], Kai Xiao[1], Yanyang Liu[1], Shengyong Yang [1], Yong Peng [1], Junhong Han [1], Xiaobo Cen [2] & Yinglan Zhao[1] ✉

Poly (ADP-ribose) polymerase inhibitors (PARPi) are selectively active in ovarian cancer (OC) with homologous recombination (HR) deficiency (HRD) caused by mutations in BRCA1/2 and other DNA repair pathway members. We sought molecular targeted therapy that induce HRD in HR-proficient cells to induce synthetic lethality with PARPi and extend the utility of PARPi. Here, we demonstrate that lysine-specific demethylase 1 (LSD1) is an important regulator for OC. Importantly, genetic depletion or pharmacological inhibition of LSD1 induces HRD and sensitizes HR-proficient OC cells to PARPi in vitro and in multiple in vivo models. Mechanistically, LSD1 inhibition directly impairs transcription of BRCA1/2 and RAD51, three genes essential for HR, dependently of its canonical demethylase function. Collectively, our work indicates combination with LSD1 inhibitor could greatly expand the utility of PARPi to patients with HR-proficient tumor, warranting assessment in human clinical trials.

Ovarian cancer (OC) is the second most common gynecological cancer with the highest mortality rate worldwide[1,2]. The high-grade serous OC subtype accounts for 70–80% of OC deaths, and its 5-year overall survival is less than 40% and has not changed for decades[3,4].

Although surgery followed by cytotoxic platinum-based chemotherapy is the standard therapy for OC patients, most patients develop resistance to platinum-based chemotherapy, resulting in recurrence and death[4].

[1]Department of Biotherapy, Cancer Center and State Key Laboratory of Biotherapy, West China Hospital, Sichuan University, 610041 Chengdu, China. [2]National Chengdu Center for Safety Evaluation of Drugs, State Key Laboratory of Biotherapy, West China Hospital, Sichuan University, 610041 Chengdu, China. [3]State Key Laboratory of Natural and Biomimetic Drugs, School of Pharmaceutical Sciences, Peking University, 100191 Beijing, China. [4]Department of Pharmacology, Shanxi Medical University, 030001 Taiyuan, China. [5]Department of Pathology, Key Laboratory of Birth Defects and Related Diseases of Women and Children of Ministry of Education, West China Second University Hospital, Sichuan University, 610041 Chengdu, China. [6]Laboratory of Anesthesiology & Critical Care Medicine, Department of Anesthesiology, West China Hospital, Sichuan University, 610041 Chengdu, China. [7]Department of Pharmacology, Key Laboratory of Drug Targeting and Drug Delivery System of the Education Ministry, Sichuan Engineering Laboratory for Plant-Sourced Drug and Sichuan Research Center for Drug Precision Industrial Technology, West China School of Pharmacy, Sichuan University, 610041 Chengdu, China. [8]Department of Pharmacy, West China Hospital, Sichuan University, 610041 Chengdu, China. [9]School of Medicine, Tibet University, 850000 Lhasa, China. [10]These authors contributed equally: Lei Tao, Yue Zhou. ✉e-mail: zhaoyinglan@scu.edu.cn

Approximately 50% of OC patients harbor aberrations in homologous recombination (HR) DNA repair which most commonly due to mutation in BRCA1/2, resulting in the accumulation of DNA double-strand breaks (DSBs)[5]. HR deficiency (HRD) creates a vulnerability that can be exploited to selectively kill cancer cells by means of synthetic lethality. The paradigm for this approach is the use of Poly-(ADP-ribose) polymerase (PARP) inhibitors (PARPi) for the treatment of OC patients with HRD[6]. For tumors with abnormal HR repair function, PARPi inhibit the activity of PARP enzyme and increase the formation of the PARP-DNA complex, leading to DNA damage repair obstacle and promoting death of tumor cells. Currently, three PARPi (olaparib, niraparib, and rucaparib) have been approved by the US Food and Drug Administration (FDA) for OC treatment[6]. Despite the notable success of PARPi in clinic for patients with BRCA1/2 mutations, as many as 50% of OC patients retaining HR proficiency only obtain limited benefit from these drugs[5]. Moreover, acquired resistance to PARPi is a major problem for OC that is initially HRD but acquires HR proficiency after PARPi treatment by multiple mechanisms, including secondary mutations that restore BRCA1/2 and RAD51 functions[7,8]. Thus, there is an urgent clinical need to identify new molecular targets and potential combination therapeutic strategies to expand PARPi utility into HR-proficient OC.

To date, no direct inhibitors specifically targeting the proteins catalyzing HR are available. Intriguingly, to expand the use of PARP inhibitors to a larger group of HR-proficient OC patients, recent studies have focused on new combination strategies using agents that can induce HRD[7,9]. Targeting actionable proteins can interfere with gene expression, nuclear localization, and/or the recruitment of HR proteins, ultimately resulting in the indirect inhibition of HR and thereby engendering PARPi sensitivity[7,9]. For example, prior studies have demonstrated that targeting PI3K/AKT, RAS/MEK, and vascular endothelial growth factor receptor (VEGFR) pathways has the potential to pharmacologically induce an HRD phenotype[10–14]. Importantly, the DNA damage repair pathways are closely associated with chromatin remodeling mediated by histone modifications, providing a rationale for combining PARPi with epigenetic agents such as DNA methyltransferase (DNMT) inhibitors, histone deacetylase (HDAC) inhibitors, bromodomain and extra-terminal domain (BET) inhibitors, enhancer of the zeste homolog 2 (EZH2) inhibitors and protein arginine methyltransferase 5 (PRMT5) inhibitors[15–20]. In this regard, selectively impairing HR in OC cells has been demonstrated to sensitize HR-proficient cancer cells to PARPi in preclinical and early clinical trials[7–20].

Lysine-specific demethylase 1 (LSD1; also known as KDM1A, AOF2; encoded by *KDM1A* in humans) is a promising target that may be exploited for molecular-based anti-cancer therapies[21]. LSD1 demethylates mono- and dimethylated histone H3 lysine 4 (H3K4me1/2), working as a transcriptional repressor in complex with Nurd or CoREST[22,23], and histone H3 lysine 9 (H3K9me1/2), behaving as a transcriptional activator in complex with androgen receptor (AR) or estrogen receptor (ER)[24–26]. Moreover, LSD1 demethylates nonhistone substrates, including p53, DNMT1, E2F1, HIF-1α, STAT3 and MYPT1[27]. The overexpression of LSD1 is observed in different types of malignancies, such as colorectal, breast, prostate, small cell lung cancer (SCLC), and acute myeloid leukemia (AML), exerting a tumor-promoting activity[28]. These studies underscore the important role of LSD1 in oncogenesis and provide evidence that inhibition of LSD1 may offer a therapeutic strategy for the treatment of cancer. In line with the above observation, LSD1 inhibitors (LSD1i) including IMG-7289, CC-90011, and SP2577 are currently undergoing clinical study for treatment of cancers, such as AML, SCLC, prostate cancer, and Ewing sarcoma[28,29]. In our previous manuscript, we have developed a specific LSD1 inhibitor named ZY0511, which inhibits LSD1 at nanomolar concentration[30], exhibits growth inhibition against human colorectal and cervical cancer xenografts in nude mice, and enhances the sensitivity of human colorectal cancer cells to 5-fluorouracil[31,32].

Until now, a few studies have shown that LSD1 mRNA and protein are highly expressed in human OC tissues, which is correlated with FIGO stage and lymphatic metastasis[33,34]. LSD1 promotes the proliferation, migration, and invasion of OC cells[35,36]. Knockdown of LSD1 impairs the proliferation and mediates cisplatin sensitivity in OC cells[37]. However, the above studies are only in the preliminary stage of exploring the role of LSD1 in OC, and the molecular mechanism of LSD1 in OC progression and the potential of LSD1i in OC treatment remains unclear. In addition, emerging studies implicate that LSD1 is associated with DNA damage repair. LSD1 is shown to be recruited to the DNA damage sites in RNF168-dependent manner and promotes the recruitment of 53BP1 and BRCA1 in U2OS cells[38]. LSD1 depletion sensitizes Hela cells to γ-irradiation and HEK293 cells to DNA damage agents (bleomycin and etoposide)[38,39]. Moreover, LSD1 directly binds to FBXW7 to destabilize FBXW7, leading to abrogating FBXW7's functions in growth suppression, nonhomologous end-joining repair (NHEJ), and radioprotection in lung cancer cells[40]. These studies suggest that LSD1 may be a potential regulator of OC and plays an important role in DNA damage response. However, the precise molecular mechanism of LSD1 underlying DNA damage repair and whether LSD1 inhibition sensitizes HR-proficient OC to PARPi remain unclear.

In this study, we explored the role and underlying mechanism of LSD1 in DSB repair in OC and addressed whether LSD1-directed therapy represents a promising strategy for OC with HR-proficient in combination with PARPi treatment. We herein report that LSD1 knockdown and pharmacological inhibition suppress OC growth in vitro and in vivo. Inhibition of LSD1 impedes HR repair through directly impairing expression of BRCA1/2 and RAD51 genes. Importantly, we demonstrated that inhibition of LSD1 induced HRD and sensitized HR-proficient OC cells to PARPi in vitro and in multiple in vivo models. Therefore, our findings reveal the critical role of LSD1 in regulating HR and demonstrate LSD1i as a strategy to enhance the efficiency and expand the utility of PARPi for treatment of OC with HR-proficient.

## Results
### Genetic depletion of LSD1 inhibits the growth of OC cells in vitro and in vivo

To investigate the clinical significance of LSD1 in OC, we examined the expression of LSD1 between human OC tissues and non-malignant normal tissues, including normal human ovarian surface epithelium (HOSE) and fallopian tube epithelium (FTE) based on five publicly available datasets, including TCGA (http://tcgaportal.org/index.html), GSE26712, GSE12470, GSE10971 and CPTAC (https://proteomics.cancer.gov/programs/cptac). The results showed that LSD1 expression was significantly enriched in the OC tissues compared with non-malignant tissues in five publicly available datasets (Supplementary Fig. 1a, b). Moreover, the Kaplan-Meier survival analysis revealed that LSD1 expression was negatively correlated with overall survival and progression-free survival of OC patients (Supplementary Fig. 1c, d). Notably, through the cBioPortal and muTarget platform, LSD1 mRNA expression levels in BRCA1/2 wild-type groups of OC patients were found to be higher than that in BRCA1/2 mutation groups (Supplementary Fig. 1e, f). In addition, we identified LSD1 capable of classifying platinum drug responses using ROC Plotter (https://www.rocplot.org/ovarian/index)[41]. The elevated expression of LSD1 in OC samples was associated with increased chemoresistance to platinum (Supplementary Fig. 1g, h), indicating potential clinical utility of LSD1 as prognostic and predictive biomarker in OC.

We next performed immunohistochemistry (IHC) analysis to examine the protein level of LSD1 by using a human OC tissue microarray (TMA) containing 45 OC samples and corresponding normal adjacent tissues (NATs) (the clinical characteristics of patients are provided in Supplementary Table 1). Consistently with the results from publicly available datasets, we observed elevated protein levels of LSD1

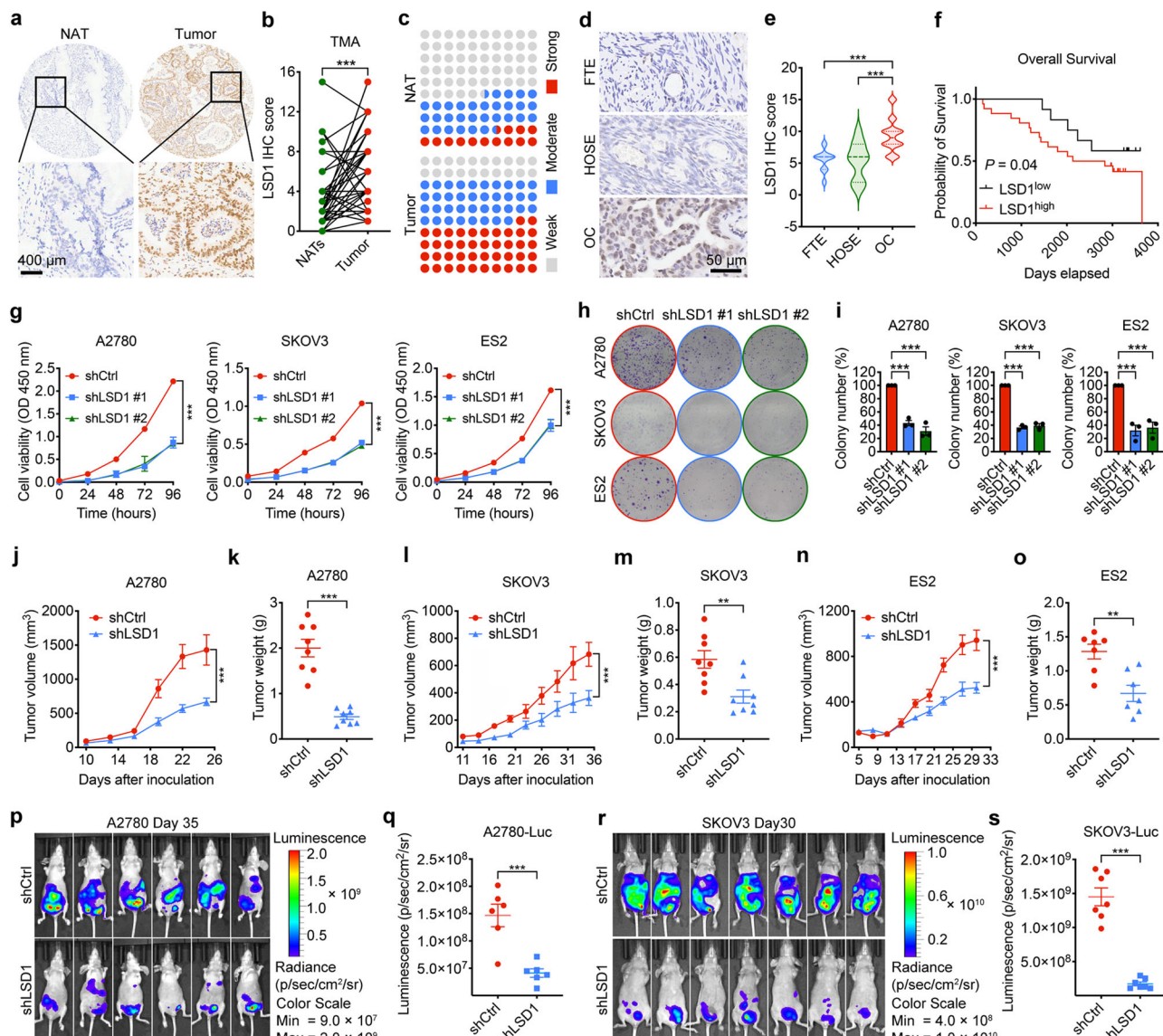

**Fig. 1 | LSD1 is key mediator in OC and promotes OC progress.**
**a**–**c** Representative IHC images (**a**) and quantification (**b, c**) of LSD1 protein level in human OC tissues and NATs from TMA (*n* = 45 paired samples). The quantification analyses were based on staining density scores of IHC (paired two-tailed *t* test). Scale bar, 400 μm. **d, e** Representative IHC images (**d**) and quantification (**e**) of LSD1 protein level in human FTE (*n* = 30 samples), HOSE (*n* = 10 samples) and OC (*n* = 39 samples) tissues. The quantification analyses were based on staining density scores of IHC (unpaired two-tailed Student's *t* test). Scale bar, 50 μm.
**f** Kaplan–Meier plot depicting overall survival of OC patients with tumors expressing high (red) or low (black) levels of LSD1 using TMA (log-rank test). **g** Cell growth of LSD1 knockdown cells detected by CCK8 assay compared with their control (shCtrl). Data represent mean ± SEM of three biologically independent experiments (two-way ANOVA). **h, i** Representative images (**h**) and quantification (**i**) of colony formation assay for OC cells with LSD1 knockdown. Samples were normalized to shCtrl. Data represent mean ± SEM of three biologically independent experiments (unpaired two-tailed Student's *t* test). **j**–**o** Tumor volume and tumor weight in shLSD1-expressing and shCtrl-expressing A2780 (**j, k**), SKOV3 (**l, m**) and ES2 (**n, o**) subcutaneous xenografts in nude mice. Data represent mean ± SEM (*n* = 8 mice per group for A2780 and SKOV3 xenograft models, *n* = 7 mice per group for ES2 xenograft models; two-way ANOVA for panels **j, l, n**, and unpaired two-tailed Student's *t* test for panels **k, m, o**). **p**–**s** Representative living luminescence images (**p, r**) and quantification of the luciferase fluorescence signal intensity (**q, s**) of shLSD1-expressing and shCtrl-expressing A2780 and SKOV3 intraperitoneal xenografts in nude mice. Data represent mean ± SEM (*n* = 6 mice per group for A2780 xenograft models, *n* = 7 mice per group for SKOV3 xenograft models; unpaired two-tailed Student's *t* test). *$p < 0.05$; **$p < 0.01$; ***$p < 0.001$. Source data and exact *p* values are provided in the Source Data file.

in OC samples compared with NATs (Fig. 1a, b). Indeed, 19 cases (42%) exhibited strong immunopositivity, 17 cases (38%) exhibited moderate immunopositivity, and 9 cases (20%) exhibited weak immunopositivity in tumor tissues. In contrast, most normal tissues (56%) exhibited no or weak LSD1 expression (Fig. 1c). In addition, LSD1 was upregulated in OC compared with normal HOSE and FTE by IHC and western blot analysis, consistent with our findings from the publicly available datasets (Fig. 1d, e and Supplementary Fig. 1i, j). Moreover, Kaplan-Meier survival analysis revealed that the expression of LSD1 in OC

tissues was negatively correlated with the patient overall survival rate (Fig. 1f). These findings indicated that LSD1 was closely correlated with clinical outcome of OC patients. Furthermore, reverse-transcriptase quantitative PCR (RT-qPCR) and western blot analysis showed LSD1 was more highly expressed in OC cell lines compared with normal ovarian epithelial cells (Supplementary Fig. 1k, l). Notably, LSD1 level in OC cells such as A2780, SKOV3 and ES2 cells which are BRCA-proficient and considered as HR-proficient (https://cancer.sanger.ac.uk/cosmic) were dominantly higher than that in HRD OC cell lines including

COV362 (with BRCA1 mutation) and Kuramochi (with BRCA2 mutation). Collectively, these results reveal that expression of LSD1 is elevated in human OC tissues and is tightly linked to OC progression, suggesting that LSD1 might be an important regulator in OC.

We next sought to investigate the potential functional roles of LSD1 in OC. We stably silenced LSD1 expression using two short-hairpin RNAs (shRNA) in three human OC cell lines (A2780, SKOV3, and ES2 cells) with relatively high expression level of LSD1 and BRCA-proficient and observed LSD1 knockdown significantly reduced cells proliferation in CCK8 and colony formation assays (Fig. 1g–i and Supplementary Fig. 2a). We further examined the effect of LSD1 knockdown in tumor growth in vivo by subcutaneously or intraperitoneally injecting shLSD1-expressing and non-targeting control shCtrl-expressing cells into nude mice, respectively. The results showed that LSD1 knockdown significantly inhibited tumor growth with a tumor growth inhibition (TGI) of 53.2%, 47.0%, and 44.3% in A2780, SKOV3, and ES2 subcutaneous xenograft models, respectively (Fig. 1j–o). This data was consistent with the results obtained from A2780 and SKOV3 intraperitoneal xenograft models in which LSD1 knockdown decreased luminescence intensity compared with that in control mice, indicating that LSD1 knockdown hindered the growth of OC cells in vivo (Fig. 1p–s). Notably, LSD1 knockdown had no apparent effect on the growth of HOSE cells (Supplementary Fig. 2b–e). Importantly, we additionally generated two LSD1 knockout (LSD1 KO) clones in A2780 and ES2 cells using the CRISPR-Cas9 methodology and validated these findings. Consistent with shRNA-mediated LSD1 knockdown, LSD1 KO cells exhibited a decrease in growth in vitro and in vivo compared with parental controls (Supplementary Fig. 2f–m). Thus, these findings indicate that LSD1 inhibition suppresses OC growth both in vitro and in vivo.

### Pharmacological inhibition of LSD1 exhibits therapeutic potential in OC

To further investigate the therapeutic potential of LSD1 inhibition in OC treatment, we evaluated the efficiency of pharmacological inhibitor of LSD1 in OC cells. We used a potent LSD1 inhibitor, named ZY0511, developed by our group (Supplementary Fig. 3a). The binding of ZY0511 with LSD1 in OC cells was investigated by cellular thermal shift assay (CETSA). The results showed that ZY0511 treatment increased the thermal stability of LSD1, with an aggregation temperature shift of LSD1 in A2780, SKOV3, and ES2 cells with 1.5, 1.4, and 1.3 °C, respectively, demonstrating the binding of ZY0511 to LSD1 in these cells (Supplementary Fig. 3b, c). Since LSD1 demethylates H3K4me1/2 and H3K9me1/2, we detected global effects on H3K4 and H3K9 methylation by western blot analysis and observed LSD1 inhibition by ZY0511 treatment resulted in some increase of H3K4me1/2 and H3K9me1/2 in a concentration- and time-dependent manner (Supplementary Fig. 3d, e). Then, the cell proliferation assays showed that ZY0511 obviously inhibited A2780, SKOV3 and ES2 cells proliferation with low $IC_{50}$ values less than 0.5 μM after 72 h treatment, whereas the normal ovarian epithelial cells, IOSE80 and HOSEpiC, were insensitive to ZY0511 treatment (Fig. 2a). We further detected whether ZY0511 efficiency in OC cells proliferation is dependent on LSD1 inhibition or not by using shLSD1 cells. Both CCK8 and colony formation assays in A2780, SKOV3, and ES2 cells showed that shLSD1 cells were less sensitive to ZY0511 treatment than non-targeting control shCtrl-expressing cells, suggesting that ZY0511 inhibited OC cell proliferation in an LSD1 expression-dependent manner (Fig. 2b, c). In addition, the colony formation assay further confirmed that ZY0511 suppressed OC cell proliferation in a concentration-dependent manner (Supplementary Fig. 3f, g). Moreover, ZY0511 exposure induced the cell cycle S phase arrest (Supplementary Fig. 3h, i) and increased the numbers of Annexin V+ cells of OC cells (Supplementary Fig. 3j–m). In line with this, ZY0511 upregulated cleaved caspase 3, cleaved caspase 9, and cleaved PARP protein expression (Supplementary Fig. 3n). Notably, ZY0511 had

just a slight effect on the proliferation of IOSE80 cells at high concentration (Supplementary Fig. 3f, g and Supplementary Fig. 3j–m). Altogether, these findings indicate that ZY0511 on-target prevents OC proliferation in vitro.

Next, the in vivo antitumor activities of ZY0511 were evaluated with both subcutaneous and intraperitoneal xenograft models in nude mice and showed that ZY0511 suppressed A2780, SKOV3 and ES2 tumor growth (Fig. 2d–f). Indeed, 60 mg/kg body weight ZY0511 resulted in 52.6% to 70.0% TGI compared with vehicle control in three subcutaneous xenograft models and two intraperitoneal xenograft models in vivo. Through CETSA conducted on tumor homogenates, we revealed a clear increase of the remaining soluble LSD1 levels in the ZY0511-treated group compared with the controls after heating intact tissue samples at 55 °C, further demonstrating the ability of ZY0511 to engage LSD1 (Fig. 2g). IHC assay showed that proliferation (Ki67+) cells were decreased, and apoptosis (Cleaved Caspase-3+) cells were increased in tumor tissues after ZY0511 treatment (Fig. 2h, i). In addition, ZY0511 also exhibited good pharmacokinetic properties allowing for use in in vivo studies (Fig. 2j). The tissue distribution assay showed ZY0511 sufficiently distributed in ovary tissues, supporting its potential for OC treatment (Fig. 2k). Furthermore, hematoxylin and eosin (H&E) staining showed that ZY0511 administration exhibited no lesions of main organs (Fig. 2l) and did not alter the hematological and blood biochemistry parameters of mice (Fig. 2m, n), suggesting that ZY0511 is well tolerated in vivo. Taken together, these data suggest that ZY0511 suppresses OC growth in vivo.

### LSD1 inhibition activates DDR and suppresses expression of HR proteins

To explore the molecular mechanism by which LSD1 inhibition suppress development of OC, RNA sequencing (RNA-seq) was performed in cells treated with either ZY0511 (LSD1 inhibitor, LSD1i) or vehicle or following knockdown of LSD1 (shLSD1-expressing) to detect the genes involved in this process. The results showed that there was some overlap in differentially expressed genes (DEGs) between LSD1i treated and LSD1 knockdown cells (32.3% commonly upregulated; 42.2% commonly downregulated) (Supplementary Fig. 4a, b). Ingenuity Pathway Analysis indicated that LSD1 inhibition led to significant suppression of genes involved in DNA damage response, DNA repair, and cell-cycle checkpoint control, while significantly upregulated genes included these involved in p53 and apoptosis signaling (Fig. 3a, b and Supplementary Fig. 4c). Moreover, there was a high coincidence of differentially expressed pathways affected by genetic silencing and pharmacological inhibition of LSD1 in both A2780 and ES2 cells (Fig. 3a, b and Supplementary Fig. 4c). For example, the ATM signaling pathway and the role of BRCA1 in DNA damage response pathway were suppressed by both ZY0511 treatment and genetic silencing of LSD1. Notably, results of gene set enrichment analysis (GSEA) further confirmed that LSD1 inhibition downregulated the genes involved in DSB repair, particularly homology-directed repair in both A2780 and ES2 cells. Additionally, LSD1 inhibition induced significant perturbation of genes including a published PARPi sensitization gene signature[42] (Fig. 3c). The heatmap analysis showed that a subset of DNA DSB repair pathway genes was downregulated in LSD1 knockdown group (Fig. 3d). As previous study showed that HRD signature consist of 230 DEGs[42], we applied the HRD gene signatures to our RNA-seq data to investigate whether LSD1 inhibition impaired HR and found that LSD1 inhibition significantly elevated HRD scores in A2780 and ES2 cells, suggesting LSD1 inhibition induced DNA repair gene defect as a sensitization mechanism to PARPi (Fig. 3e).

We further validated the changes of selected DNA DSB repair genes in three independent cell lines by RT-qPCR and western blot analysis. The results showed that both LSD1 knockdown and pharmacological inhibition of LSD1 resulted in the downregulation of DNA DSB repair genes at the mRNA level (Supplementary Fig. 4d). We noted

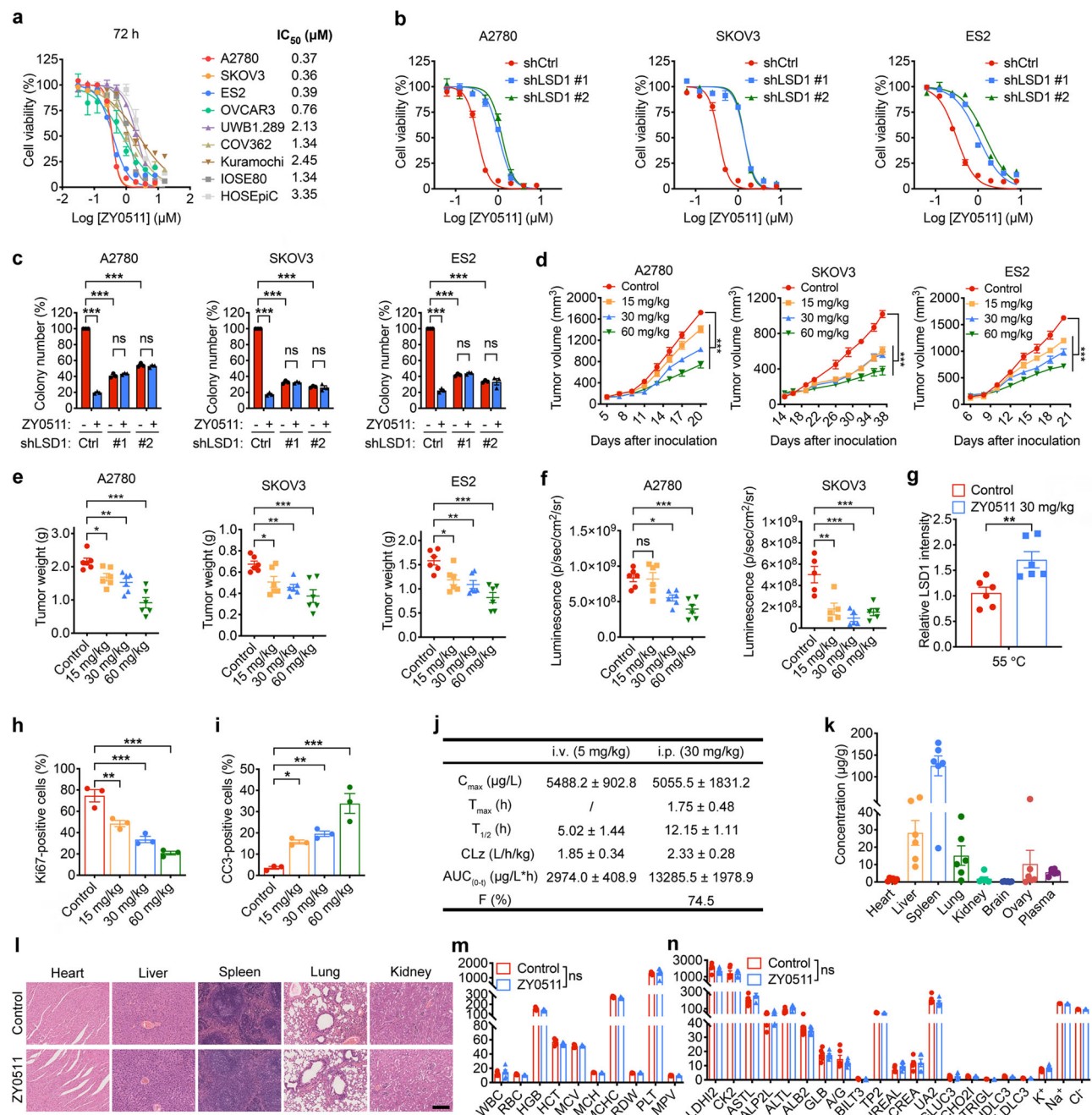

**Fig. 2 | LSD1 pharmacological inhibition has therapeutic potential in OC. a** Cell viability in the indicated OC cell lines after ZY0511 treatment for 72 h. Data represent mean ± SEM of three biologically independent experiments. **b** CCK8 assay at varied concentrations of ZY0511 in shCtrl-expressing and shLSD1-expressing A2780, SKOV3 and ES2 cells. Data represent mean ± SEM of three biologically independent experiments). **c** Quantification of colony formation assay. Data represent mean ± SEM of three biologically independent experiments; two-way ANOVA). **d, e** Tumor volume (**d**) and tumor weight (**e**) of mice bearing A2780, SKOV3 and ES2 subcutaneous xenografts. Data represent mean ± SEM (n = 6 mice per group; two-way ANOVA for panel **d** and one-way ANOVA for panel **e**). **f** The luciferase fluorescence signal intensity of mice bearing A2780-Luc and SKOV3-Luc intraperitoneal xenografts. Data represent mean ± SEM (n = 6 mice per group for A2780 xenograft models, n = 5 mice per group for SKOV3 xenograft models; one-way ANOVA). **g** CETSA performed in harvested tissues. Data represent mean ± SEM (n = 6 mice per group; unpaired two-tailed Student's t test). **h, i** Quantification of IHC of the indicated proteins in tumor tissues from mice. Data represent mean ± SEM of from three different mice (one-way ANOVA). CC3 Cleaved Caspase 3. **j** Pharmacokinetics of LSD1 inhibitor (ZY0511). SD rats were administered 5 mg/kg ZY0511 intravenously (n = 5) or 30 mg/kg ZY0511 intraperitoneally (n = 4). Data represent mean ± SEM. **k** Distribution of ZY0511 in main organs and plasma. Data represent mean ± SEM (n = 6 animals per group). **l** Representative H&E staining images of the heart, liver, spleen, lung, and kidney at the end of the dosing. Scale bar, 200 μm. **m, n** Blood routine assay (**m**) and blood biochemical assay (**n**) performed at the end of treatment. Data represent mean ± SEM (n = 6 mice per group; unpaired two-tailed Student's t test). ns not significant, p > 0.05; *p < 0.05; **p < 0.01; ***p < 0.001. Source data and exact p values are provided in the Source Data file.

that LSD1i (ZY0511) and shLSD1 markedly and consistently decreased BRCA1, BRCA2 and RAD51 that commits cells to HR repair in all lines assessed (Fig. 3f–h and Supplementary Figs. 4d and 5a, b). Furthermore, we found that LSD1 inhibition extensively rewired protein networks, including multiple components of the DNA damage response pathway (p-ATM(S1981), p-53BP1(S1778), p-CHK2(T68), p-RPA32(S4/S8), and p-RPA32(S33)) and induced DNA damage (γH2AX) (Fig. 3h). However, in contrast to BRCA1, BRCA2 and RAD51, which

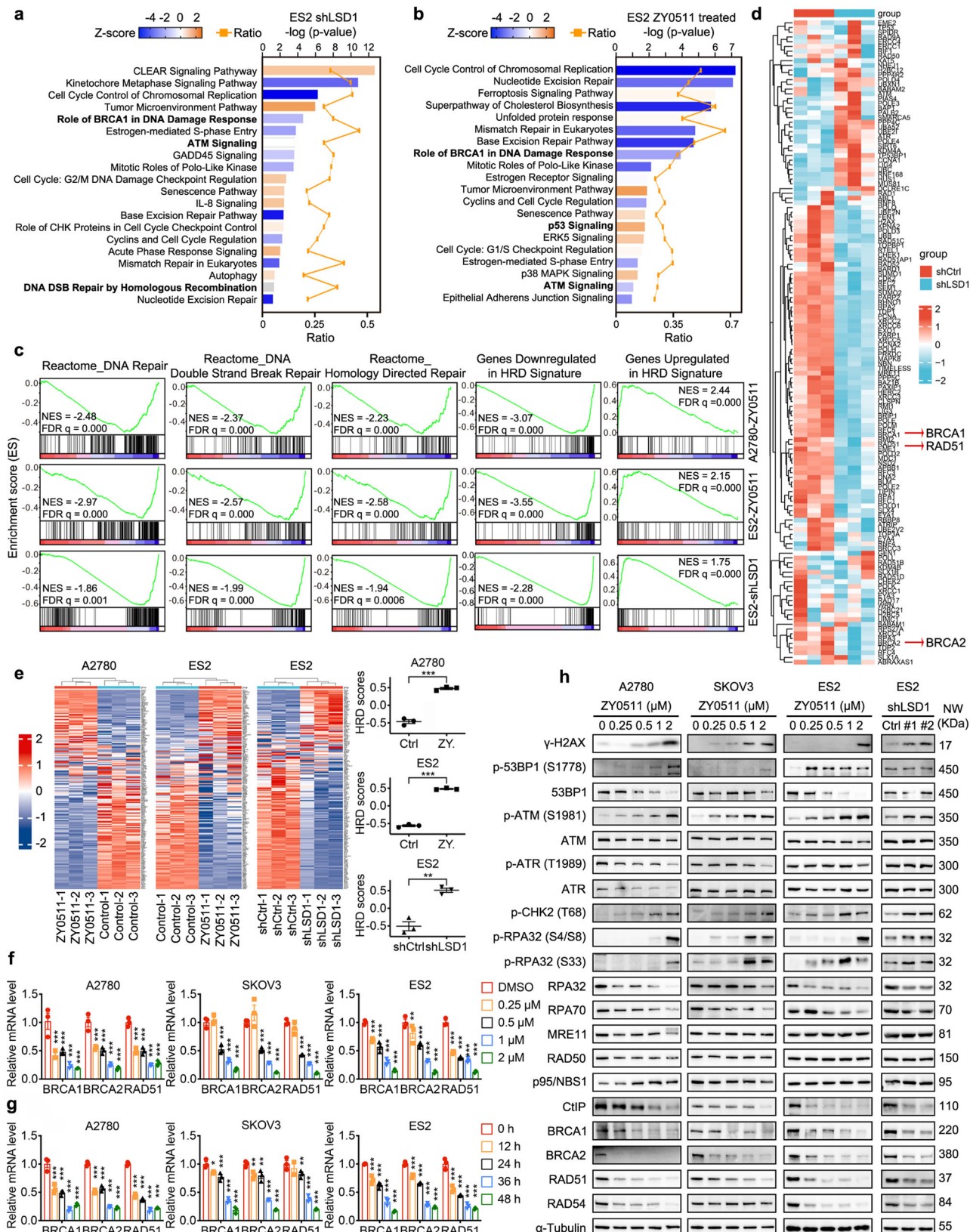

were consistently downregulated under all conditions, the effects of LSD1 inhibition on RPA32, RPA70, CtIP, RAD54, and DNA ligase IV were modest and variable. Notably, LSD1 inhibition did not affect the protein expression of other core proteins of NHEJ pathway such as Ku70 and Ku80 (Fig. 3h and Supplementary Fig. 5a). Moreover, to demonstrate generalizability and since ZY0511 is not a clinical candidate, we additionally assessed SP2577, a compound related to ZY0511 that recently completed a phase I clinical trial for the treatment of advanced solid tumors (NCT03600649) and entered phase I/II trials for continued access to SP2577 (NCT05266196 and NCT03600649). SP2577 also decreased BRCA1/2 and RAD51 protein levels, similar to the effects of ZY0511 (Supplementary Fig. 5b). Importantly, the expression

**Fig. 3 | LSD1 inhibition activates DDR and suppresses expression of HR proteins. a, b** Ingenuity Pathway Analysis of the representative twenty significantly regulated pathways of shLSD1-expressing versus non-targeting control shCtrl-expressing ES2 cells (**a**) and LSD1i (ZY0511)-treated versus untreated ES2 cells (**b**). Upregulated pathways are presented in orange and downregulated pathways are in blue. *p* values generated by right-tailed Fisher's exact test. **c** GSEA enrichment score curves of LSD1 knockdown or LSD1i (ZY0511) treatment regulated genes of A2780 and ES2 cells. ES, enrichment score; NES, normalized enrichment score; FDR, false discovery rate. **d** Heatmap showing gene expression changes between shLSD1-expressing and non-targeting control shCtrl-expressing ES2 cells with respect to genes contained in the "Reactome_DNA_Double_Strand_Break_Repair" gene set. **e** Heatmap (left) and HRD scores (right) from unsupervised clustering of HRD gene signatures using the RNA-seq dataset of A2780 and ES2 treated by LSD1i (ZY0511) or

LSD1 knockdown (shLSD1). Higher scores represent defective HR. Data represent mean ± SEM of three biologically independent experiments (unpaired two-tailed Student's *t* test). **f, g** RT-qPCR analysis of indicated gene expression in A2780, SKOV3 and ES2 cells treated with different concentrations of LSD1i (ZY0511) for 36 h (**f**) or with 1 μM LSD1i (ZY0511) for different time periods (**g**). GAPDH was used as the loading control. Data represent mean ± SEM of three biologically independent experiments (one-way ANOVA). **h** Western blot analysis of indicated proteins in A2780, SKOV3 and ES2 cells treated with the indicated dose of ZY0511 for 48 h or in ES2 cells treated with shRNAs targeting LSD1 (shLSD1 #1 and shLSD1 #2) and non-targeting control (shCtrl). α-Tubulin was used as the loading control. ns, not significant, *p* > 0.05; *\**p* < 0.05; \*\**p* < 0.01; \*\*\**p* < 0.001. Source data and exact *p* values are provided in the Source Data file.

changes of BRCA1/2 and RAD51 were also validated in LSD1 KO cells (Supplementary Figs. 4e and 5b). Western blot analysis of previous A2780 subcutaneous tumors determined that the expression of BRCA1/2 and RAD51 were decreased in tumors treated with LSD1i (ZY0511) (Supplementary Fig. 5c, d). Furthermore, correlation analysis of cancer cell lines from Cancer Cell Line Encyclopedia (CCLE) data showed that LSD1 mRNA expression was positively correlated with BRCA1/2, and RAD51 mRNA expression (Supplementary Fig. 5e–g). Collectively, LSD1 inhibition suppresses expression of HR proteins, especially BRCA1/2 and RAD51, suggesting LSD1 inhibition induces defect of HR-related genes which may enhance sensitivity of OC cells to PARPi.

## LSD1 inhibition suppresses HR and increases DNA DSBs

Given that LSD1 inhibition decreased protein expression of BRCA1/2 and RAD51, which are the key factors in HR repair, we next explored whether LSD1 inhibition attenuated HR repair and enhanced the DNA damage. Firstly, we conducted neutral comet assays to directly examine whether LSD1 inhibition would increase the generation of DSBs, which are characteristic of HR-deficient cells[43,44]. After either LSD1i (ZY0511) treatment or LSD1 knockdown (shLSD1), DNA tail moments were greatly increased, suggesting that the loss of LSD1 activity or expression induced substantial increase of DSBs (Fig. 4a–d). In addition, data from these different cell lines consistently showed that the tail moments of both LSD1i treatment and LSD1 knockdown cells were markedly higher than these in untreated cells at 24 h after ionizing radiation (IR) treatment, although no significant differences were observed at 0.5 h after IR treatment (Fig. 4a–d). It is well known that the S139 phosphorylation of histone H2AX (γH2AX) and phosphorylated 53BP1(p-53BP1) are markers of cellular response to DNA DSBs[43]. Thus, we detected it and found that LSD1i (ZY0511 and SP2577) and knockdown of LSD1 increased the number of γH2AX and p-53BP1 foci in OC cells (Fig. 4e, f and Supplementary Fig. 6a, b).

One measure of HR ability is the formation of RAD51 foci in response to DNA damage, which is an important step in HR-mediated DSB repair[45]. We observed that cells with LSD1i (ZY0511 or SP2577) treatment or LSD1 knockdown had a deficiency in RAD51-foci formation after treatment with IR, in agreement with decreased HR (Fig. 4g, h). Importantly, we also validated these findings using LSD1 KO cells (Supplementary Fig. 6c–j).

We further compared the kinetics of γH2AX and RAD51 induced by IR. Consistently with neutral comet assays results (Fig. 4a–d), the immunofluorescent analysis showed that the number of γH2AX foci in cells treated with LSD1i (ZY0511 or SP2577) were nearly the same as that in control cells at the initial stage (1 h after IR), whereas the number was significantly higher than that in the control cells after IR of 6, 12 h, and 24 h (Fig. 4i). Consistently, LSD1 knockdown increased the induction of γH2AX after IR (Fig. 4j). We also observed that LSD1i (ZY0511 or SP2577) and LSD1 knockdown blunted the induction of RAD51 foci, which peaked at 4–6 h after IR (Fig. 4k, l). These results suggest that LSD1 inhibition results in deficiency of DNA damage repair.

Next, we sought to evaluate the extent of the HR and NHEJ using DR-GFP and EJ5-GFP reporter assay, respectively, which are chromosomal reporter assay for HR and NHEJ widely used as benchmark assay in the DNA-repair field[46,47]. We found that LSD1i (ZY0511 or SP2577) treatment and LSD1 knockdown by siRNAs targeting LSD1 resulted in substantial suppression of HR in A2780 and ES2 DR-GFP cell models, and this suppression was similar in extent to that seen with siRNAs targeting two key HR genes, BRCA2 and RAD51. However, LSD1 inhibition slightly affected the NHEJ capacity in A2780 and ES2 EJ5-GFP cell models, while knockdown of Ku80 significantly reduced the NHEJ capacity in these models (Fig. 4m–o and Supplementary Fig. 6k). Together, these results demonstrate that LSD1 inhibition suppresses HR and increases DNA DSBs.

## LSD1 upregulates expression of BRCA1/2 and RAD51 by binding to their promoter and promoting their transcription

To explore the mechanism by which LSD1 inhibition represses the expression of key HR genes including BRCA1/2 and RAD51, we first hypothesized that LSD1 inhibition likely alter these factors expression through transcriptional effects. Cleavage Under Targets and Tagmentation sequencing (CUT&Tag-seq) studies were utilized to examine the genomic distribution of LSD1 in ES2 cells in the absence and presence of LSD1 inhibition. We performed an integrative analysis of the genes that were differentially expressed with LSD1 inhibition in ES2 cells and bounded by LSD1 in CUT&Tag-seq in ES2 cells. Of these LSD1-bound peaks, 56% of these LSD1-bound peaks were localized to intronic and intergenic regions, whereas the majority (33%) localized to promoters (Fig. 5a). Notably, a majority of the DEGs in RNA-seq with LSD1 inhibition were directly bound by LSD1 (downregulated following LSD1 inhibition, 1113/1280, 86.95%; upregulated following LSD1 inhibition, 879/1040, 84.52%), suggesting that the inhibition of LSD1 both directly and indirectly affected the expression of genes important for ES2 cells growth (Supplementary Fig. 7a, b). Furthermore, the Ingenuity Pathway Analysis of the total 1113 overlap genes that were both downregulated by LSD1 inhibition and directly bound by LSD1 revealed a significant enrichment in the DNA damage repair pathway (Supplementary Fig. 7c). Analysis of the average LSD1, H3K9me2, and H3K4me2 density at genomic regions (gene body ± 5,000 bp surrounding transcription start site [TSS]) revealed enrichment for both histone marks and LSD1 at regions of the promoter (Fig. 5b). Notably, LSD1 inhibition by LSD1i (ZY0511) resulted in decreased enrichment of LSD1 at the promoter regions. Next, to determine if the dynamic of H3K4me2 and H3K9me2 accounts for the alteration of gene expression, we performed H3K4me2 or H3K9me2 CUT&Tag-seq in ES2 cells knocked down LSD1 with shRNA. We found that the enrichment of the transcription-repressive mark H3K9me2 at promoter regions was slightly increased after LSD1 knockdown, while H3K4me2 changes were virtually absent at the LSD1-bound regions after LSD1 suppression (Fig. 5b), suggesting that H3K9me2 changes probably account for the downregulated gene expression after LSD1 inhibition. Next, we examined whether LSD1 impacted global chromatin accessibility using

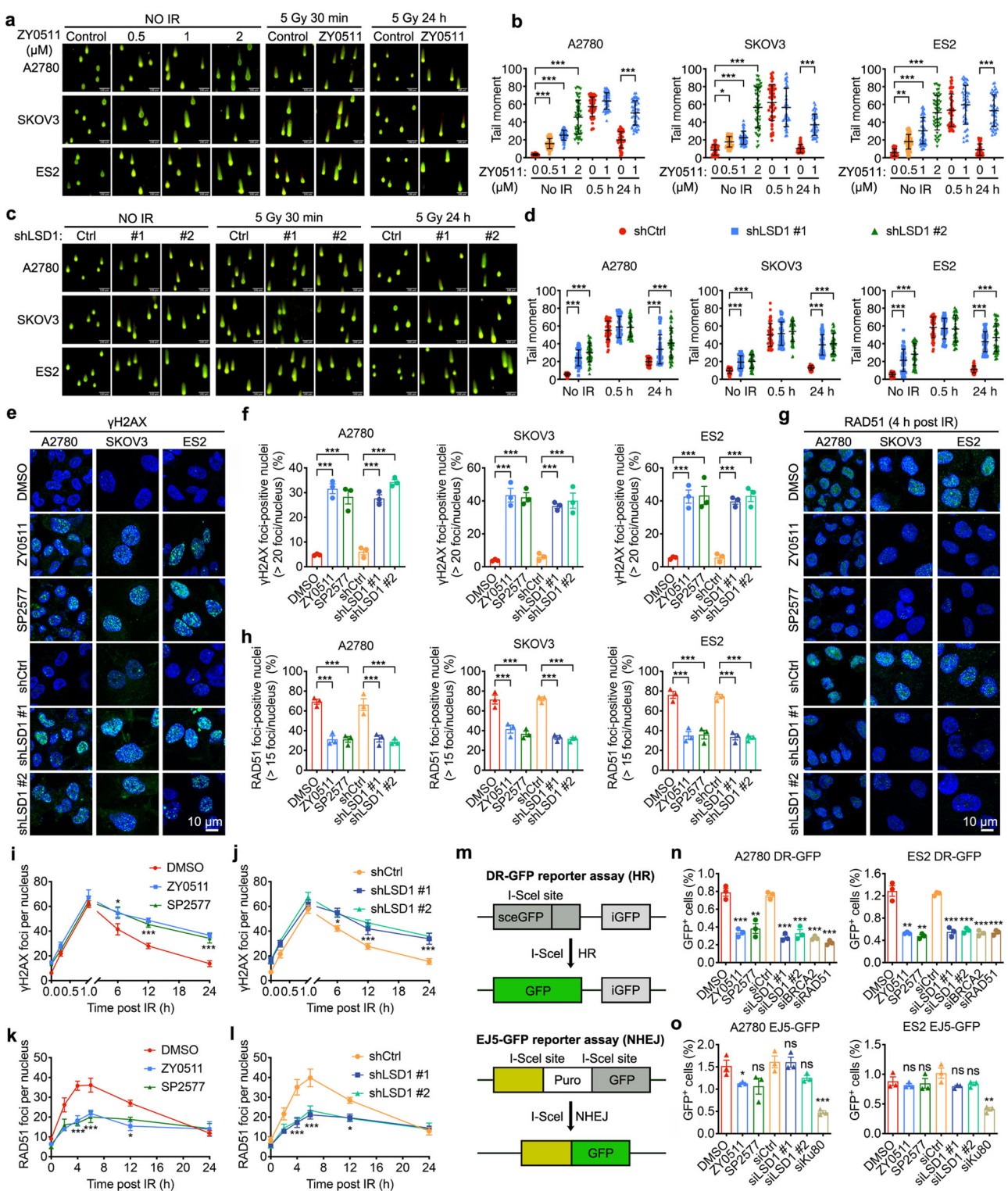

assay for transposase-accessible chromatin with sequencing (ATAC-seq). Both LSD1i and LSD1 knockdown (shLSD1) decreased chromatin accessibility (Fig. 5c). Of note, peak visualization analysis using integrative genomics viewer (IGV) plots showed that LSD1 bound at promoter regions of BRCA1/2 and RAD51 (Fig. 5d). LSD1 inhibition decreased LSD1 and increased H3K9me2 binding in gene promoter regions of BRCA1/2 and RAD51 but had no obvious effect on H3K4me2 binding in these gene promoter regions. In addition, LSD1i and LSD1 knockdown (shLSD1) decreased chromatin accessibility at these regions (Fig. 5d). Together, these data suggest that the loss of LSD1

activity might inhibit HR by promoting H3K9 methylation-dependent transcription repression of HR genes.

Consistent with genome-wide studies, these results were also validated by chromatin immunoprecipitation followed by qPCR (chromatin immunoprecipitation (ChIP)-qPCR) in A2780, SKOV3, and ES2 cells (Fig. 5e). ChIP-qPCR of LSD1 antibody, with primers located at BRCA1/2, and RAD51 promoter, demonstrated LSD1 associated with these HR factors promoter, which was significantly decreased with LSD1i (ZY0511) treatment. In addition, ZY0511 treatment markedly increased the enrichment of H3K9me2, but modestly affect H3K4me2

**Fig. 4 | LSD1 inhibition suppresses HR and increases DNA DSBs.**
**a**–**d** Representative images (**a**, **c**) and quantification (**b**, **d**) of neutral comet assays in A2780, SKOV3 and ES2 cells treated with indicated ZY0511 for 48 h or LSD1 knockdown (shLSD1) treatment after 5 Gy ionizing radiation (IR) treatment. Scale bar, 100 μm. Data represent mean ± SEM (unpaired two-tailed Student's *t* test). The experiments were repeated three times. **e**, **f** Representative images (**e**) and quantification (**f**) of γH2AX-foci staining performed in A2780, SKOV3 and ES2 cells with or without 1 μM LSD1i (ZY0511 or SP2577) for 48 h or LSD1 knockdown (shLSD1) treatment. Green, γH2AX; blue, DAPI. Scale bar, 10 μm. Data represent mean ± SEM of three biologically independent experiments (unpaired two-tailed Student's *t* test). **g**, **h** Representative images (**g**) and quantification (**h**) of and of RAD51 nuclear foci in A2780, SKOV3 and ES2 cells with or without 1 μM LSD1i (ZY0511 or SP2577) for 48 h or LSD1 knockdown (shLSD1) treatment at 4 h after 2 Gy IR treatment.

Green, RAD51; blue, DAPI. Scale bar, 10 μm. Data represent mean ± SEM of three biologically independent experiments (unpaired two-tailed Student's *t* test). **i**–**l** Quantification of γH2AX foci and RAD51 foci per nucleus at the indicated time points after 2 Gy IR treatment in ES2 cells treated with or without 1 μM LSD1i (ZY0511 or SP2577) for 48 h (**i**, **k**), or shRNA suppression of LSD1 (**j**, **l**). Data represent mean ± SEM of three biologically independent experiments (two-way ANOVA). **m** Schematic illustration of the GFP-based HR reporter assay (DR-GFP) and NHEJ reporter assay (EJ5-GFP). iGFP, internal GFP repeat. **n**, **o** Quantification of HR and NHEJ using DR-GFP and EJ5-GFP reporter assay, respectively. Data represent mean ± SEM of three biologically independent experiments (unpaired two-tailed Student's *t* test). ns, not significant, $p > 0.05$; \**p < 0.05*; \*\**p < 0.01*; \*\*\**p < 0.001*. Source data and exact *p* values are provided as the Source Data file.

at these gene promoters (Fig. 5e). The LSD1 K661A mutant has been widely used as a catalytically inactive LSD1 in in vitro and in vivo experiments. However, recent studies have shown that LSD1(K661A) mutant retains demethylase activity on nucleosome substrates to some extent, while the LSD1(A539E/K661A) double mutation completely abrogates LSD1 enzymatic activity[48]. To further evaluate whether LSD1 enzymatic activity is required for H3K9me2 demethylation at HR gene loci, we reconstituted LSD1 knockdown OC cells with wild-type LSD1 (LSD1-WT) or catalytically inactive LSD1 (LSD1-K661A and a double mutant LSD1-A539E/K661A (LSD1-DM)). LSD1 knockdown-induced increase in H3K9me2 level at BRCA1/2 and RAD51 was abolished by restored expression of LSD1-WT, but not catalytically inactive LSD1 (either LSD1-K661A or LSD1-DM). Neither LSD1 knockdown nor restored expression of either LSD1-WT or catalytically inactive LSD1 had any overt effect on the levels of H3K4me2 in these examined gene loci (Fig. 5f, g). Most importantly, knockdown of endogenous LSD1 decreased mRNA levels of BRCA1/2 and RAD51, and this effect was reversed by restored expression of wild-type LSD1 but not catalytically inactive LSD1 (Fig. 5h, i). Correspondingly, we observed that ectopic expression of LSD1-WT, but not catalytically inactive LSD1, restored the protein expression of BRCA1/2 and RAD51 by western blot analysis (Fig. 5j). Moreover, overexpression of LSD1-WT, but not catalytically inactive LSD1, restored cell HR competence and the cell sensitivity to olaparib induced by LSD1 knockdown (Fig. 5k, l), demonstrating the importance of canonical demethylase-dependent functions of LSD1. Once again, we reconstituted LSD1 knockout OC cells with LSD1-WT, LSD1-K661A, and LSD1-A539E/K661A and recapitulated the above rescue experiments consistent with the results observed in LSD1 knockdown OC cells (Supplementary Fig. 8a–g).

Collectively, these data support the contention that BRCA1/2 and RAD51 are direct targets of LSD1, and LSD1 mainly regulates these gene transcription dependently of its canonical demethylase function.

## LSD1 inhibition enhances sensitivity of OC cells to PARPi

PARPi were developed to capitalize on synthetic lethality with HRD[6]. Given LSD1 inhibition induced HRD in OC cells, we hypothesized that downregulation of HR proteins by LSD1 inhibition would hypersensitize OC cells to PARPi like olaparib, niraparib, and rucaparib. Indeed, we observed dramatic sensitivity to the PARPi drugs (olaparib, niraparib, and rucaparib) in OC cells lacking LSD1 expression (Fig. 6a). Furthermore, we examined the antiproliferative effects for the combination of ZY0511 and PARPi using CCK8 assay. A Bliss analysis of potential synergy found that the combination of ZY0511 and PARPi treatment was synergistic at several concentrations in A2780, SKOV3, and ES2 cells (Fig. 6b). Although clinical PARPi can be ranked by their ability to trap PARP (from the most to the least potent: niraparib > olaparib ≈ rucaparib)[6], LSD1 inhibitors synergized with PARP inhibitors with similar combination indices regardless their PARP trapping activity. To demonstrate generalizability, we additionally assessed SP2577, another clinical candidate LSD1 inhibitor, and demonstrated that SP2577 also exhibited similar patterns of synergy with PARP

inhibitors (Supplementary Fig. 9a). Moreover, consistent with our short-term cell viability assay, in a long-term colony formation assay, ZY0511 treatment sensitized A2780 cells to increasing concentration of PARPi (Fig. 6c–e). In contrast, the combination was not synergistic in non-tumorigenic HOSEpiC and IOSE80 (Fig. 6b).

## LSD1 inhibition enhances PARPi-induced DNA damage and apoptosis

Next, we performed comet assay to directly examine whether LSD1i enhances PARPi-induced DNA DSBs in OC cells. We observed LSD1i (ZY0511) or PARPi (olaparib, niraparib, or rucaparib) monotherapy modestly induced DNA damage, whereas the combination prominently increased accumulation of damaged DNA in A2780 and ES2 cells (Fig. 7a, b). Consistent with this data, the numbers of γH2AX-positive foci were increased in cells subjected to combination treatment compared with cells treated with PARPi or ZY0511 alone in A2780 and ES2 cells (Fig. 7c, d).

Given that unrepaired DSBs can trigger apoptosis, we measured annexin V-positive cells to determine whether the combination of LSD1i and PARPi induces greater levels of apoptosis. LSD1i (ZY0511) combined with PARPi treatment induced significantly higher levels of apoptosis than either single agent (Fig. 7e, f). In line with these synergistic effects, we observed enhanced apoptotic cells induced by PARPi when combined with shLSD1 (Fig. 7g, h). Furthermore, combined LSD1i (ZY0511) and PARPi treatment caused an increase in the protein expression of the apoptotic markers cleaved PARP and cleaved caspase-3/9 as compared with ZY0511 or PARPi alone (Fig. 7i).

## LSD1 inhibition sensitizes HR-proficient OC cells to PARPi treatment in vivo

Based on the synergy of LSD1i and PARPi in vitro, we investigated LSD1 inhibition and PARPi combinations in different in vivo models. We evaluated the effects of combined LSD1i (ZY0511) and PARPi treatment versus the respective single agents alone on the growth of three subcutaneous tumor xenografts in mice. The combination of LSD1i (ZY0511) and PARPi (olaparib, niraparib, and rucaparib) resulted in a significant inhibition of tumor growth in A2780, SKOV3, and ES2 subcutaneous xenograft models (Fig. 8a–d). Mean body weights were not significantly different between the combination treatment and single drugs alone, suggesting that all treatment protocols were well tolerated (Fig. 8e).

In addition, we assayed the effect of PARPi on the growth of tumor xenografts formed in immunodeficient mice by subcutaneously injection of SKOV3 and ES2 expressing a non-targeting control (shCtrl) or an shRNA to LSD1 (shLSD1). We observed robust suppression of tumor growth by the PARPi in shLSD1 tumor models compared with that in shCtrl tumor models (Fig. 8f, g). Furthermore, we constructed patient-derived xenograft (PDX) models by using HR-proficient human OC tissues and assessed the combination efficacy of LSD1i (ZY0511) and olaparib against tumor growth. The combination markedly inhibited tumor growth to a much greater degree than either

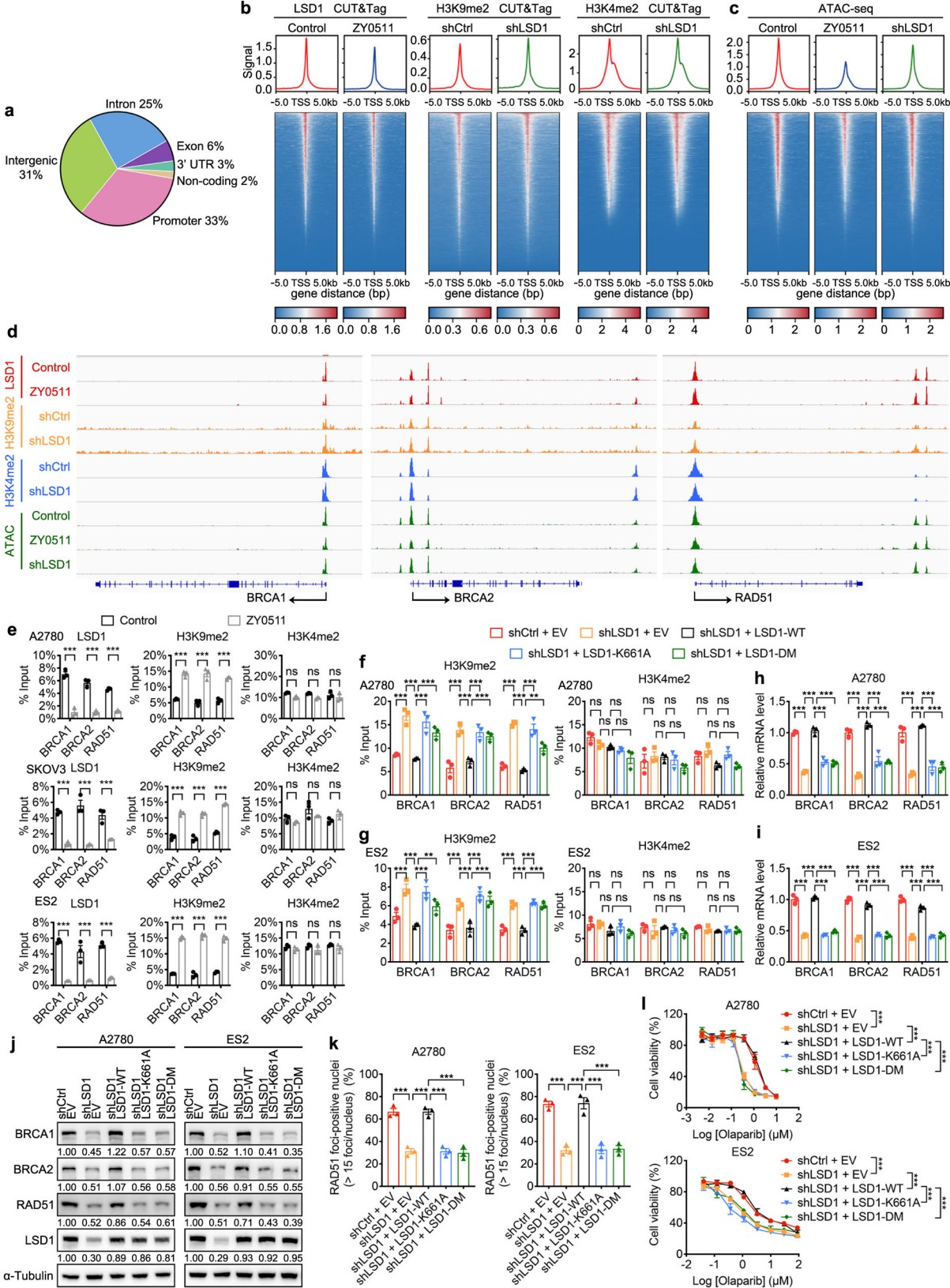

compound alone (Fig. 8h, i). Indeed, in the PDX model, ZY0511 resulted in 52.3% TGI, olaparib showed modestly effect at 40.6% TGI and the combination treatment resulted in 77.4% TGI. In line with the nude mice, the combination of ZY0511 and olaparib was also well tolerated in NCG mice without different mean body weights among the four groups (Fig. 8j).

Patient-derived organoids (PDOs) have recently emerged as robust preclinical models and have the advantage of mimicking the biological characteristics of the original patient tumors both phenotypically and genetically[49]. We therefore tested the combination effects in two PDOs derived from OC patients, HR-proficient (KO-25127) and HR-deficient (KO-96412), respectively. The results showed

**Fig. 5 | LSD1 binds BRCA1, BRCA2, and RAD51 gene promoter, regulating these gene transcription dependently of its canonical demethylase function. a** Pie chart showing the genomic distribution of LSD1 peaks based on RefSeq. **b** Levels of LSD1, H3K9me2, and H3K4me2 bound at the TSS of peaks in ES2 cells, as measured by CUT&Tag-seq analysis. Transcription start site, TSS. **c** Levels of ATAC bound at the TSS in ES2 cells, as measured by ATAC-seq analysis. **d** IGV plot showing the distributions of LSD1, H3K9me2, H3K4me2, and ATAC-seq peaks binding in the promoters of BRCA1, BRCA2, and RAD51 in ES2 cells. **e** ChIP-qPCR analysis showing the enrichment levels of LSD1, H3K9me2 and H3K4me2 at the BRCA1, BRCA2 and RAD51 gene promoter in A2780, SKOV3 and ES2 cells. Data represent the percent of total chromatin input ±SEM of three biologically independent experiments; unpaired two-tailed Student's $t$ test; ns, not significant). **f, g** ChIP-qPCR analysis showing the enrichment levels of H3K9me2 and H3K4me2 at the BRCA1, BRCA2 and RAD51 gene promoter. Data represent the percent of total chromatin

input ± SEM of three biologically independent experiments (unpaired two-tailed Student's $t$ test; ns, not significant). **h, i** RT-qPCR analysis of indicated gene expression. GAPDH was used as the loading control. Data represent mean ± SEM of three biologically independent experiments (unpaired two-tailed Student's $t$ test). **j** Western blot analysis of indicated proteins. α-Tubulin was used as the loading control. Numbers below western blot panels represent relative quantification of the respective bands normalized to loading control by densitometry. **k** Quantification of RAD51 nuclear foci at 4 h after 2 Gy IR treatment in A2780 and ES2 cells. Data represent mean ± SEM of three biologically independent experiments (unpaired two-tailed Student's $t$ test). **l** Cell viability in response to olaparib in A2780 and ES2 cells. Data represent mean ± SEM of three biologically independent experiments (two-way ANOVA). ns, not significant, $p > 0.05$; *$p < 0.05$; **$p < 0.01$; ***$p < 0.001$. Source data and exact $p$ values are provided as the Source Data file.

that LSD1i (ZY0511) markedly potentiated the killing effects of olaparib in HR-proficient PDOs, but not HR-deficient PDOs (Supplementary Fig. 10a, b). This implicated the therapeutic potential of LSD1 inhibition in combination with PARP inhibitors in HR-proficient OC patients, consistent with the results observed in vitro and in vivo tumor xenografts (Figs. 6 and 8). We further confirmed the combination effect in a syngeneic OC mouse model, which consists of intact functional immune system. Similar to our observations in cell-derived xenografts and human PDX models, the combination of LSD1i (ZY0511) and PARPi (olaparib) markedly resulted in 69.8% TGI than either compound alone (Supplementary Fig. 10c, d).

Endpoint studies determined that expression of BRCA1/2 and RAD51 were reduced in tumors treated with LSD1i (ZY0511) or the combination, whereas protein levels of DNA damage (γH2AX) and apoptosis (cleaved caspase 3 and cleaved PARP) were elevated in the combination group compared with either single drug alone (Fig. 8k). Furthermore, IHC of PDX xenograft tumors at study termination recapitulated the in vitro studies. ZY0511 increased γH2AX and cleaved caspase 3, which were further increased by combination with olaparib. And compared with single drug alone groups, there were reductions in proliferation (Ki67-positive cells) in combination group (Fig. 8l, m). We then assessed the impact of LSD1i (ZY0511) treatment on the expression of BRCA1/2 and RAD51 in normal tissues isolated from ZY0511-treated mice. However, no significant effect of ZY0511 treatment was observed in certain normal tissues, including the liver and spleen. These data suggest that ZY0511 specifically down-regulates BRCA1/2 and RAD51 in tumor cells but with no corresponding effect on these gene expression in non-malignant and healthy tissue (Supplementary Fig. 10e). Additionally, ZY0511 treatment did not significantly alter the levels of H3K4me2 and H3K9me2 in the liver, spleen and kidney of mice, whereas ZY0511 treatment resulted in an increase in H3K4me2 and H3K9me2 amount specifically in the tumor of xenografts (Supplementary Fig. 10f). These findings suggest that LSD1 may play a role in regulating DNA damage repair in a cell- or tissue-specific manner. These observations provide insights into the lack of synergy between PARPi and LSD1i combination treatments across normal cell lines and the similar mean body weights of mice under different treatments. The cell/tissue-specific regulation of DNA damage repair by LSD1 may, at least in part, contribute to these outcomes.

To further evaluate the safety of the combination, we performed a toxicity analysis of ZY0511 with olaparib. No changes in red blood cells, white blood cells, platelets or hemoglobin were detected. Blood biochemical panels did not reveal changes (Supplementary Fig. 11a, b). H&E staining showed no lesions in the main organs among the four groups (Supplementary Fig. 11c).

## Discussion

Selectively impairing HR of cancer cells has been proven to be an effective therapeutic strategy in the case of PARP inhibitors. In this regard, several approaches have been designed and evaluated to

induce sensitivity of HR-proficient cancer cells to PARPi in preclinical and early clinical trials[9,11–20]. Here, we demonstrate that LSD1 is an important regulator for OC. Importantly, LSD1 inhibition decreased BRCA1/2 and RAD51 transcription and induced HRD, leading to synergistical effect of PARPi with LSD1i in HR-proficient OC cells both in vitro and in vivo. Our results identify LSD1i sensitize HR-proficient tumors to PARPi by converting HR-proficient tumors to HRD tumors, thus supporting the clinical applications of PARPi to HR-proficient patients and providing a significant advancement in the treatment of OC.

Dysregulation of LSD1 and other different histone modifications and epigenetic effectors is common in cancer[50]. This is the reason why molecules targeting epigenetic traits have been tested as single drugs (monotherapy) or in combination with cytotoxic chemotherapy for OC treatment[51,52]. As for LSD1i, some of them have already been developed and entered human clinical trials for treatment of AML and SCLC, but, at the best of our knowledge, few for OC[29]. In this study, we demonstrate that LSD1 is a potentially effective therapeutic target in OC through both genetic and pharmacologic approaches of disrupting LSD1 activity. The LSD1i, named ZY0511, suppresses OC growth in vitro and multiple in vivo models. The applications of ZY0511 in human colorectal and cervical cancer either alone or in combination with 5-fluorouracil further support to therapeutic efficacy of ZY0511[31,32]. However, more detailed experiments like orthotopic transplantation tumor models and safety studies in larger animals like beagle will better define the clinical translatability of ZY0511.

PARPi are selectively active in cells with HRD caused by mutations in BRCA1/2 and other DNA repair pathway members like RAD51. Inhibition of HR *via* pharmacological targeting of epigenetic regulators were used to induce sensitivity to PARPi. In this study, we identified that LSD1 directly binds BRCA1/2 and RAD51 gene promoter. LSD1 inhibition increased the level of transcription-repressive mark H3K9me2 in BRCA1/2 and RAD51 promoter, thus downregulating transcription of these genes and subsequently inducing HR impairment in HR-proficient cancer. In agreement with down-regulated gene expression of BRCA1/2 after loss of LSD1, it has been reported that interference of human LSD1 mRNA by siRNA decreases expression of human BRCA1 and BRCA2 mRNA in LNCaP and C4-2B prostate cancer cell lines based on microarray results[53]. In addition, in agreement with increased enrichment of H3K9me2 and further interfering HR gene expression, it has been reported that genetic depletion or pharmacological inhibition of LSD1 robustly triggers cellular senescence by abolishing the process of H3K9 demethylation[54]. Indeed, activation of the DNA damage response has been linked to oncogene-induced senescence[55]. Furthermore, the effect of H3K9me2 changed by LSD1 inhibition was almost completely abolished by restored expression of wild-type LSD1 but not catalytically inactive LSD1, suggesting enzymatic activity of LSD1 is essential for H3K9me2 demethylation at BRCA1/2 and RAD51 gene promoter loci. Moreover, overexpression of LSD1-WT, but not catalytically inactive LSD1, restored HR gene

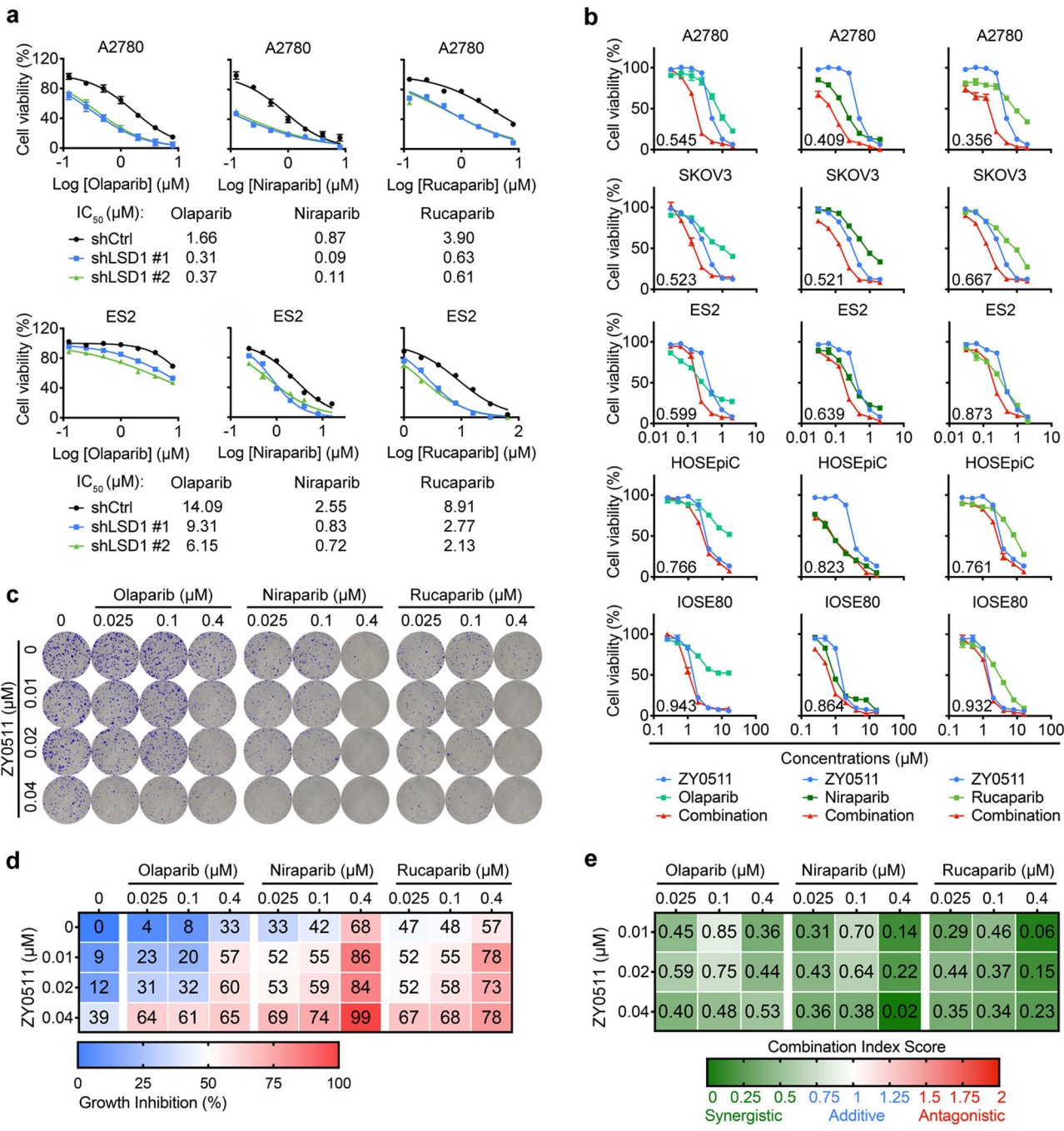

**Fig. 6 | LSD1 inhibition enhances PARPi sensitivity in OC cells. a** Cell viability in response to the PARPi (olaparib, niraparib, or rucaparib) in A2780 and ES2 cells with or without knockdown of LSD1. The IC$_{50}$ values were calculated using GraphPad software. Data represent mean ± SEM of three biologically independent experiments. **b** Dose-response curves of ZY0511 or PARPi (olaparib, niraparib, or rucaparib) alone or combined in A2780, SKOV3, and ES2 cell lines or in normal ovarian epithelial HOSEpiC and IOSE80 cells lines treated with varying concentrations of ZY0511 and PARPi for 72 h. Combination index (CI) was calculated using CompuSyn software with the Chou-Talalay equation. Data represent mean ± SEM of three biologically independent experiments. **c** Representative images of colony formation assay for A2780 cells treated with LSD1i (ZY0511), PARPi (olaparib, niraparib, or rucaparib), or their combination as indicated. **d** Percentage inhibition at each concentration of LSD1i (ZY0511), PARPi (olaparib, niraparib, or rucaparib), or their combination in A2780 cells. Data represent mean ± SEM of three biologically independent experiments. **e** Combination index (CI) scores for A2780 cells treated with LSD1i (ZY0511) in combination with PARPi (olaparib, niraparib, or rucaparib) at the indicated concentrations. Each CI score represents data from three biologically independent experiments. ns, not significant, $p > 0.05$; *$p < 0.05$; **$p < 0.01$; ***$p < 0.001$. Source data are provided as the Source Data file.

expression, HR function, and cell survival in OC cells, further demonstrating that the canonical demethylase functions of LSD1 is essential for OC cells. Besides LSD1, enzymes involved in histone lysine methyltransferases and lysine demethylases, such as SUV39H1 (KMT1A), SETDB1 (KMT1E), KDM4B, and KDM5A, have been found to be involved in the DDR, suggesting that their inhibitors have combination therapy potential with PARPi for cancer treatment[56,57]. However, the development of inhibitors targeting these epigenetic targets is still in preclinical research, and there have been no inhibitors entering clinical oncology studies yet[51]. Our results reveal that LSD1 inhibitor suppress HR repair and sensitize HR-proficient OC to PARPi, at least in part through impairing BRCA1/2 and RAD51 gene

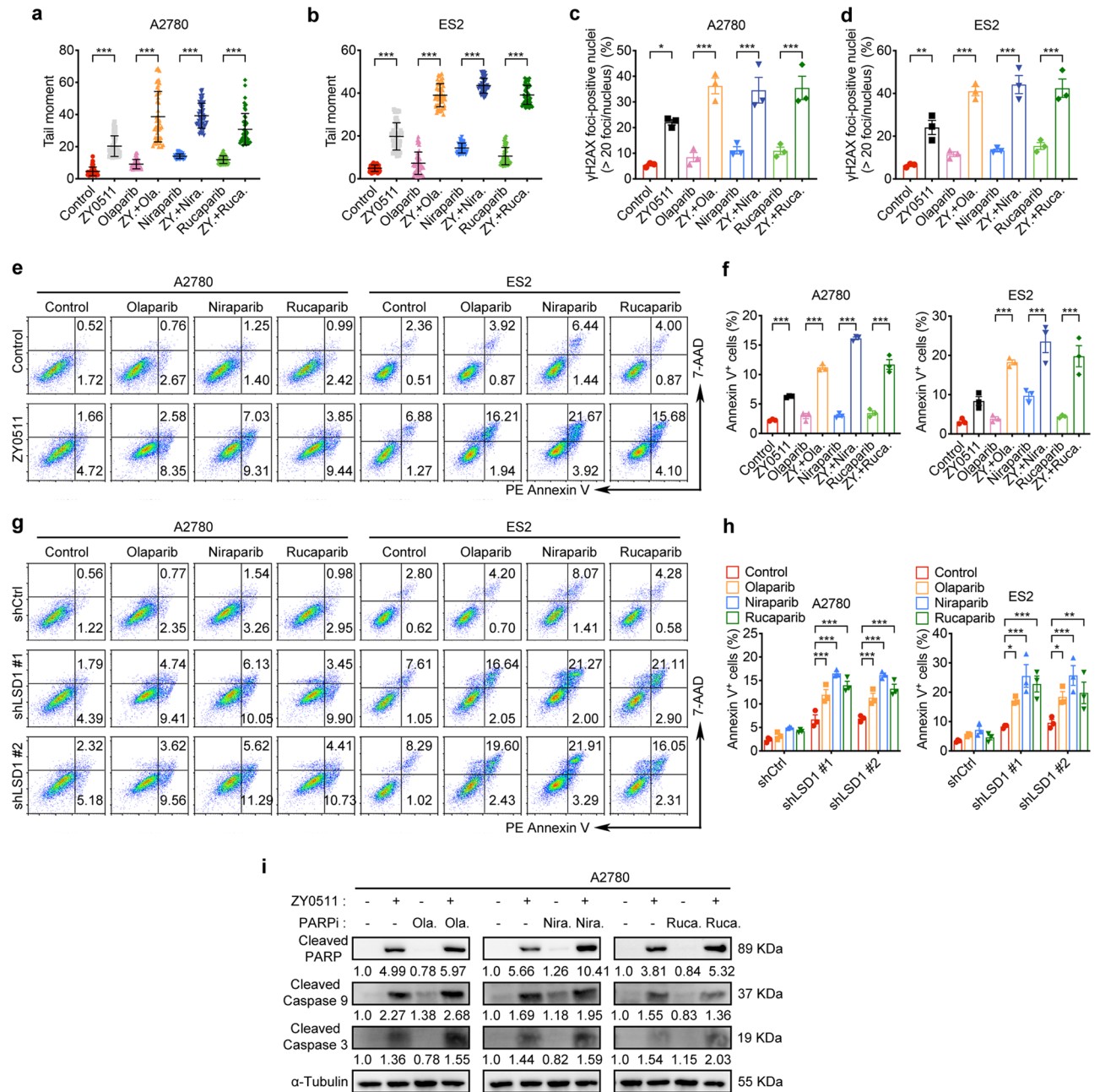

**Fig. 7 | LSD1 inhibition enhance PARPi-induced DNA damage and apoptosis.**
**a**, **b** Quantification of neutral comet assays in A2780 cells (**a**) treated with vehicle, LSD1i (1 μM ZY0511), olaparib (4 μM), niraparib (4 μM), rucaparib (4 μM) alone or combined for 48 h, and in ES2 cells (**b**) treated with vehicle, LSD1i (1 μM ZY0511), olaparib (20 μM), niraparib (10 μM), rucaparib (20 μM) alone or combined for 48 h. Data represent mean ± SEM (unpaired two-tailed Student's *t* test). The experiments were repeated three times. **c**, **d** Quantification of γH2AX foci in A2780 cells (**c**) and ES2 cells (**d**) treated with same concentration for 48 h as shown in Fig. 7a, b. Data represent mean ± SEM of three biologically independent experiments (unpaired two-tailed Student's *t* test). **e**, **f** Representative images (**e**) and quantification (**f**) of cell apoptosis analysis in A2780 and ES2 cells treated with same concentration for 48 h as shown in Fig. 7a, b. Annexin V-positive cells were analyzed by flow cytometry after treatment. Data represent mean ± SEM of three biologically

independent experiments (unpaired two-tailed Student's *t* test).
**g**, **h** Representative images (**g**) and quantification (**h**) of cell apoptosis analysis in shLSD1-expressing and shCtrl-expressing A2780 and ES2 cells. A2780 cells were treated with olaparib (4 μM), niraparib (4 μM), or rucaparib (4 μM), while ES2 cells were treated with olaparib (20 μM), niraparib (10 μM), or rucaparib (20 μM). Annexin V-positive cells were analyzed by flow cytometry at 48 h after treatment. Data represent mean ± SEM of three biologically independent experiments (unpaired two-tailed Student's t test). **i** Western blot analysis of indicated proteins in A2780 cells treated with vehicle, 1 μM ZY0511, 4 μM PARPi (olaparib, niraparib or rucaparib), or a combination for 48 h. Numbers below western blot panels represent relative quantification of the respective bands normalized to loading control by densitometry. ns, not significant, $p > 0.05$; *$p < 0.05$; **$p < 0.01$; ***$p < 0.001$. Source data and exact *p* values are provided as the Source Data file.

transcription dependently of canonical demethylase function of LSD1, supporting the potential strategy that histone methylation regulators in combination with PARPi for HR-proficient OC therapy.

Our study was limited to focusing on LSD1i-targeted genes which are directly involved in DNA damage repair. Although we believe that

direct transcriptional repression of HR genes is the dominant mechanism, we cannot exclude other indirect mechanisms that may also cooperate or contribute to this synergistic effect. For example, knockout of LSD1 or LSD1i (HCI-2509) decreases the expression of c-MYC protein[58,59], and the dual class I HDAC and LSD1 inhibitor known

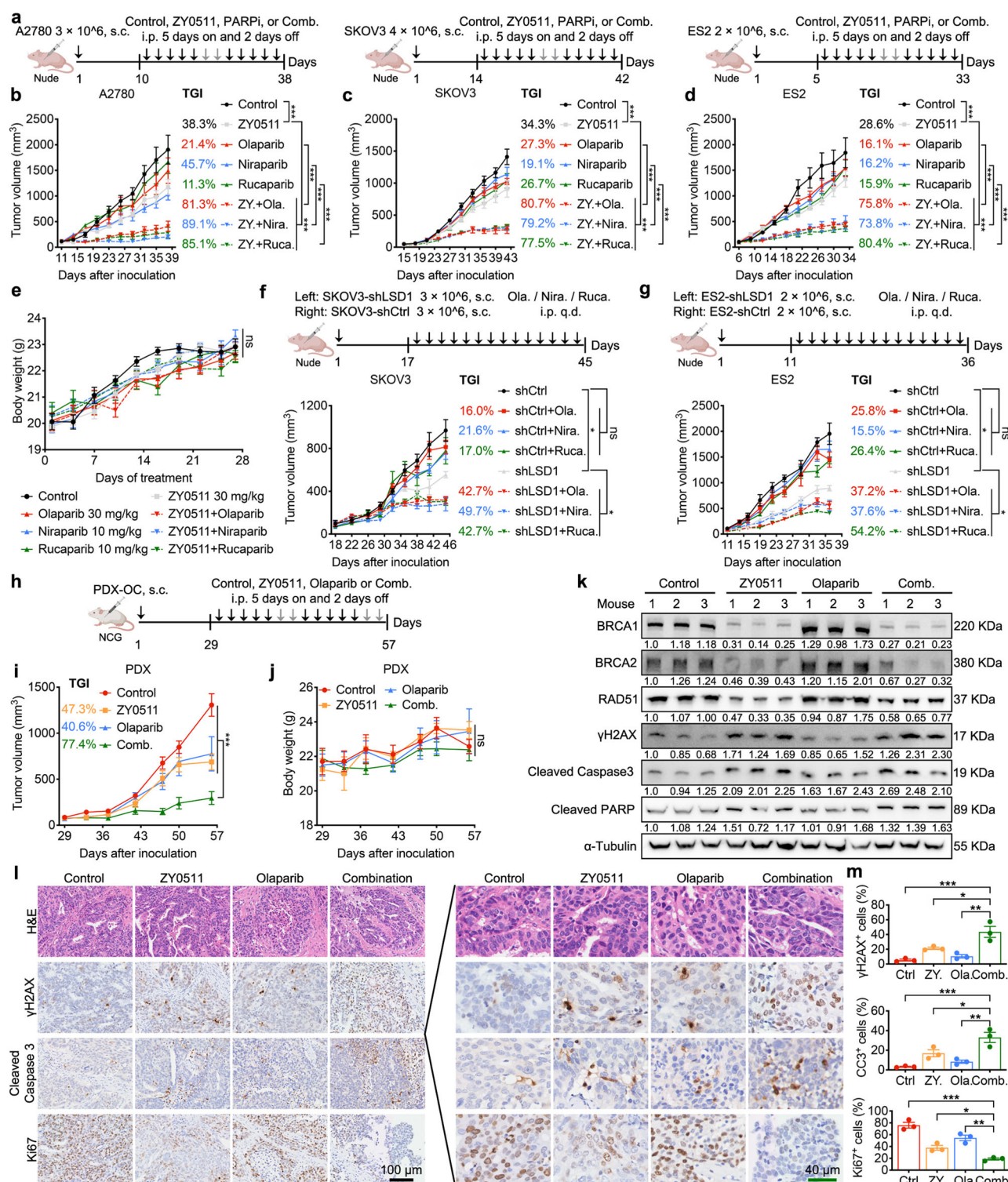

as Domatinostat reduces expression of FOXM1[60]. Given that both c-MYC and FOXM1 regulate the genes that control DSB repair[61], c-MYC or FOXM1 reduced by LSD1 inhibition might also indirectly contribute to reduction of HR gene expression and subsequently result in HR deficiency. Moreover, previous studies have suggested that LSD1 plays at least an indirect role in DDR by demethylating p53, thereby inhibiting p53-mediated transcriptional activation and apoptosis[62]. Besides, LSD1i may also directly influence the DNA damage response by disrupting chromatin signaling and impairing HR factor recruitment. Sulkowski et al. have reported that oncometabolites suppress HR *via* direct inhibition of the lysine demethylase KDM4B, leading to

global elevation of H3K9me3 chromatin marks[57], which impedes Tip60 recruitment to DSBs, causing HR failure, persistence of DSBs, and ultimately PARPi sensitization. As H3K9me2 is the substrate of H3K9me3, we found that LSD1 inhibition resulted in increase of not only H3K9me2 but also H3K9me3 protein levels (Supplementary Fig. 3d), in agreement with previous observation[54]. Therefore, we speculate that Tip60 might also be impaired at DNA breaks, with diminished recruitment of downstream repair factors and impeded HR activity, because of H3K9me3 accumulation caused by LSD1 inhibition. Therefore, it is still necessary to further characterize how other mechanisms contribute to the synergistic effect of LSD1i and PARPi in

**Fig. 8 | LSD1 inhibition sensitizes HR-proficient tumors to PARPi treatment in vivo. a** Schematic diagram of A2780, SKOV3 and ES2 subcutaneous tumor model and drug delivery. Mice were treated with vehicle (30 mg/mL PEG4000 plus 12 mg/mL Tween 20 in water and 5% DMSO plus 30% PEG300 in water intraperitoneally), ZY0511 (intraperitoneally 30 mg/kg), PARPi (olaparib 30 mg/kg, niraparib 10 mg/kg and rucaparib 10 mg/kg intraperitoneally), or a combination of ZY0511 and PARPi as indicated. Schematic diagram was created with BioRender.com. **b–d** Tumor volume of mice bearing A2780 (**b**), SKOV3 (**c**) and ES2 (**d**) subcutaneous xenografts and treated with vehicle or drugs showed in **a**. Data represent mean ± SEM (*n* = 6 mice per group; two-way ANOVA). **e** Body weight curves of mice treated with vehicle or drugs showed in **a**. Data represent mean ± SEM. (*n* = 18 mice per group from A2780, SKOV3 and ES2 subcutaneous xenografts; two-way ANOVA). **f**, **g** Tumor volume of mice bearing SKOV3 subcutaneous xenografts (**f**) and ES2 subcutaneous xenografts (**g**). Data represent mean ± SEM (*n* = 6 mice per group; two-way ANOVA). **h** Schematic diagram of PDX model and drug delivery. Created with BioRender.com. **i**, **j** Tumor volume (**i**) and body weight curves (**j**) of mice bearing PDX and treated with vehicle or drugs showed in **h**. Data represent mean ± SEM (*n* = 6 mice per group; two-way ANOVA). **k–m** Western blot analysis (**k**) and representative images (**l**) and quantification (**m**) of IHC of indicated proteins in tumor tissues from PDX subcutaneous xenografts. α-Tubulin was used as loading control. Numbers below western blot panels represent relative quantification of the respective bands normalized to loading control by densitometry. CC3, Cleaved Caspase 3. Scale bars, 100 μm (black), 40 μm (green). Data represent mean ± SEM of three random fields of view from three different mice (one-way ANOVA). ns not significant, *p* > 0.05; *\**p* < 0.05; **\*\**p* < 0.01; **\*\*\**p* < 0.001. Source data and exact *p* values are provided as the Source Data file.

some cellular contexts. Moreover, although our study identifies LSD1 selectively and directly regulates H3K9 demethylation at HR gene loci, we cannot exclude the indirect regulation through H3K9 modulators. SUV39H1/2, SETDB2, G9a and GLP mainly target H3K9 for methylation, whereas KDM3, KDM4, PHF2 and PHF8 mainly target H3K9 for demethylation. In addition to its demethylation of histone lysine residues, LSD1 is able to demethylate nonhistone proteins and regulate protein stability through demethylase-independent activity[27,63]. LSD1 can also interact with various chromatin-modifying enzymes and transcription factors, forming complexes that regulate gene expression[64], suggesting that LSD1 has the potential to indirectly regulate H3K9me2 levels by modulating the activity or recruitment of these H3K9 modulators.

LSD1 has been reported to repress and activate transcription by mediating histone H3K4me1/2 and H3K9me1/2 demethylation[21,28], respectively, whereas the molecular mechanism that underlies this dual substrate specificity has remained unknown. Consistent with our observation, LSD1 co-occupied with E2F1, demethylated H3K9me2, and promoted the LSD1-E2F1 co-target genes expression, but has no effects on H3K4me2, thereby promoting S-phase entry and tumorigenesis in prostate cancer cells[56]. These findings indicate that the manner in which LSD1 selectively demethylates H3K4me or H3K9me depends on the cell type, developmental stage, or the phases of the cell cycle. LSD1 was also characterized as histone demethylase dedicated to removing mono- and dimethylated H3K9me2, where AR or ER is required[24–26]. Interestingly, a recent study reported that LSD1 isoform, LSD1 + 8a mediates H3K9me2 demethylation in collaboration with supervillin (SVIL) but not H3K4me2 at its target promoters and regulates neuronal differentiation in neuronal cells[65]. These results suggest that LSD1-associated factors, such as AR[24], ER[25], SVIL[65] and Proline-, glutamic acid-, and leucine-rich protein-1 (PELP1)[66], play a crucial role in tipping H3K4me1/2 demethylation toward H3K9me1/2 demethylation. In line with these findings, our study identifies LSD1 selectively regulates H3K9me2 demethylation at HR gene loci. However, it will be interesting to determine which LSD1 isoforms are functioning, and which LSD1-associated factors are operating in OC cells. In addition, it is possible that the LSD1 targeting H3K4 and H3K9 are in different protein complexes. Identifying the context that affects LSD1's choice to demethylate H3K4 or H3K9 is also very interesting.

It also has been shown that HR-deficient cancers are often more sensitive to crosslinking agents including cisplatin than their HR-proficient counterparts and secondary mutations that restore BRCA function or HR-proficient subtype favor acquired platinum resistance. Thus, the elevated expression of LSD1 in OC samples was associated with BRCA1/2 wild-type status and increased chemoresistance to platinum (Supplementary Fig. 1e–h), indicating potential clinical utility of LSD1 as prognostic and predictive biomarker in OC. Given that deficiency of HR increases the sensitivity of cancer cells to treatment with DNA damage, our study also provides a strong rationale for clinical application of LSD1i in combination with DNA damage agents, such as DNA crosslinker (cisplatin), topoisomerase I inhibitors (irinotecan),

DNA replication targeting agents (doxorubicin) and radiotherapy, in other HR-proficient tumors. Moreover, acquired resistance to PARPi is an almost universal occurrence for OC that is initially HRD but acquires HR proficiency after PARPi treatment by multiple mechanisms, including secondary mutations that restore BRCA1/2 and RAD51 functions, and loss of expression of PARP1, 53BP1, or REV7[8]. Thus, LSD1i may resensitize PARPi-resistant cells to PARPi, and further studies by using PARPi-resistant models to investigate this potential strategy are needed to be performed.

Importantly, the in vivo studies did not show significant toxicity based on weight loss, hematological and hemato-biochemical parameters, and organ histopathological coefficients. This may, in part, be due to the cell/tissue-specific regulation of DNA damage repair by LSD1. The lack of synergy of this combination across a range of normal cell lines due to differential levels of replication stress and persistent DNA damage between normal and malignant cells further supports the potential for tolerability in patients. Although our in vivo data strongly support the efficacy of LSD1i in combination of either of three PARPi (olaparib, niraparib, or rucaparib), low toxicity of the combination were mainly conducted with olaparib. Besides, the usage of PARPi in the management of various OC patient populations are slightly different according to the newest EMSO guideline[67]. we have not demonstrated the activity and tolerability of the combination strategy in patients. Thus, the safety profile and therapy of the PARPi/LSD1i combination in patients carefully warrant exploration in human clinical trials.

It is worth noting that the mutants (LSD1-K661A and LSD1(A539E/K661A) double mutation) showed identical rescue gene expression and phenotypes. However, a previous study has shown that the LSD1(K661A) mutant retains demethylase activity on nucleosome substrates to some extent, while the LSD1(A539E/K661A) double mutation completely abrogates LSD1 enzymatic activity[48]. The difference in LSD1(K661A) demethylation activity may be attributed to different experimental conditions between biochemical and cellular assays. Our study suggests that the K661 residue is a critical catalytic site of LSD1 in OC cells. Moreover, further exploration of the key catalytic residues of LSD1 in vitro and in vivo is worth being examined in future studies.

In conclusion, our results demonstrate that inhibition of LSD1 induces HR deficiency through depletion of BRCA1/2 and RAD51 and sensitizes HR-proficient OC to PARP inhibition. Despite these limitations above, our findings provide a strong rationale for clinical application of PARPi in combination with LSD1i for patients with de novo or acquired resistance to PARPi.

## Methods

Details of experimental procedures, cell lines and transfection, animals, chemicals, antibodies, cell proliferation assay, colony formation assay, cell cycle analysis, apoptosis analysis, RT-qPCR, western blot analysis, HR and NHEJ reporter assays, neutral comet assay, immunofluorescence, in vivo tumor xenograft, PDOs culture and viability assay, immunohistochemistry, ChIP-qPCR, RNA-seq, CUT&Tag-seq, ATAC-seq, and GSEA analysis are included in Methods.

## Cell lines and transfection

The human OC cell lines A2780 (Cat. # 93112519) and COV362 (Cat. # 07071910) were purchased from Sigma. The human OC cell lines SKOV3 (Cat. # HTB-77), ES2 (Cat. # CRL-1978), and OVCAR3 (Cat. # HTB-161) were purchased from American Type Culture Collection (ATCC). The human OC cells Kuramochi (Cat. # JCRB0098) were purchased from Japanese Collection of Research Bioresources (JCRB) Cell Bank. The human ovarian epithelial cells HOSEpiC (Cat. #7310) were purchased from ScienCell Research Laboratories. The non-malignant human ovarian surface epithelial cells IOSE80 (Cat. # CTCC-400-0117) were purchased from Meisen Chinese Tissue Culture Collection. The mouse OC cells ID8 were kindly gifted by Dr. Zhou at Sichuan University. The A2780, OVCAR3, IOSE80, and HOSEpiC were cultured in RPMI 1640 medium (Gibco) with 10% fetal bovine serum (FBS). The SKOV3, ES2, COV362, Kuramochi, and ID8 cells were cultured in Dulbecco's modified Eagle's medium (DMEM) (Gibco) with 10% FBS. UWB1.289 cells were kindly gifted by Dr. Yu at Westlake University and were cultured RPMI 1640: MEGM (1: 1) with 3% FBS (Lonza, CC-3150). All cell lines were maintained in standard conditions at 37 °C and 5% $CO_2$. All human cell lines were authenticated by fingerprinting using short tandem repeat testing and were verified to be free of mycoplasma contamination.

A2780, SKOV3, and ID8 cells were transduced with a lentiviral vector containing a luciferase reporter together with the blasticidin resistance gene (then termed A2780-Luc, SKOV3-Luc, and ID8-Luc cells) and selected by Blasticidin S (10 µg/mL, Selleckchem, S7419) for establishment of intraperitoneal tumor model.

For short hairpin RNA (shRNA) experiments, shRNAs using to knockdown LSD1 were cloned into GV298 lentiviral vector (Genechem). The sequence targeting LSD1 (#1: 5′-CCACGAGTCAAACCTT-TATTT-3′; #2: 5′-GCAGCTCGACAGTTACAAA-3′) and a non-targeting control shRNA (TTCTCCGAACGTGTCACGT) were synthesized by Genechem (Shanghai, China). Viral packaging and infection of cells following the manufacture's recommended protocol. The viruses were collected and added to cells in the presence of polybrene (5 µg/mL) and replaced with fresh medium after 12 h. In 72 h, puromycin (2 µg/mL, Selleckchem, S7417) was added to culture medium for cell screening lasted for 1 week and culture medium was replaced with puromycin (1 µg /mL) every day.

To obtain LSD1 knockout clones using the CRISPR-Cas9 system, the sgRNA sequence targeting LSD1 (GTCGGACCAGCCGGCGCAAG (sgRNA #1) and CGCGGAGGCTCTTTCTTGCG (sgRNA #2)) were synthesized by GenScript. The cells were enriched by fluorescent-based sorting using a FACS Aria Sorp (BD Biosciences) and transferred into 96-well plates at - 1 cell per well after transfection 24 h. The candidate clones were analyzed by western blot and Sanger sequencing.

RNA interference (RNAi) transfection was carried out using Lipofectamine 2000 Reagent according to the manufacturer's instructions (Invitrogen, 11558019). Small-interfering RNA (siRNA) duplexes targeting LSD1 (#1: CCACGAGUCAAACCUUUAUUU; #2: GCAGCUCGA-CAGUUACAAA); targeting BRCA2 (GAAGAACAAUAUCCUACUA); targeting RAD51 (AAGGGAAUUAGUGAAGCCAAA); targeting Ku80 (GCGAGUAACCAGCUCAUAA), and a non-targeting control siRNA (CGUACGCGGAAUACUUCGA dTdT) were synthesized by RiboBio (Guangzhou, China)

## Animals

Five- to six-week-old female BALB/c nude mice and C57BL/6 mice were purchased from Charles River Laboratories (Beijing, China). Five- to six-week-old female immunodeficient NCG mice were purchased from GemPharmatech (Nanjing, China). The mice were housed in specific pathogen-free conditions with controlled temperature (22–26 °C), humidity (55 ± 5%), and a 12 h light/dark cycle, with 5 mice per cage. The animal experiments were performed in strict accordance with the People's Republic of China Legislation Regarding the Use and Care of Laboratory Animals. All protocols used in this study were approved by the Institutional Animal Care and Treatment Committee of Sichuan University in China (permit number: 20180106).

## Chemicals

ZY0511 was prepared according to the reported procedures[30] and dissolved in DMSO to yield 20 mM stock solutions and stored at −80 °C. SP2577 (S6722), olaparib (S1060), niraparib (S2741) and rucaparib (S4948) were obtained from Selleckchem (Shanghai, China). GSK2879552 for in vitro studies was dissolved in ddH$_2$O to yield 100 mM stock solutions, other compounds for in vitro studies were dissolved in DMSO to yield 40 mM stock solutions. All compounds were stored at −80 °C.

## Antibodies

The primary antibodies for western blot analysis were diluted as followed: rabbit anti-LSD1 (Cell Signaling Technology, 2139 S and 2184 S, 1:1000), rabbit anti-H3 (Abcam, ab1791, 1:5000), rabbit anti-H3K4me1 (Abcam, ab8895, 1:1000), rabbit anti-H3K4me2 (Abcam, ab32356, 1:1000), mouse anti-H3K9me1 (Abcam, ab8896, 1:1000), mouse anti-H3K9me2 (Abcam, ab1220, 1:1000), rabbit anti-H3K9me3 (Cell Signaling Technology, 13969 S, 1:1000), mouse anti-γH2AX (Millipore, 05-636, 1:1000), rabbit anti-ATM (Cell Signaling Technology, 2873 T, 1:1000), rabbit anti-phospho-ATM (Ser1981) (Cell Signaling Technology, 13050 T, 1:1000), rabbit anti-RAD51 (Cell Signaling Technology, 8875 S, 1:1000), rabbit anti-BRCA1 (Cell Signaling Technology, 9010 S, 1:1000; Proteintech, 22362-1-AP, 1:500), rabbit anti-BRCA2 (Cell Signaling Technology, 10741 S, 1:1000; ABclonal, A2435, 1:500), rabbit anti-RAD54 (Cell Signaling Technology, 15016 T, 1:1000), rabbit anti-p95/NBS1 (Cell Signaling Technology, 14956 T, 1:1000), rabbit anti-CtIP (Cell Signaling Technology, 9201S, 1:1000), rabbit anti-phospho-Chk2 (Thr68) (Cell Signaling Technology, 2197 T, 1:1000), rabbit anti-phospho-p53 (Ser15) (Cell Signaling Technology, 9286 S, 1:1000), rabbit anti-phospho-ATR (Ser428) (Abcam, ab178407, 1:1000), rabbit anti-ATR (Proteintech, 19787-1-AP, 1:1000), rabbit anti-DNAPKcs (Cell Signaling Technology, 38168 T, 1:1000), rabbit anti-phospho-DNAPKcs (Ser2056) (Cell Signaling Technology, 68716 T, 1:1000), rabbit anti-Ku70 (Cell Signaling Technology, 4588 T, 1:1000), rabbit anti-Ku80 (Cell Signaling Technology, 2180 T, 1:1000), rabbit anti-DNA Ligase IV (Cell Signaling Technology, 14649 T, 1:1000), rabbit anti-XLF (Cell Signaling Technology, 2854 T, 1:1000), rabbit anti-Artemis (Cell Signaling Technology, 13381T, 1:1000), rabbit anti-53BP1 (Abcam, ab175933, 1:1000), rabbit anti-phospho-53BP1 (Ser1778) (Cell Signaling Technology, 2675 S, 1:1000), rabbit anti-RPA32/RPA2 (Abcam, ab76420, 1:1000), rabbit anti-RPA70 (Abcam, ab79398, 1:1000), rabbit anti-RAD50 (Abcam, ab124682, 1:1000), rabbit anti-MRE11 (Abcam, ab208020, 1:1000), rabbit anti-phospho-RPA32 (Ser4/Ser8) (Bethyl Laboratories, A300-245A, 1:1000), rabbit anti-phospho-RPA32 (Ser33) (Bethyl Laboratories, A300-246A, 1:1000), rabbit anti-Caspase 3 (Cell Signaling Technology, 14220 T, 1:1000), rabbit anti-Cleaved Caspase 3 (Asp175) (Cell Signaling Technology, 9664 T, 1:1000), mouse anti-Caspase 9 (Cell Signaling Technology, 9508 T, 1:1000), rabbit anti-Cleaved Caspase 9 (Asp330) (Cell Signaling Technology, 52873 T, 1:1000), rabbit anti-PARP (Cell Signaling Technology, 9542 T, 1:1000), rabbit anti-Cleaved PARP (Asp214) (Cell Signaling Technology, 5625 T, 1:1000), rabbit anti-α-Tubulin (Proteintech, 66031-1-Ig, 1:2000), rabbit anti-GAPDH (Cell Signaling Technology, 8884 S, 1:2000), rabbit anti-β-actin (Cell Signaling Technology, 12620 S, 1:2000).

The following antibodies were used to perform immunofluorescence: mouse anti-γH2AX (Millipore, 05-636, 1:500), rabbit anti-RAD51 (Abcam, ab133534, 1:250), rabbit anti-phospho-53BP1 (Ser1778) (Cell Signaling Technology, 2675 S, 1:500).

The following antibodies were used to perform CUT & Tag and ChIP: anti-H3K4me2 (Abcam, ab32356, 4 µg for each Ig), anti-H3K9me2

(Abcam, ab1220, 4 μg for each Ig), anti-LSD1 (Millipore, 17-10531, 2 μg for each Ig).

The following antibodies were used to perform Immunohistochemistry: rabbit anti-Cleaved Caspase 3 (Asp175) (Cell Signaling Technology, 9664 S, 1:50), rabbit anti-LSD1 (Cell Signaling Technology, 2139 S, 1:100), rabbit anti-Ki67 (Abcam, ab1667, 1:100), rabbit anti-γH2AX (Cell Signaling Technology, 9718 S, 1:100); rabbit anti-H3K4me2 (Cell Signaling Technology, 9725 S, 1:100), rabbit anti-H3K9me2 (ABclonal, A2359, 1:50).

## Cell proliferation assay
For testing cell viability in response to different compound concentrations, CCK8 assay (Selleck) was used. Briefly, 1500–2000 cells were seeded in 96-well plates (Corning). On the following day, cells were treated with the indicated concentrations of drugs for 72 h. Subsequently, the cell viability was determined using CCK8 assay according to manufacturer's instructions. The absorbance readout was performed on a Thermo plate reader. Background values from empty wells were subtracted, and data were normalized to vehicle-treated control. $IC_{50}$ values were determined by nonlinear regression and a variable slope dose-response model using GraphPad Prism 9 software.

Synergistic effects between both compounds were calculated using the Chou-Talalay equation in CompuSyn software[68] (http://www.combosyn.com), which is based on the median-effect principle and the combination index-isobologram theorem. CompuSyn software generates combination index (CI) values, where CI < 0.75 indicates synergism, CI = 0.75–1.25 indicates additive effects, and CI > 1.25 indicates antagonism. Following the instruction of the software, drug combinations at constant ratios were used to calculate the CI values in our study.

## Colony formation assay
In all, 500–2000 cells were plated into 6-well plates and treated with the indicated compounds. Medium was changed every 3 days with appropriate drug doses for 12 days or until control wells became confluent. Colonies were washed twice with PBS, fixed with 100% methanol for 20 min and stained with 0.1% crystal violet. Integrated density was measured using Fiji software. For drug sensitivity, treated cells were normalized to untreated samples.

Synergistic effects between both compounds and CI values were calculated using the Chou-Talalay equation in CompuSyn software[68] (http://www.combosyn.com). CI < 0.75 indicates synergism, CI = 0.75–1.25 indicates additive effects, and CI > 1.25 indicates antagonism. Following the instruction of the software, drug combinations at non-constant ratios were used to calculate the CI values in our study.

## Cell cycle analysis
In all, $2 \times 10^5$ cells were plated in 6-well plates and treated with the indicated compounds for 24 h. The PI/RNase staining kit was used to assess the cell cycle distribution according to the manufacturer's instructions (BD Biosciences, 550825). Flow cytometry analysis was performed on a NovoCyte Advanteon cytometer (ACEA Biosciences, CA).

## Apoptosis analysis
In all, $2 \times 10^5$ cells were plated in 6-well plates and treated with the indicated compounds for 72 h, and then stained by the PE Annexin V Apoptosis Detection Kit I according to the manufacturer's instructions (BD Biosciences, 559763 or 556547). Flow cytometry analysis was performed on a NovoCyte Advanteon cytometer (ACEA Biosciences, CA) and NovoExpress 1.6.1 (Agilent Biosciences).

## Reverse-transcriptase quantitative PCR
Total RNAs were extracted with the FastPure Cell Total RNA Isolation Kit (Vazyme, RC112-01), and reverse transcribed to cDNA with the

PrimeScript Reverse Transcription reagent kit (Takara, RR047A) according to the manufacturer's protocol. RT-qPCR was performed using SYBR Green Master Mix (Bio-Rad, 1725124) with a Life Technologies QuantStudio 1. Data were analyzed by the $\triangle\triangle CT$ method using *GAPDH* as a housekeeping gene. The sequences of primers used are listed in Supplementary Table 2.

## Western blot analysis
Cells and tissues were harvested and lysed with Mammalian Cell & Tissue Extraction Kit (BioVision, K269) with protease inhibitors (Roche, 11873580001) and phosphatase inhibitor cocktail (Bimake, B15001). Protein concentration was measured using the Quick Start Bradford 1× Dye Reagent (Bio-Rad, 5000205). Cell lysates were separated on SDS-PAGE gels and transferred onto 0.45 or 0.2 μm PVDF membrane (Millipore, IPVH00010 or ISEQ00010). Blots were blocked in 5% milk in TBST (TBS/0.1% Tween-20) and stained with primary antibodies at 4 °C overnight. Blots were washed 3 × 10 min with TBST, incubated with secondary antibodies conjugated to a horseradish peroxidase (HRP) for 1 h at room temperature and washed again 3 times for 10 min with TBST. Immunoblots were developed using Western ECL Substrate (Millipore, WBKLS0500). Band intensity was quantified using Fiji software.

## HR and NHEJ reporter assays
For measuring HR efficiency, the HR reporter plasmid pDRGFP and endonuclease encoding pCBAScel (both gifts from Maria Jasin; Addgene plasmid # 26475 and # 26477, respectively)[46,69] were used. For measure NHEJ efficiency, the pimEJ5GFP plasmid (gift from Jeremy Stark, Addgene plasmid # 44026)[47] and pCBAScel plasmid were used. A2780 DR-GFP, ES2 DR-GFP, A2780 EJ5-GFP, and ES2 EJ5-GFP cell line were generated by our lab. Briefly, cells were transfected with pDR-GFP or pEJ5-GFP plasmids and selected with puromycin for 2 months. To examine the role of LSD1i in DSB repair, cells were treated with 0.5 μM LSD1i (ZY0511 or SP2577) for 24 h and then transfected with 3 μg plasmid expressing I-Scel endonuclease (pCBAScel) and incubation for 72 h with or without 0.5 μM LSD1i (ZY0511 or SP2577). To examine the role of individual genes in DSB repair, cells were transfected with LSD1, BRCA2, RAD51 or Ku80 siRNA together with pCBAScel plasmid using Lipofectamine 2000 transfection kit (Invitrogen, 11558019). After 72 h, cells were harvested and analyzed by flow cytometry (ACEA Biosciences, CA) and NovoExpress 1.6.1 (Agilent Biosciences). At least 10,000 cells were counted. HR and NHEJ efficiency of treated cells was compared with DMSO or non-targeting control siRNA, respectively. Knockdown of BRCA2 or RAD51 by siRNA act as a positive control for inhibiting HR repair, and knockdown of Ku80 by siRNA act as a positive control for inhibiting NHEJ repair.

## Neutral comet Assays
Neutral comet assays were performed with Comet Assay Kit (Trevigen, 4250-050-K) using manufacturer's instructions. Briefly, cells were treated with the indicated compounds for 48 h and harvested and rinsed twice with ice-cold PBS. After cell suspensions were embedded in LM (low melting) Agarose and deposited on comet slides. The slides were put in the 4 °C refrigerator for 30 min and then treated with neutral lysis buffer overnight. Next, the slides were subjected to electrophoresis at 21 V for 40 min. The slides were immersed in 70% ethanol for 5 min, dried at 37 °C for 15 min and then stained in SYBR Gold nucleic acid gel stain (Invitrogen, S11494) for 30 min in the dark. Analysis of neutral comet assay plus radiation was performed 24 h after 5 Gy ionizing radiation (IR) using RS-2000 X-ray biological irradiator (Rad Source Technologies, USA). Images were captured using a fluorescence microscope (Olympus CKX53). DNA damage quantified *via* the tail moment using the CometScore software. For each condition, at least 50 cells were analyzed.

## Immunofluorescence

Cells were grown on glass coverslips (VWR, 631-0150), fixed with 4% (w/v) paraformaldehyde (Beyotime, P0099) for 20 min at room temperature, washed with PBS, permeabilized with 0.1% (v/v) Triton X-100 for 10 min and blocked with 1% BSA in PBS for 30 min. Subsequently, cells were washed twice for 10 min with PBS and then incubated with the primary antibody diluted in PBS containing 1% BSA at 4 °C overnight. Cells were next washed twice with PBS and then incubated with the appropriated secondary antibody for 1 h at room temperature. Corresponding Alexa Fluor 488 goat anti-mouse (Invitrogen, A11029), Alexa Fluor 488 goat anti-rabbit (Invitrogen, A11034), and Alexa Fluor 594 goat anti-rabbit (Invitrogen, A11037) were used as secondary antibody. Cell nuclei were stained using DAPI (Beyotime, C1002). Confocal images were acquired using a Lecia TCS SP8 laser scanning microscope. γH2AX, p53BP1, and RAD51 foci pictures of each individual experiment were obtained with the same exposure parameters and quantified using Fiji software. Analysis of kinetics of γH2AX and RAD51 foci was performed after 2 Gy ionizing radiation (IR) using RS-2000 X-ray biological irradiator (Rad Source Technologies, USA). At least 100 cells from three to five fields of view and three independent experiments were counted. Cells with >20 foci per nucleus were considered γH2AX positive. For p53BP1 and RAD51 analysis, >15 foci per nucleus were considered foci-positive cells.

## In vivo tumor models

Three models were used in this study. In the subcutaneous tumor model, the tumorigenic line of A2780, SKOV3, and ES2 were injected subcutaneously ($5 \times 10^6$ cells) into female BALB/c nude mice. When tumors reached 50–150 mm³, mice were randomly assigned to treatment with vehicle or compounds as described. Tumor growth was monitored every 3 days by measurement of tumor diameters, and the tumor volume was calculated as follows: $0.5 \times \text{length} \times \text{width}^2$. At the end of treatment, all tumors were excised, weighed, and confirmed by histology.

In the intraperitoneal tumor model, A2780 and SKOV3 luciferase expressing cells ($5 \times 10^6$) were injected intraperitoneally into female BALB/c nude mice. For the ID8 OC mouse model, luciferase-expressing ID8 cells ($2 \times 10^6$) were injected intraperitoneally into C57BL/6 mice. After 10 days of inoculation, mice were treated with vehicle (control) or ZY0511 intraperitoneally once daily and monitored for survival. The mice were intraperitoneally injected with D-luciferin (150 mg/kg, 7903, Biovision) and subsequently anesthetized with isoflurane inhalation and photographed after 10 min injection using the PerkinElmer IVIS Lumina III. The Radiance (photons) within each area of interest was determined using the Living Image Software 4.5.2 (Perkin Elmer).

In the PDX model, the establishment of OC PDX models were performed as described[70]. Briefly, minced fresh tumor tissue (-1 mm³ per mouse) was transplanted subcutaneously into flanks of NCG mice. After palpable tumors formed, mice were randomly assigned to treatment with vehicle or drugs as described. Mice were treated until day 28 after administration and sacrificed for tissue harvest.

## Patient-derived organoids culture and viability assay

The human OC samples were obtained from two patients with their informed consent at the Chinese People's Liberation Army General Hospital (permit number: S2021-566-03). Through OncoCode Panel (673 genes) NGS testing and olaparib sensitivity testing, KO-96412 (with BRCA2 K3326* mutation, sensitive to olaparib) was used as an HR-deficient model, while KO-25127 (insensitive to olaparib) was used as an HR-proficient model. The organoids were maintained in 3D culture system and were seeded in low adherent 96-well plates, which were then treated with the indicated compounds for 5 days. Cell viability was determined by a CellTiter-Glo 3D Cell Viability Assay

(Promega, G9683). All the PDOs used in the study were provided by K2 Oncology Inc, Beijing.

## Immunohistochemistry

Tissues were fixed in 4% paraformaldehyde overnight and embedded in paraffin. 4 µm paraffin-embedded sections were subjected to staining with H&E and IHC following standard protocols. Briefly, antigen retrieval was performed by boiling the slides in citrate buffer (10 mM, pH 6.0) for 20 min and endogenous peroxidase was blocked by incubation with 3% hydrogen peroxide for 30 min. After slides were rinsed in PBS and blocked for 30 min with 5% bovine serum albumin (BSA). Slides were incubated overnight at 4 °C with primary antibodies. Slides were next washed three times with PBS and then incubated with the HRP-labeled secondary antibody for 1 h at room temperature. Subsequently, a two-step detection kit (ZSGB-BIO, PV-9001, and PV-9002) was used for IHC and hematoxylin for nuclear staining. After mounting, slides were photographed by Vectra Polaris (Perkin Elmer).

A tissue microarray (TMA) containing 45 pairs of human OC tissues and corresponding normal adjacent tissues (NATs) were purchased from the National Engineering Center for Biochips (Shanghai, China). All samples were histologically examined. Detailed clinicopathologic features were listed in Supplementary Table 1. The paraffin-embedded FTE, HOSE, and OC tissues of patients were collected with the approval of the Biomedical Ethics Review Committee, West China Hospital, Sichuan University (permit number: 2018SZ0241).

The IHC score for LSD1 staining was the average of tumor density score multiplied by the score of staining intensity. The tumor density was assigned a score using a semi-quantitative five-category grading system: 0, no tumor-cell staining; 1, 1–10% tumor-cell staining; 2, 11–25% tumor-cell staining; 3, 26–50% tumor-cell staining; 4, 51–75% tumor-cell staining; and 5, >75% tumor-cell staining. The staining intensity was assigned a score using a semi-quantitative four-category grading system: 0, no staining; 1, weak staining; 2, moderate staining; and 3, strong staining. Each slide was determined independently by two pathologists, in a blinded fashion.

## Chromatin immunoprecipitation assays

ChIP assays were performed according to the manufacturer's instructions for the Chromatin Immunoprecipitation Kit (Millipore, #17-10086). Briefly, cells were fixed with 1% formaldehyde, and cross-linked chromatin was sonicated to produce 200–1000 bp DNA fragments. The lysate incubated with protein A/G agarose and specific antibodies, namely, anti-LSD1 (Millipore #17-10531), anti-H3K9me2 (ab1220, Abcam), anti-H3K4me2 (ab32356, Abcam), or IgG (#17-10086, Millipore), at 4 °C overnight. Then, the protein/DNA complexes were eluted according to the instructions. Purified DNA was purified and analyzed using qPCR. All primers are listed in Supplementary Table 2.

## *KDM1A* level in human OC tissues and survival analysis by using datasets

The Oncomine database and Ualcan (http://ualcan.path.uab.edu/analysis-prot.html) were searched to compare the expression level of *KDM1A* in OC datasets. Kaplan-Meier survival analyses for disease outcomes were conducted using the online database (http://www.kmplot.com). The cBioPortal (https://www.cbioportal.org/) and muTarget (https://www.mutarget.com/) platforms were explored to compare the expression level of *KDM1A* in mutated BRCA1/2 and wild-type samples. The ROC Plotter (https://www.rocplot.org/ovarian/index) was used to analyze *KDM1A* transcriptome-level and response to platinum therapy.

## HRD score acquisition from HRD signature

HRD signature consisting of 230 differentially expressed genes was obtained as previously described[42]. Normalized gene expression data

after LSD1 inhibition were subjected to unsupervised clustering by Euclidean distance with these 230 genes. HRD scores were determined by calculating the Pearson's correlations between median centered gene expression levels for HRD signature and gene expression levels after LSD1 inhibition.

## RNA sequencing analysis

Total RNA was purified from A2780 and ES2 cells after dimethyl sulfoxide (DMSO) or ZY0511 (1 μM) treatment for 24 h and stored in TRIzol (Invitrogen, USA). Triplicate samples were harvested for each group. RNA-seq libraries were constructed using an Illumina stranded mRNA sample preparation kit (NEB, E7770) according to the manufacturer's protocol and were sequenced on an Illumina NovaSeq 6000 sequencing machine with 150-base pair (bp) paired-end reads. Genes with |log_2 fold change| ≥ 0.585 and adjusted $p$-value < 0.05 were counted as differentially expressed genes.

The DEGs for each comparison were uploaded for Ingenuity pathway analysis and all mapped genes were analyzed using Core Analysis to identify statistically significant canonical pathways *via* right-tailed Fisher's exact test. Ingenuity pathway databases also estimate regulatory direction for a subset of the canonical pathways and an activation z value is calculated to identify the direction and statistical significance of the regulation for each of the pathways.

## Cleavage Under Targets and Tagmentation sequencing analysis

The NovoNGS CUT&Tag 3.0 High-Sensitivity kit (Novoprotein Scientific, N259-YH01) was used to prepare the library of CUT&Tag-seq assay in each sample. And the Illumina NovaSeq 6000 was used to sequence the library of CUT&Tag-seq assay in each sample with the mode of paired-end 150 bp. The bowtie2 (v2.3.4.2, RRID:SCR_016368) was used to align the sequencing data with the hg19 as the reference with parameters of --very-sensitive -X 2000. The samtools (v0.1.18, RRID:SCR_002105) was used to convert the sam files to bam files. The MarkDuplicates implemented in GATK (v4.1.3.0, RRID:SCR_001876) was used to filter the duplicated alignment reads in bam files. The SEACR (v1.3, RRID:SCR_001876) was used to call the peak signals and regions in each sample by selecting the top 1% of regions by area under the curve (AUC). The bamCoverage (v3.5.0) was used to generate the bw files in each sample with BPM normalization. The deeptools (v3.5.0, RRID:SCR_016366) was used to visualize the global signal of peaks on TSS regions. The getCounts implemented in chromVAR (v1.8.0) was used to quantify the counts of each peak signal in each sample. The regions detected less than 10 counts would be filtered for subsequent analysis. The DESeq2 (v1.26.0, RRID:SCR_015687) was used to remove the effects of library size and region length in counts data. The statistic values and variation degree of peaks signal and regions were calculated by DESeq2 (v1.26.0). The significantly differential peaks were selected by the cutoff of $p$-value < 0.05 and |log_2 fold change| > 0.5. The ChIPseeker (v1.22.1, RRID:SCR_021322) was used to annotate the peaks signal and regions in each sample. The analysis workflow has been reported previously.

## Assay for transposase-accessible chromatin with sequencing analysis

The ATAC assay was performed using a Chromatin Profile Kit for Illumina (Novoprotein Scientific, N248). The library was sequenced by Illumina NovaSeq 6000 sequencing machine with 150-bp paired-end reads. The NGmerge (v0.3) was used to remove the adapters in raw sequencing data. The bowtie2 (v2.3.4.2, RRID:SCR_016368) was used to align the sequencing data with the hg19 as the reference with parameters of --very-sensitive -X 2000 and the duplicated reads were filter by the GATK pipeline (v4.1.3.0, RRID:SCR_001876). The samtools (v0.1.18, RRID:SCR_002105) was used to convert the sam files to bam files, and to remove the reads mapped on the mitochondrial genome. The bamCoverage (v3.5.0) was used to generate the bw files in each sample with BPM normalization. The HMMRATAC (v1.2.5) was used to detect the open region and peak signals and only the enriched scores are higher than 10 would be reminded. The deeptools (v3.5.0, RRID:SCR_016366) was used to visualize the global signal of peaks on TSS regions. The function implemented in soGGi (v1.18.0) was used to identify and generate the common open regions detected in all samples. The Rsubread (v2.0.1, RRID:SCR_016945) was used to quantify the counts of each peak signal in each sample. The DESeq2 (v1.26.0, RRID:SCR_015687) pipeline was used to identify the significantly differential regions with cutoff of $p$-value < 0.05 and |log_2 fold change| >0.5. The PCA-tools (v 2.5.15) was used to calculate and visualize the dispersions and distributions of each sample based on the principal component analysis. The ChIPseeker (v1.22.1, RRID:SCR_021322) was used to annotate the peaks signal and regions in each sample.

## Gene set enrichment analysis

GSEA was performed using GSEA software (http://www.gsea-msigdb.org/gsea/) with 1000 permutations. Gene sets used were obtained from MSigDB (Hallmark gene sets; reactome subset of canonical pathway from C2 databases). A custom HRD-associated genes were defined in ref. 42 and this gene set was split into up-regulated and down-regulated in HRD as used as input into GSEA. $P$ values < 0.05 and false discovery rate (FDR) < 0.25 were used to select statistically significant gene sets.

## Statistical analysis

Statistical analyses were performed using Prism 9 (GraphPad) for Mac OS. Comparisons between two groups were done with unpaired two-tailed Student's $t$ test. For comparisons among multiple groups, one-way ANOVA was used. For comparisons of curves over time, two-way ANOVA analysis followed by Dunnett test was used. The correlations were calculated by linear regression (Pearson's r). The survival curves were tested with log-rank test. $*p < 0.05$, $**p < 0.01$, and $***p < 0.001$ were considered significant and $p > 0.05$ was considered not significant (ns). Details of statistical analyses and biological replicates are described in each figure legends.

## Reporting summary

Further information on research design is available in the Nature Portfolio Reporting Summary linked to this article.

# Data availability

The RNA-seq, CUT&Tag-seq, and ATAC-seq data have been deposited in NCBI's Gene Expression Omnibus (GEO) and are available through the GEO series accession number GSE218798. TCGA and Cancer Cell Line Encyclopedia (CCLE) dataset of OC was downloaded from cBio-Portal (https://www.cbioportal.org/). hg19 was downloaded from UCSC (https://hgdownload.soe.ucsc.edu/goldenPath/hs1/vsHg19/). Source data are provided with this paper.

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

## Acknowledgements

This work was supported by grants from Project of the National Natural Science Foundation of China (grant 882272716 to Y.Z.), 1·3·5 Project for Disciplines of Excellence, West China Hospital, Sichuan University (grant ZYGD23011 to X.C.), and Natural Sciences Foundation of Sichuan (grant 2022NSFSC0053 to Y.Z.). Schematic diagrams in Fig. 8a, f, g, h were created with BioRender.com (https://biorender.com/).

## Author contributions

L.T., Yi.Z., and X.C. conceived the study and contributed to scientific hypothesis. L.T. and Yi.Z. designed and performed the majority of the experiments, analyzed the data, and wrote the manuscript with input from all authors. L.T., Yue.Z., X.P., and Y.Lu. analyzed bioinformatics data. X.Zhou, Z.Z., and Ya.Z. prepared the compound of ZY0511. Yue.Z., Y.Lu., and Z.C. contributed to neutral comet assays and in vivo tumor xenograft assays. Yue.Z., J.Q., and Y.Lu. contributed to HR and NHEJ reporter assays. L.X, Z.Y. and Yu.Z. contributed to collect tissue samples and IHC assay. Yue.Z. and Y.Liu. contributed to obtain LSD1 knockout clones using the CRISPR-Cas9 system. Y.Li., C.L., L.W., X.L. and N.Su. contributed to the experiments and data analysis. K.G., N.Sa., H.L., J.Z., Y.X., X.Zhon., J.X., and X.Y. contributed to in vivo tumor assays. K.X. contributed critical experimental materials of PDX model. X.T., Y.Liu., S.Y., Y.P., J.H. and X.C. revised the manuscript. All authors read and approved the manuscript.

## Competing interests

The authors declare no competing interests.
