## [Peer Review File · Nature Communications]

Repression of LSD1 potentiates homologous recombination-proficient ovarian cancer to PARP inhibitors through down-regulation of BRCA1/2 and RAD51REVIEWER COMMENTS

Reviewer #1 (Remarks to the Author):

Tao et al. explore the role of a potential combination therapy of inhibition of the histone demethylase LSD1 together with PARP for the treatment of ovarian cancer. The study involves LSD1 depletion utilizing genetic strategies as well as inhibition with LSD1 inhibitors particularly ZY0511, that was developed by their group recently. The study utilizes a variety of in vitro and in vivo xenograft model systems to support their hypothesis. The manuscript is fairly detailed in its approach with a wide variety of experimental techniques. While there have been several studies involving LSD1 manipulation/ inhibition in ovarian cancer cells, fewer studies have focused on a combination therapy strategy involving PARP co-inhibition. The concept of HR proficient tumors therapeutically made HR pathway deficient and thereby enhancing the PARP inhibition has been explored extensively in several cancer types, including ovarian cancers and therefore is not novel. It is also important to note that PARP inhibition in ovarian cancer is extensively explored and is utilized as a therapeutic modality in HR proficient as well as deficient ovarian cancers with varying levels of success depending on the ovarian cancer type.

The major finding of this manuscript rests on LSD1 inhibition/depletion mediated transcriptional downregulation of HR pathway factors like BRCA1, BRCA2, RAD51 and thereby making the ovarian cancer cells more susceptible to PARP inhibition treatment. However, the data as presented (discussed below) do not fully support the LSD1 inhibition in the context of transcriptomic regulation of HR associated genes and there are several major points as well as minor issues that are required to be addressed at a fundamental level to make the manuscript suitable for publication.

Major issues:

1. The authors have chosen a panel of different Ovarian cancer cell lines to support their hypothesis of increased LSD1 expression in cancer cell lines. Including additional HR deficient cell lines other than UWB1.289 will further strengthen their observations. One of the major inconsistencies that has been found through the entire manuscript is utilization of different cell lines for different sets of experiments, which significantly distracts the flow of the data and makes the interpretations more challenging. In majority of the experiments the non-cancerous ovarian cell lines have been missing as controls. As for example, the authors interrogated the effects of LSD1 knockdown on various cellular outcomes and functional assays in three different cell lines in Fig1 but the control cell lines were not utilized. In the same figure, in vivo tumor studies were done only with A2780. However, for many other in vivo tumor assays later (Fig 8), other different cell lines/or all cell lines have been used intermittently in different figures.
2. shRNA is used for a number of assays, however, based on the western blot analyses in Fig 1f, Fig 5j, extended Fig 4b, the shRNAs do not seem to be entirely effective in knocking down LSD1, have been inconsistent and show cell line to cell line variations. Some key experiments like the basic cell viability, functional assays, RNA seq and LSD1 Cut and Tag experiments should be done using Crispr strategies with 2 different guide RNAs.
3. Fig 3 investigates LSD1 dependent transcriptomic changes using the ZY501 inhibitor compound. How does LSD1 inhibition compare with LSD1 knockdown? Rnaseq analyses should be performed on LSD1 kd cells in multiple cell lines to compare the transcriptomic changes.
4. One of the central premises of the manuscript is the potential transcriptomic output regulation by LSD1. It is very hard to interpret the data as shown in Fig 3f and Fig 3g. Unless we have completely misinterpreted the qPCR normalization, there seems to be extremely modest or no transcriptomic changes in the DDR pathway responsible target genes at the transcript level, particularly those of BRCA1, BRCA2 and RAD51. The panel of HR associated effectors in GSEA analyses must be shown in heatmaps for better representation. There are also significant discrepancies between ZY0511 treated and shRNA treated samples with respect to fold changes in these DDR pathway genes, with some of the analyses missing statistics. These modest transcriptomic changes also do not corroborate with the dramatic protein level changes observed in the western blot analyses panel in Fig 3h. Collectively, these data, point to potentially translation block in these DDR specific transcript as opposed to direct transcriptional changes upon LSD1 inhibition.

5. The mechanism of action of ZY501 still remains unclear, and the authors have tested some of the other LSD1 inhibitors like the GSK LSD1 inhibitor. Different LSD1 inhibitors inhibit LSD1 function mechanistically in different ways (Enzymatic versus Non-Enzymatic inhibition, disruption of protein-protein interactions) and it is unclear why these different inhibitors were utilized to support enzymatic action of LSD1? The authors should compare their transcriptomic datasets along with their cut and tag results with an LSD1 inhibitor specifically targeting LSD1 enzymatic activity (TAK448 or TAK418 compounds).

6. One of the major concerns revolves around the analyses of the LSD1 cut and tag data. It is unclear in Fig 5a whether the LSD1 bound peaks in the Upregulated and Downregulated RNAseq targets are different between the Control and ZY0511. What proportion of differentially bound peaks overlap with differentially expressed genes upon ZY0511 treatment? It is absolutely necessary to generate a dataset with LSD1 kd or ko and overlap Kd RNA seq data set to legitimately identify the functional enrichment of LSD1 on the target gene loci. This should also be followed up with the global analyses of H3K9 and H3K4 methylation to effectively assess LSD1 enrichment. The authors picked only a few target genes, but it will be useful to also look globally at upregulated and downregulated genes and their associated upstream and downstream regions to interrogate methylation changes in H3K4 and H3K9.

7. There are also some conceptual issues around the epigenetic analyses, which should be discussed. Concerns also arise from the IGV tracks of H3K9me2. H3K9me2, unlike H3K4 methylation mark, is generally broadly distributed histone modification mark, however, as demonstrated in Fig 5f, these broad distribution characteristic H3K9me2 enrichment is missing from the IGV representation and almost appears identical to H3K4me2 mark. Other genomic regions, upregulated, downregulated target gene regions along with unchanged genomic coordinates should be shown. The H3K9me2 should be repeated with another antibody to compare the signal to noise ratio and the relative enrichments.

8. Another major concern arises from the enzymatic rescue and validating knockdown results with rescue experiments involving the enzymatic dead LSD1. Kim et al in their Molecular Cell paper (Pubmed ID: 32396821) has demonstrated that the K661A point mutation retained LSD1 enzymatic action to a great extent and introducing additional point mutation in A539E completely abrogates LSD1 enzymatic action. All the rescue experiments (ChIP, biological assays, western blots) should be repeated using the complete enzymatic dead LSD1 mutant for interpreting enzymatic versus non-enzymatic contributions of LSD1.

9. In Fig 6, LSD1 inhibition in combination with a panel of PARPi has been tested in a variety of HR proficient cell lines along with non-cancerous OC cell lines HOSEpiC and IOSE80. Fig 6b shows synergy curves. However, even though in Fig 6c, utilizing colony forming assays, great synergy is observed at lower doses (less than 1 μ m, around 40 μ m for ZY0511 and 400 nm for PARPi) of inhibitors the synergy curves show completely different sensitivities (1 μ m upwards) in the cell viability assays. Why is this disconnect in doses observed? Also in several cell lines, the synergism data are quite modest, and it seems like higher doses of ZY501 is sufficient for the dramatic reduction in cell viability irrespective of PARP inhibition. This is also observed in non-cancerous cell lines. Is this effect simply a result of high dosage associated drug toxicity?

10. In fig 7, e, f, I, j- the representative flow cytometry images are very difficult to see and read. This is also true for the Extended data Fig 2 f, g, h. Extended data Fig 2f with the different phases of cell cycle, the bar graphs do not have statistics, to compare different cell cycle phases across treatments.

11. Fig 8 collectively is the most impactful data of the manuscript with the impressive combination treatment regime in nude mice as well as PDX models in NCG mice. However, it is difficult to speculate what the data will look like in syngeneic tumor models with intact functional immune system. LSD1 is a major player in cancer immunotherapy and is also a master regulator of hematopoiesis, it will be functionally more relevant to translate these experiments in a syngeneic mouse model. It will be interesting to see how ZY501 performs in a more biologically relevant tumor microenvironment, engaging the immune modulators.

Minor issues:

1. The authors evaluate LSD1 expression levels in normal ovary versus cancer tissues from the various publicly available datasets and show survival analyses from different datasets. However not all

the datasets show a negative correlation between the LSD1 levels of expression and overall survival and progression free survival of OC patients. Is it due to the possibility that data are not grouped based on ovarian cancer subtypes? It will also be very informative if the TCGA, GSE26712 and GSE12470 data in extended figure 1a can be separated based on the ovarian cancer subtypes, the cell type of origin and most importantly BRCA1/2 and or HR pathway member mutation status.

2. OC tissue microarray IHC analyses are a very important addition to the study. Based on the available patient information and details, are there any direct correlations between the different OC subtypes, their BRCA1/2 mutation status and LSD1 overexpression? Based on fig 1C significant heterogeneity exists in ovarian cancer sample tissues and it will therefore be a useful addition. Is the Kaplan Meier survival plot in fig 1d. based on the OC tissue microarray data? Teasing out the subtype of cancers with the BRCA1/2 mutation status will significantly elevate the manuscript.

3. Just a minor observation, with detailed cellular origin analyses, OVCAR 5 has recently been classified to more of a G.I tract cancer cell line than a ovarian cancer cell line. Therefore, this cell line probably should be removed from the analyses.

4. Some of the experiments have been done with siRNA treatments like the DR-GFP reporter assays and EJ5-GFP reporter assays. It is unclear why this switch to siRNA was undertaken along with introduction of U2OS cell line in Fig 4O. It is unclear, what is the effective knockdown efficiency of the target genes. Also the baseline responses of HR in DR-GFP experiments are also majorly different in DMSO and siCtrl groups in the A2780 cell lines. Is siRNA transfection already causing stress to the cells making them more prone to DNA damage?

Reviewer #2 (Remarks to the Author):

The study provides new evidence supporting a role for the epigenetic regulator LSD1 HRD status and response of ovarian cancer cells to Parp inhibitors. The study carries disease value and expand the existing knowledge of the regulation of HRD status in cancer cells. The manuscript is well written and the data is for the most part well presented. However, there are few aspects of the study requiring additional experimentation to support the main conclusions of the manuscript and enhance the novelty and significance of the study. First, the mechanisms of coupling LSD1 to HRD targets should be defined. Is it transcription factor based? Are the signaling regulating LSD1 targeting known oncogenic cascades in ovarian cancer? This is critical to define the disease value of the manuscript. Second, the use of ZY0511 is novel but the LSD1 inhibitor is not, the authors should try expanding the translational potential and test the inhibitor in human HRD and non-HRD ovarian cancer PDOs or PDXs. Third, the data of invasion and proliferation are difficult to interpret as the migration effect could be explain by the effect on growth and cells death. And fourth, all the human correlations should include RAD51 and PALB2.

Reviewer #3 (Remarks to the Author):

PARP inhibitors have represented a paradigm shift for the treatment of ovarian cancer, however the impact of PARP inhibitors has largely been restricted to patients with homologous recombination (HR) deficient tumors. Also, the authors mention the clinical problem of PARP inhibitor resistance and there are no approved strategies for treating resistant disease. Tao L and colleagues present a report that demonstrates the utility of targeting an epigenetic modifier, LSD1, to induce a HR deficient state in a HR proficient tumor and increase PARP inhibitor sensitivity. The investigators demonstrate that LSD1 directly regulates the transcription of critical HR components - BRCA1, BRCA2, and RAD51. Consistently, targeting LSD1 leads to the induction of DNA double-strand breaks and increased PARP inhibitor sensitivity. The strengths of the study include multiple cell line models, orthogonal approaches to targeting LSD1, biomarkers paired with functional assays, a patient-derived xenograft model, innovation with the LSD1 targeting agent, next-generation chromatin profiling, and clearly presented data. The findings will be of interest to investigators in the fields of ovarian cancer, DNA

damage, and epigenetics. While the study is considered to be methodologically rigorous and novel, there are weaknesses that should be noted prior to publication.

- 1) Most high-grade serous ovarian carcinoma is appreciated to originate in the fallopian tube, thus the “normal” comparisons to ovarian surface epithelium while important are not optimal. The authors should address these concerns through the examination of already existing datasets or through additional comparisons.
- 2) The conclusion that LSD1 targeting had no apparent effect on NHEJ activity is not well supported based on the U2OS cell line functional assay or the 53BP1 data. Furthermore, LSD1 inhibition led to an apparent significant decrease in both Ku70/80 at the mRNA level.
- 3) The downregulation of BRCA1, BRCA2, and RAD51 at the protein level is not recapitulated well at the mRNA level, which is counter to the proposed mechanism of LSD1-mediated transcriptional regulation. The authors should address this apparent disconnect.
- 4) The justification to focus the study on LSD1 is not well established as there are numerous regulators of histone H3 K4 and K9 methylation.
- 5) The investigators should place their work in the context of other studies that target the epigenetic environment in the context of maintaining genomic stability and PARP inhibitor response, such as work highlighting HDAC, PRMT, and KMT.
- 6) The investigators should discuss the potential mechanism as the “normal” cells are not responsive to LSD1 targeting. Related, in tumor sections from LSD1 targeting in vivo models are the investigators able to appreciate differences in H3K4 and H3K9 methylation in non-tumor tissues?
- 7) PARPi response can partially be attributed to enhanced immune cell activation, LSD1 is expressed in immune cells, and LSD1 inhibitor accumulates in the spleen, investigators should highlight they only use immune compromised models, and this is a limitation.
- 8) LSD1 targeting is proposed to regulate H3K4 and H3K9 methylation, but outside of CUT&Tag there are no immunoblots indicating the LSD1-dependent changes in these modifications or how quickly the LSDi is remodeling the epigenome environment.
- 9) MINOR – the cell lines used throughout the study ES-2, A2780, and SKOV3 are controversial in the ovarian cancer field and while the study is relatively cell line agnostic, the wildtype p53 status of A2780 may explain possible differences observed between this cell line compared to the others.

Reviewer #4 (Remarks to the Author):

Reviewer’s comments on Tao et al @ Nature Comm

Tao et al investigated the effect of suppressing LSD1 on sensitizing ovarian cancer (OC) to PARP inhibitors. Using the patient samples, cell lines and various mouse and PDX models they have shown that the histone demethylase LSD1 is overexpressed in OC and upon shRNA depletion or pharmacological inhibition with a LSD1 inhibitor ZY0511, OC cells had reduced oncogenic properties in vitro and in vivo, via modulating the expression of genes involved in the homologous recombination DNA repair pathway. They have done a tremendous amount of experiments and presented the data in a well organized and logical way. Most of the experiments are well controlled and the results are convincing and meet publication quality. However, the reviewer found a few minor issues regarding the interpretation and explanation of data, and concern about the novelty of this work:

Major concern:

The authors have published papers on the discovery of the ZY0511 LSD1 inhibitor by Li et al, 2019 and applied the inhibitor to cervical cancer (Hela cells), colorectal cancer (HCT115 cells) by Li et al, 2021 as well as B-cell lymphoma by Liu et al, 2021. In this study they further applied the ZY0511 LSD1 inhibitor on ovarian cancer and showed that inhibiting LSD1 by ZY0511 can repress the transcription of HR pathway genes and subsequently confer sensitivity to PARP inhibition. Though the mechanism is different from that in B-cell lymphoma in which inhibiting LSD1 induced apoptosis and autophagy, the molecular mechanism of LSD1 in demethylating H3K9 and H3K4 has been well reported. The results presented here are of immediate interest to cancer biologists and clinicians and

are of great potential to be applied to OC patients, yet, the reviewer is not sure whether this work should be published in NC or other cancer specific journals.

Minor concerns:

1. LSD1 CUT&Tag results showed that the LSD1 bound to a significant number of genes that are repressed upon LSD1 inhibition, likely through H3K9me2 dependent and independent mechanisms. What about the genes that are upregulated upon inhibiting/depleting LSD1 (Figure 5a). The authors did not mention about this part of the differentially expressed genes in results / discussion. Would any DNA damage repair genes/pathways being affect? It would be interesting to analyze and show the results.

2. Why the spleen has a very high concentration of ZY0511 (Figure 2m)? It is known that the spleen is important for the degradation of aged RBC and the immune system. Figure 2n showed no defect on the overall physiology of the spleen and other organs (e.g. liver, which has trapped the 2nd highest amount of ZY0511) is normal, and the cell counts of hematopoietic lineage did not change, suggesting that the functions of these organs are not affected / to the extent to show a phenotype detected by H&E staining. The reviewer noticed that the authors have monitored the body weight of mouse with different treatments (Fig 8d, extended data Fig 6). However, whether the histone modifications including H3K4 and H3K9 methylations are altered and the gene expression profiles of the spleen and liver are changed remain unknown.

3. What account for the difference in ATAC-seq between ZY0511 inhibition and shLSD1 (Fig 5d)? The percent inhibition by ZY0511 and degree of knockdown by shRNA of this experiment is not known. Also, what is the molecular basis behind the ZY0511 inhibition and the binding of LSD1 to the chromatin (Fig 5c)?

4. typo: Line 387, but not catalytically inactive LSD1, restored cell HR....

Point by Point Response to Reviewers' Comments

We appreciate the insightful and constructive comments provided by the reviewers, which were helpful for improving the quality of our work. We would like to thank the Editorial team's clear instructions for revision. As requested, we have added a large amount of new data, re-analyzed some data and re-written some parts of the manuscript to address these comments. I hope that there is no doubt that we have taken the Reviewers' and Editors' comments very seriously. We believe that we have produced a more solid and cohesive manuscript by addressing the concerns raised by the reviewers. Below we provide a point-by-point response to the reviewers' concerns. Reviewer comments are shown in black, while our responses are in **blue**. Please note that the figure citations in our response below refer to the new (post-revision) figures. We have highlighted the changes within the manuscript in **red**. We hope the Reviewers and the Editors will find this manuscript to be much improved and suitable for publication.

Reviewer #1 (Remarks to the Author):

Tao et al. explore the role of a potential combination therapy of inhibition of the histone demethylase LSD1 together with PARP for the treatment of ovarian cancer. The study involves LSD1 depletion utilizing genetic strategies as well as inhibition with LSD1 inhibitors particularly ZY0511, that was developed by their group recently. The study utilizes a variety of in vitro and in vivo xenograft model systems to support their hypothesis. The manuscript is **fairly detailed** in its approach with a wide variety of experimental techniques. While there have been several studies involving LSD1 manipulation/ inhibition in ovarian cancer cells, fewer studies have focused on a combination therapy strategy involving PARP co-inhibition. The concept of HR proficient tumors therapeutically made HR pathway deficient and thereby enhancing the PARP inhibition has been explored extensively in several cancer types, including ovarian cancers and therefore is not novel. It is also important to note that PARP inhibition in ovarian cancer is extensively explored and is utilized as a therapeutic

modality in HR proficient as well as deficient ovarian cancers with varying levels of success depending on the ovarian cancer type.

The major finding of this manuscript rests on LSD1 inhibition/depletion mediated transcriptional downregulation of HR pathway factors like BRCA1, BRCA2, RAD51 and thereby making the ovarian cancer cells more susceptible to PARP inhibition treatment. However, the data as presented (discussed below) do not fully support the LSD1 inhibition in the context of transcriptomic regulation of HR associated genes and there are several major points as well as minor issues that are required to be addressed at a fundamental level to make the manuscript suitable for publication.

Response: We thank the reviewer for the appreciation of our study and the constructive comments. Following the suggestions, we have conducted additional experiments and addressed this reviewer's major concerns as described below. We believe that the quality of our study is significantly improved by addressing these concerns and we hope that our response sufficiently addresses the concerns raised by the reviewer.

Major issues:

1. The authors have chosen a panel of different Ovarian cancer cell lines to support their hypothesis of increased LSD1 expression in cancer cell lines. Including additional HR deficient cell lines other than UWB1.289 will further strengthen their observations. One of the major inconsistencies that has been found through the entire manuscript is utilization of different cell lines for different sets of experiments, which significantly distracts the flow of the data and makes the interpretations more challenging. In majority of the experiments the non-cancerous ovarian cell lines have been missing as controls. As for example, the authors interrogated the effects of LSD1 knockdown on various cellular outcomes and functional assays in three different cell lines in Fig1 but the control cell lines were not utilized. In the same figure, in vivo tumor studies were done only with A2780. However, for many other in vivo tumor assays later (Fig 8), other different cell lines/or all cell lines have been used intermittently in different figures.

Response: We appreciate reviewer's valuable comments. According to the reviewer's comments, we performed following experiments:

(1) We have added two other HR deficient (HRD) OC cell lines including COV362 (with BRCA1 mutation) and Kuramochi (with BRCA2 mutation) and performed RT-qPCR and western blot analysis to measure LSD1 levels. As shown in **Supplementary Fig. 1k, l**, LSD1 was highly expressed in HR proficient OC cell lines compared with normal ovarian epithelial cells, whereas LSD1 expression in HRD OC cell lines was almost equivalent with normal ovarian epithelial cells. These data were consistent with our data from patients OC tissues showing that LSD1 mRNA expression levels in BRCA1/2 wild-type groups were higher than that in BRCA1/2 mutation groups (**Supplementary Fig. 1e, f**).

We have supplemented these new data in our revised manuscript. Changes in corresponding *Materials and Methods* (Lines 684-685), *Results* (Lines 172-178) descriptions and *Figure legends* (Lines 1615-1622) are marked in red.

(2) We have performed cell viability, colony formation and cell apoptosis experiments after knockdown or pharmacological inhibition of LSD1 by using nonmalignant human ovarian surface epithelial cells IOSE80 as control. Notably, LSD1 knockdown and pharmacological inhibition had just slight effect on the proliferation of HOSE cells compared with the effects in OC cells (**Supplementary Fig. 2b-e**, **Supplementary Fig. 3f, g** and **Supplementary Fig. 3j-m**)

We have supplemented these new data in our revised manuscript. Changes in corresponding *Results* (Lines 195-196, 230-232) descriptions and *Figure legends* (Lines 1626-1635, 1656-1659, 1662-1667) are marked in red.

(3) In addition to A2780 xenograft models (Fig. 1j, k, p, q), we have added *in vivo* xenograft tumor studies using SKOV3 and ES2 cells. The results showed that LSD1 knockdown inhibited the tumor growth in SKOV3 and ES2 xenograft models, which were similar with results in A2780 xenograft models (Fig. 11-o and Fig. 1r, s).

We have supplemented these new data in our revised manuscript. Changes in corresponding *Results* (Lines 188-194) descriptions and *Figure legends* (Lines 1305-1314) are marked in red.

2. shRNA is used for a number of assays, however, based on the western blot analyses in Fig 1f, Fig 5j, extended Fig 4b, the shRNAs do not seem to be entirely effective in knocking down LSD1, have been inconsistent and show cell line to cell line variations. Some key experiments like the basic cell viability, functional assays, RNA seq and LSD1 Cut and Tag experiments should be done using Crispr strategies with 2 different guide RNAs.

Response: We thank the reviewer for raising this insightful point. According to reviewer's suggestion, we built up LSD1 knockout (KO) cells by CRISPR-Cas9 using two sgRNAs in two human OC cell lines (A2780 and ES2) and performed the key experiments. The results showed that LSD1 KO inhibited cell growth *in vitro* and *in vivo* (**Supplementary Fig. 2f-m**). Besides, LSD1 KO decreased BRCA1/2 and RAD51 expression in mRNA and protein level (**Supplementary Fig. 4e and Supplementary Fig. 5b**), suppressed homologous recombination repair (**Supplementary Fig. 6c-j**), and increased sensitivity of cells to PARPi (olaparib) which could be rescued by wild-type (WT) LSD1, but not catalytically inactive LSD1 (LSD1-K661A mutation or LSD1-DM (LSD1-A539E/K661A mutation)) (**Supplementary Fig. 8a-g**). Therefore, the results indicated that our findings could be repeated in LSD1 KO cell lines. We have supplemented these new data in our revised manuscript. Changes in corresponding *Materials and Methods* (Lines 706-711), *Results* (Lines 196-201, 300-302, 333-334, 425-428) descriptions and *Figure*

legends (Lines 1636-1644, 1684-1686, 1690-1694, 1711-1731, 1747-1774) are marked in red.

Besides, we conducted the mRNA-seq experiment by using LSD1 knockdown cell lines instead of using the LSD1 KO cell lines based on your **Major Comment 3**. The detail explanation and results were shown in next response.

In addition, we attempted to perform LSD1 CUT&Tag-seq experiments using LSD1 KO cell lines. However, we found that after LSD1 knockout, there was a very minimal amount of DNA bound by the LSD1 antibody, which was insufficient for library preparation and sequencing. Thus, we can't finish this experiment, and we hope that the reviewer will be understanding of the frustration we experienced during this process.

3. Fig 3 investigates LSD1 dependent transcriptomic changes using the ZY501 inhibitor compound. How does LSD1 inhibition compare with LSD1 knockdown? Rnaseq analyses should be performed on LSD1 kd cells in multiple cell lines to

compare the transcriptomic changes.

Response: We appreciate the reviewer's comments. According to reviewer's comment, we have performed RNA-seq using LSD1 knockdown ES2 cells and compared them with ZY0511-treated ES2 cells. As shown in **Supplementary Fig. 4a, b**, there was some overlap in differentially expressed genes (DEGs) between LSD1i treated and LSD1 knockdown cells (32.3% commonly upregulated; 42.2% commonly downregulated). Ingenuity Pathway Analysis indicated that there was a high coincidence of differentially expressed pathways affected by genetic knockdown and pharmacological inhibition of LSD1. For example, BRCA1 in DNA damage response pathway and the ATM signaling pathway was suppressed by both ZY0511 treatment and genetic knockdown of LSD1 (**Fig. 3a, b**). Notably, results of gene set enrichment analysis (GSEA) further confirmed that LSD1 inhibition downregulated the genes involved in DSB repair, particularly homology directed repair in both genetic depletion and pharmacological inhibition of LSD1 cells. Additionally, LSD1 knockdown induced significant perturbation of genes including a published PARPi sensitization gene signature (*Peng et al. Nat Commun. 2014, PMID: 24553445*), consistent with the observation in ZY0511-treated cells (**Fig. 3c**). Importantly, LSD1 inhibition significantly elevated HRD scores, suggesting LSD1 inhibition impaired HR and induced DNA repair gene defect as a sensitization mechanism to PARPi (**Fig. 3e**).

We have supplemented these new data in our revised manuscript. Changes in corresponding **Results** (Lines 255-279) descriptions and **Figure legends** (Lines 1362-1369, 1373-1376, 1675-1676) are marked in red.

Supplementary Fig.4

Fig. 3

Fig. 3

4. One of the central premises of the manuscript is the potential transcriptomic output regulation by LSD1. It is very hard to interpret the data as shown in Fig 3f and Fig 3g. Unless we have completely misinterpreted the qPCR normalization, there seems to be extremely modest or no transcriptomic changes in the DDR pathway responsible

target genes at the transcript level, particularly those of BRCA1, BRCA2 and RAD51. The panel of HR associated effectors in GSEA analyses must be shown in heatmaps for better representation. There are also significant discrepancies between ZY0511 treated and shRNA treated samples with respect to fold changes in these DDR pathway genes, with some of the analyses missing statistics. These modest transcriptomic changes also do not corroborate with the dramatic protein level changes observed in the western blot analyses panel in Fig 3h. Collectively, these data, point to potentially translation block in these DDR specific transcript as opposed to direct transcriptional changes upon LSD1 inhibition.

Response: We appreciate the reviewer's valuable comments and apologize for any confusion caused in our previous manuscript. To clarify, we have revised the figure and improved the statistical analysis in the revised manuscript to address these concerns and avoid any confusion.

In original Fig. 3f and Fig. 3g, we expressed the genes change by using relative mRNA level which was normalized to control (**previous Fig. 3f**) or shCtrl (**previous Fig. 3g**). To avoid the confusion caused by presentation such as the difference in maximum value of Y-axis between LSD1 inhibitor and shLSD1, we represented these data in heatmaps (**Supplementary Fig. 4d**) and we hope the new figure will be easy to read. We observed a notable decrease in the expression of the HR key genes BRCA1/2 and RAD51 in both LSD1 knockdown and pharmacological inhibition conditions (1 μ M, 36 h) in the two cell lines. The relative mRNA levels of BRCA1/2 and RAD51 compared with control ranged from 0.275 to 0.473 (**Supplementary Fig. 4d**). Additionally, we performed validation experiments in the LSD1 KO cell lines and similar results were obtained (**Supplementary Fig. 4e**).

Moreover, as suggested by the reviewer, the panel of DNA DSB repair pathway genes in GSEA analyses have been shown in heatmap. Specifically, the heatmap analysis demonstrated the downregulation of a subset of DNA DSB repair pathway genes in the LSD1 knockdown group (**Fig. 3d**).

Concerning about the consistency of transcriptomic changes and protein level changes in original Fig 3h, the difference of treatment time might be the reason. The

mRNA levels were detected at 24 h, whereas protein levels were detected at 48 h after LSD1i treatment. To further validate the extent of changes in BRCA1/2 and RAD51 mRNA and protein levels following LSD1i treatment, we performed RT-qPCR to analyze BRCA1/2 and RAD51 genes level after various concentrations (0.25-2 μ M) and periods (12-48 h) of ZY0511 treatment. The results showed that the mRNA expression of BRCA1/2 and RAD51 was decreased in a time- and concentration-dependent manner upon treatment with LSD1i (ZY0511) in all three cell lines (**Fig. 3f, g**). For example, the relative mRNA expression of RAD51 after LSD1i (1 μ M, 48 h) were 0.200 to 0.233 in three OC cell lines. Furthermore, we confirmed these findings at the protein level in the three cell lines by western blot analysis. The semi-quantitative analysis of protein levels using grayscale analysis demonstrated that the trends and extent of changes in protein levels were similar to the changes observed at the RNA levels (**Fig. 3h and Supplementary Fig. 5b**). For example, the relative protein level of RAD51 after LSD1i (1 μ M, 48 h) were 0.16-0.31 in three OC cell lines.

Therefore, we believed that LSD1 knockdown, knockout, and pharmacological inhibition impact gene transcription, leading to downregulation of DNA DSB repair genes at both the mRNA and protein levels. Among these genes, BRCA1/2 and RAD51 exhibit the most significant changes. We appreciate the valuable feedback provided by the reviewer, and we believe that these optimizations have addressed the concerns raised.

We have supplemented these new data in our revised manuscript. Changes in corresponding **Results** (Lines 273-274, 280-286, 300-302) descriptions and **Figure legends** (Lines 1370-1372, 1377-1384, 1680-1686, 1690-1694) are marked in red.

5. The mechanism of action of ZY5011 still remains unclear, and the authors have tested some of the other LSD1 inhibitors like the GSK LSD1 inhibitor. Different LSD1 inhibitors inhibit LSD1 function mechanistically in different ways (Enzymatic versus Non-Enzymatic inhibition, disruption of protein protein interactions) and it is unclear why these different inhibitors were utilized to support enzymatic action of LSD1? The authors should compare their transcriptomic datasets along with their cut and tag results with an LSD1 inhibitor specifically targeting LSD1 enzymatic activity (TAK448 or TAK418 compounds).

Response: We appreciate the reviewer's valuable comments. Our previous study of the computer-based molecular docking showed that ZY5011 bound to LSD1 (PDB ID: 5LHH) at Gly358 of AOL domain via hydrogen bond (*Li et al. Cancer Lett. 2019, PMID: 30978443*). Moreover, the stability and binding mechanism of ZY5011 in the binding site of LSD1 were further evaluated by molecular docking, ligand-based

pharmacophore, molecular dynamics (MD) simulations, molecular mechanics generalized born surface area (MM/GBSA) analysis, quantum mechanics/molecular mechanics (QM/ MM) calculation and Hirshfeld surface analysis (*Zhang et al. Biochim Biophys Acta Gen Subj. 2021, PMID: 34390793*). The study showed that ZY0511 may be allosteric inhibitor that showed high binding affinity towards H3 binding pocket of LSD1 with the neighboring amino acids (Gly358, Cys360, Leu362, Asp375, and Glu379). Although we attempted to experimentally validate the binding site of ZY0511 with LSD1 by co-crystallization, we can't success yet.

To demonstrate generalizability and since ZY0511 is not a clinical candidate, we additionally assessed SP2577, a compound related to ZY0511 that recently completed a phase I clinical trial for the treatment of advanced solid tumors (NCT03600649) and entered phase I/II trials for continued access (NCT05266196 and NCT03600649). The results showed that SP2577 also decreased BRCA1/2 and RAD51 levels, suppressed HR capacity and potentiated HR-proficient OC cells to PARPi, similar to the effects of ZY0511 (**Fig. 4a-o, Supplementary Fig. 5b and Supplementary Fig. 9a**). In our previous manuscript, we tested another LSD1i named GSK2879552 which also decreased BRCA1/2 and RAD51 levels and sensitized OC cells to PARPi (**previous Extended Data Fig. 4b and Extended Data Fig. 5g, h**). Although GSK2879552 was accommodated at the FAD binding pocket by forming covalent adducts while ZY0511 and SP2577 may particularly bind to the H3 pocket within LSD1, they ultimately block the demethylase activity of LSD1 (*Fang et al. J Hematol Oncol. 2019, PMID: 31801559*). Therefore, we choose these compounds to support the notion that LSD1 regulates BRCA1/2 and RAD51 transcription and mediates sensitivity of OC cells to PARPi dependently of its canonical demethylase function. However, the clinical studies of GSK2879552 were terminated because of unfavorable risk-benefit profiles. Thus, we have excluded the data related to GSK2879552 in our revised manuscript.

Besides, according to the reviewer's suggestion, we have performed RNA-seq and CUT&Tag-seq after TAK-418 treatment ES2 cells and compared them with ZY0511-treated ES2 cells. As shown in **Response Fig. 1a, b**, there was some overlap in DEGs between ZY0511-treated and TAK-418-treated cells (27.3% commonly

upregulated; 43.6% commonly downregulated). Ingenuity Pathway Analysis showed that TAK-418 treatment led to significant suppression of genes involved in DNA repair and cell-cycle checkpoint control, while significantly upregulated genes included those involved in p53 and senescence signaling, consistent with the observation in ZY0511 treatment (**Response Fig. 1c**). In addition, TAK-418 increased HRD score, suggesting TAK-418 decreased HR competence (**Response Fig. 1d**). TAK-418 treatment only slightly decreased enrichment of LSD1 at the promoter regions, in contrast to the stronger effect observed after treatment with ZY0511 (**Response Fig. 1e**). This difference in effect could potentially be due to the fact that TAK-418 and ZY0511 bind to different binding sites on LSD1, with ZY0511 exhibiting high binding affinity towards the H3 binding pocket of LSD1 and TAK-418 accommodating at the FAD binding pocket by forming covalent adducts.

6. One of the major concerns revolves around the analyses of the LSD1 cut and tag data. It is unclear in Fig 5a whether the LSD1 bound peaks in the Upregulated and Downregulated RNAseq targets are different between the Control and ZY0511. What proportion of differentially bound peaks overlap with differentially expressed genes upon ZY0511 treatment? It is absolutely necessary to generate a dataset with LSD1 kd or ko and overlap Kd RNA seq data set to legitimately identify the functional enrichment of LSD1 on the target gene loci. This should also be followed up with the global analyses of H3K9 and H3K4 methylation to effectively assess LSD1 enrichment. The authors picked only a few target genes, but it will be useful to also look globally at upregulated and downregulated genes and their associated upstream and downstream regions to interrogate methylation changes in H3K4 and H3K9.

Response: We thank reviewer for this insightful comment. According to the reviewer's suggestion, we have showed the proportion of differentially LSD1-bound peaks overlap with differentially expressed genes upon ZY0511 treatment. As showed in **Response Fig. 2a**, 36.2% upregulated genes and 41.6% downregulated genes were overlapped with reduced LSD1 binding after ZY0511 treatment.

We have also tried CUT&Tag-seq in LSD1 knockout cells as suggested by the reviewer. However, there was a very minimal amount of DNA bound by the LSD1 antibody after LSD1 knockout, which was insufficient sequencing. This can be explained by that the two sgRNAs were both designed to target the first exon of LSD1, resulting in the complete deletion of LSD1. The single knockout clones were validated via western blot analysis (**Supplementary Fig. 2f**).

7. There are also some conceptual issues around the epigenetic analyses, which should be discussed. Concerns also arise from the IGV tracks of H3K9me2. H3K9me2, unlike H3K4 methylation mark, is generally broadly distributed histone modification mark, however, as demonstrated in Fig 5f, these broad distribution characteristic H3K9me2 enrichment is missing from the IGV representation and almost appears identical to H3K4me2 mark. Other genomic regions, upregulated, downregulated target gene regions along with unchanged genomic coordinates should be shown. The H3K9me2 should be repeated with another antibody to compare the signal to noise ratio and the relative enrichments.

Response: We thank reviewer for this insightful comment. To address the concerns raised, we first analyzed the global distribution of H3K4me2-bound and H3K9me2-bound peaks. As showed in **Response Fig. 3a**, the majority (43.42%) of H3K4me2-bound peaks resided in the promoter regions, while the majority (34.53%) of H3K9me2-bound peaks were located in the intergenic regions with H3K9me2 peaks at promoter and intronic regions comprising 29.77% and 20.53%, respectively. In addition, we displayed expanded genomic regions (Chr3: 43992698-53992698, 10 Mb; Chr6: 29562754-44566270, 15 Mb; Chr11: 68677904-71677904, 3 Mb), and the broader IGV representation clearly demonstrated the widely distributed characteristics of H3K9me2 across the genome, which was distinct from the H3K4me2 mark (**Response Fig. 3b**).

Furthermore, we performed additional CUT&Tag-seq using another

commercially available H3K9me2 antibody (Active Motif, 39239). Unfortunately, in our experiments, the tested H3K9me2 antibody exhibited a relatively poor signal-to-noise ratio. This may be partly due to the suitability of this commercial antibody for use in CUT&Tag-seq, despite its validation in ChIP-qPCR (<https://www.ptgcn.com/products/Histone-H3K9me2-antibody-pAb-39239-AM.htm>). Moreover, since the introduction of the CUT&Tag technique by the Henikoff team (*Nat Commun.* 2019, PMID: 31036827; *Nat Protoc.* 2020, PMID: 32913232), there have been not publicly available H3K9me2 CUT&Tag-seq data. As a result, we also analyzed previously published H3K9me2 ChIP-seq data. We found four published papers reporting H3K9me2 peaks located within ± 5 kb of the TSS, although the proportion of H3K9me2 peaks binding to the promoter region in these four papers was not as high as our 29.77% (*Li et al. Nat Commun.* 2017, PMID: 28440295; *Hou et al. Sci Adv.* 2021, PMID: 33523893; *Cai et al. Nucleic Acids Res.* 2014, PMID: 24598257; *Smith et al. Genome Res.* 2009, PMID: 19546172). Although we used the same commercial H3K9me2 antibody (Abcam, ab1220) in our experiments, the observed differences may be partly attributed to the different methodologies employed in CUT&Tag-seq and ChIP-seq.

Furthermore, our CUT&Tag-seq data have been validated by a tremendous amount ChIP-qPCR of LSD1 along with H3K9me2 and H3K4me2 and rescue experiments (**Fig. 5d-i and Supplementary Fig. 8a-d**). More specifically, since LSD1 inhibition decreased BRCA1/2 and RAD51 expression in mRNA and protein levels and suppressed HR ability (**Fig. 3a-h and Fig. 4a-o**), we further examined the enrichment of LSD1 along with H3K9me2 and H3K4me2 at these three key genes (**Fig. 5d**). The results showed that LSD1 was highly enriched at the transcriptional start site of BRCA1/2 and RAD51 and LSD1 depletion resulted in enhanced H3K9me2 enrichment but had no obvious effect on H3K4me2 marks at these promoter regions. Moreover, we performed a tremendous amount ChIP-qPCR of LSD1 along with H3K9me2 and H3K4me2 and rescue experiments to validate this (**Fig. 5f-i and Supplementary Fig. 8a-d**).

We have supplemented these new data in our revised manuscript. Changes in

corresponding *Results* (Lines 389-419, 425-428) descriptions and *Figure legends* (Lines 1448-1469, 1747-1757) are marked in red.

8. Another major concern arises from the enzymatic rescue and validating knockdown results with rescue experiments involving the enzymatic dead LSD1. Kim et al in their Molecular Cell paper (Pubmed ID: 32396821) has demonstrated that the K661A point mutation retained LSD1 enzymatic action to a great extent and introducing additional point mutation in A539E completely abrogates LSD1 enzymatic action. All the rescue experiments (ChIP, biological assays, western blots) should be repeated using the complete enzymatic dead LSD1 mutant for interpreting enzymatic versus non enzymatic contributions of LSD1.

Response: We thank reviewer for this insightful comment. We performed new rescue

experiments as the reviewer suggested. Specifically, we reconstituted LSD1 knockdown and LSD1 knockout OC cells with wild-type LSD1 (LSD1-WT) or catalytically inactive LSD1 mutants (LSD1-K661A and a double mutant LSD1-A539E/K661A (LSD1-DM)) and repeated all the rescue experiments, including ChIP-qPCR, RT-qPCR, western blot analysis, assessment of HR ability, and evaluation of cell sensitivity to PARPi (**Fig. 5f-l and Supplementary Fig. 8a-g**). The results showed that genetic depletion of LSD1 increased H3K9me2 level at BRCA1/2 and RAD51 loci, decreased BRCA1/2 and RAD51 expression in mRNA and protein level, suppressed HR repair, and increased sensitivity of OC cells to PARPi (olaparib) which could be rescued by wild-type (WT) LSD1, but not catalytically inactive LSD1.

By performing these additional rescue experiments, we provide more comprehensive insights into the role of LSD1 and confirm that BRCA1/2 and RAD51 are direct targets of LSD1, and LSD1 mainly regulates these gene transcription dependently of its canonical demethylase function.

We have supplemented these new data in our revised manuscript. Changes in corresponding *Results* (Lines 404-428) descriptions and *Figure legends* (Lines 1457-1486, 1747-1774) are marked in red.

9. In Fig 6, LSD1 inhibition in combination with a panel of PARPi has been tested in a variety of HR proficient cell lines along with non-cancerous OC cell lines HOSEpic and IOSE80. Fig 6b shows synergy curves. However, even though in Fig 6c, utilizing colony forming assays, great synergy is observed at lower doses (less than 1 μM ,

around 40 μ m for ZY0511 and 400 nm for PARPi) of inhibitors the synergy curves show completely different sensitivities (1 μ m upwards) in the cell viability assays. Why is this disconnect in doses observed? Also in several cell lines, the synergism data are quite modest, and it seems like higher doses of ZY501 is sufficient for the dramatic reduction in cell viability irrespective of PARP inhibition. This is also observed in non-cancerous cell lines. Is this effect simply a result of high dosage associated drug toxicity?

Response: We thank the reviewer for this comment. The discrepancy in doses between Fig. 6b and Fig. 6c is due to the different methods used and the duration of compounds exposure. In Fig. 6b, cell viability was measured using the CCK8 assay after 72 h of compounds treatment. On the other hand, Fig. 6c employed colony formation assays to evaluate long-term drug effects over approximately 12 days. The concentrations of compounds used in Fig. 6c were much lower than those in Fig. 6b due to the extended duration of treatment. Since the experimental methods used are different, the combination index (CI) values are not suitable for direct comparison.

Although high concentration of ZY0511 exhibited strong inhibitory activity on cell viability *in vitro*, combination therapy is employed to enhance efficacy and decrease the potential toxicity, which are frequently-used strategy in anti-cancer drugs therapy (Morel *et al. Nat Rev Clin Oncol. 2020, PMID: 31570827*; Pilié *et al. Nat Rev Clin Oncol. 2019, PMID: 30356138*). Our results have demonstrated that the combinations of LSD1i (ZY0511) and PARPi (olaparib, niraparib and rucaparib) are a rational approach both *in vitro* and *in vivo* (**Fig. 6a-e, Fig. 8a-j and Supplementary Fig. 10a-d**). Optimizing outcomes would also rely on determining the appropriate dosing and scheduling of each agent to maximize benefits and minimize adverse events.

Moreover, we validated that ZY0511 mainly on-target prevented OC proliferation *in vitro* by cell viability assay, colony formation assay, cellular thermal shift assay (**Fig. 2a-c and Supplementary Fig. 3b, c**), and the computer-based molecular docking, etc. (Li *et al. Cancer Lett. 2019, PMID: 30978443*). Although ZY0511 inhibited non-cancerous cell proliferation, its IC₅₀ value (IC₅₀ = 1.34 - 3.35

μM) was much higher than that in OC cell lines ($\text{IC}_{50} = 0.36\text{-}0.39 \mu\text{M}$), supporting that ZY0511 inhibited OC cells proliferation mainly depends on on-target effect. However, we agreed with reviewer's comment that we cannot exclude the off-target or toxicity effect of ZY0511 as all anti-drugs exhibits effect which can't be explained by its target.

10. In fig 7, e,f, I, j- the representative flowcytometry images are very difficult to see and read. This is also true for the Extended data Fig 2 f,g, h. Extended data Fig 2f with the different phases of cell cycle, the bar graphs do not have statistics, to compare different cell cycle phases across treatments.

Response: We thank the reviewer for this valuable comment, which has helped us improve the visibility and statistical presentation in the figures. We have improved this by unifying the font sizes in the figures and have included the statistical results. Please see revised **Fig. 7e, g and Supplementary Fig. h, i, j, l**. We hope the current version has improved.

11. Fig 8 collectively is the most impactful data of the manuscript with the impressive combination treatment regime in nude mice as well as PDX models in NCG mice. However, it is difficult to speculate what the data will look like in syngeneic tumor models with intact functional immune system. LSD1 is a major player in cancer immunotherapy and is also a master regulator of hematopoiesis, it will be functionally more relevant to translate these experiments in a syngeneic mouse model. It will be interesting to see how ZY501 performs in a more biologically relevant tumor microenvironment, engaging the immune modulators.

Response: Thank you for your insightful comment. As suggested, we additionally set up a syngeneic mouse model in immune competent C57BL/6 mice using mouse OC cell line ID8 and investigated combination effects of LSD1i and PARPi. We consistently observed that LSD1i and PARPi combination markedly inhibited tumor growth to a much greater degree than either compound alone. Indeed, in the syngeneic mouse model, LSD1i (ZY0511) and PARPi (olaparib) showed minimal effect at 35.4%

and 36.1% tumor growth inhibition (TGI), respectively and the combination treatment resulted in 69.8% TGI (**Supplementary Fig. 10 c, d**). These results indicate that the combination therapy of LSD1i and PARPi not only inhibits tumor growth in immunocompromised mice but also retains its efficacy in immune competent mice. We also recognize the significance of LSD1 in cancer immunotherapy, and that PARPi response could partly be attributed to enhanced immune cell activation. Therefore, LSD1i might be used to enhance cancer immunotherapy in the future.

We have supplemented these new data in our revised manuscript. Changes in corresponding *Materials and Methods* (Lines 723-724, 913-916), *Results* (Lines 503-507) descriptions and *Figure legends* (Lines 1788-1795) are marked in red.

Minor issues:

1. The authors evaluate LSD1 expression levels in normal ovary versus cancer tissues from the various publicly available datasets and show survival analyses from different datasets. However not all the datasets show a negative correlation between the LSD1 levels of expression and overall survival and progression free survival of OC patients. Is it due to the possibility that data are not grouped based on ovarian cancer subtypes? It will also be very informative if the TCGA, GSE26712 and GSE12470 data in extended figure 1a can be separated based on the ovarian cancer subtypes, the cell type of origin and most importantly BRCA1/2 and or HR pathway member mutation status.

Response: We thank the reviewer for raising this insightful point. we have examined

the expression of LSD1 between human OC tissues and non-malignant normal tissues and Kaplan-Meier survival analysis from various datasets. The results showed that LSD1 expression was significantly enriched in the OC tissues compared with non-malignant tissues in five publicly available datasets, including TCGA, GSE26712, GSE12470, GSE10971 and CPTAC (**Supplementary Fig. 1a, b**). Moreover, the Kaplan-Meier survival analysis revealed that LSD1 expression was negatively correlated with overall survival from GSE27651, GSE63885, GSE3149 and GSE9891 datasets and progression-free survival of OC patients from GSE9891 and GSE63885 datasets (**Supplementary Fig. 1c, d**).

According to the reviewer's suggestion, we attempted to analyze the TCGA, GSE26712, and GSE12470 databases based on OC subtypes, cell origin, and BRCA1/2 and/or HR pathway member mutation status. However, we found that these datasets lacked information regarding the cell type of origin, and all three datasets consisted of samples of ovarian serous adenocarcinoma. Therefore, we were unable to conduct these two analyses including the cell type of origin and OC subtypes. We also cannot compare the expression of LSD1 in cell origins and OC subtypes in these datasets (GSE27651, GSE63885, GSE3149, GSE9891, and GSE63885) due to lacking related information or limited sample size. Notably, by analyzing the TCGA data through the cBioPortal platform, it was found that LSD1 mRNA expression levels in OC patients with BRCA1/2 wild-type were higher compared with those with BRCA1/2 mutations (**Supplementary Fig. 1e**).

Additionally, we attempted to further analyze the CSIOVDB database, a transcriptomic microarray database of 3431 human OC samples that includes clinical-pathological parameters and follow-up information of OC patients (<http://csiovdb.mc.ntu.edu.tw/CSIOVDB.html>). In the CSIOVDB database, we observed elevated levels of LSD1 in serous OC samples compared with mucinous OC samples (**Response Fig. 4a**). LSD1 was highly expressed in the Stem-A molecular subtype of OC, which is associated with a poor prognosis in OC patients (*Tan et al. EMBO Mol Med. 2013, PMID: 23666744*) (**Response Fig. 4b**). Furthermore, CSIOVDB analysis revealed that LSD1 expression was significantly up-regulated in

OC with higher differentiation degree and in cases of refractory or resistant disease
(Response Fig. 4c, d).

Moreover, we analyzed the IHC data from a human OC tissue microarray (TMA).
 There was no significant association between LSD1 expression and pathological
 classifications. Please see response for your *Minor issues #2* for detail explanation.

2. OC tissue microarray IHC analyses are a very important addition to the study. Based on the available patient information and details, are there any direct correlations between the different OC subtypes, their BRCA1/2 mutation status and LSD1 overexpression? Based on fig 1C significant heterogeneity exists in ovarian cancer sample tissues and it will therefore be a useful addition. Is the Kaplan Meier survival plot in fig 1d. based on the OC tissue microarray data? Teasing out the subtype of cancers with the BRCA1/2 mutation status will significantly elevate the manuscript.

Response: Thanks for the reviewer's suggestions. The Kaplan Meier survival plot in previous Fig 1d. (**now Fig. 1f**) was based on the OC tissue microarray data. According to reviewer's comment, we analyzed the WHO pathological classifications of OC samples used in this study, and found that most of them (80.0%, 36/45) were serous carcinoma, 17.8% (8/45) were endometrioid carcinoma and the rest of them (2.2%, 2/45) were clear cell carcinoma (the clinical characteristics of patients are provided in **Supplementary Table 1**). There was no significant association between LSD1 expression and pathological classifications. Besides, the information about BRCA1/2 mutation status were not included when collected the clinicopathological features.

We agreed with reviewer's valuable suggestion. The LSD1 protein level in different OC subtypes, cell origin, and HR pathway members mutation status and their correlation with OC patient's survival worth further study in the future.

3. Just a minor observation, with detailed cellular origin analyses, OVCAR5 has recently been classified to more of a G.I tract cancer cell line than a ovarian cancer cell line. Therefore, this cell line probably should be removed from the analyses.

Response: We appreciate the reviewer for this helpful suggestion. As suggested, we have excluded the OVCAR5 cell line from the analyses.

4. Some of the experiments have been done with siRNA treatments like the DR-GFP reporter assays and EJ5-GFP reporter assays. It is unclear why this switch to siRNA

was undertaken along with introduction of U2OS cell line in Fig 4O. It is unclear, what is the effective knockdown efficiency of the target genes. Also the baseline responses of HR in DR-GFP experiments are also majorly different in DMSO and siCtrl groups in the A2780 cell lines. Is siRNA transfection already causing stress to the cells making them more prone to DNA damage?

Response: We thank the reviewer for this comment.

(1) As U2OS cell line is a widely used tool cell line for DR-GFP and EJ5-GFP reporter studies (*Obara et al. Nat Commun. 2020, PMID: 32948765; Han et al. Cancer Cell. 2020, PMID: 33186520; Mohni et al. Cell. 2019, PMID: 30554877; Kaplan et al. Sci Transl Med. 2019, PMID: 31092693*), we choose U2OS cell lines in this assay. To avoid confusion, we established A2780 and ES2 EJ5-GFP cells instead of U2OS EJ5-GFP cells and evaluated the extent of the HR and NHEJ using DR-GFP and EJ5-GFP reporter assay based on A2780 and ES2 cells (**Fig. 4m-o**).

The utilization of siRNA was to ensure consistency among all cells before transfection with the pCBASceI plasmid. This approach aimed to prevent any potential impact on the relative capacity of HR and NHEJ due to variations in transfection efficiency. The efficiency of siRNA-mediated protein knockdown was validated using western blot analysis, and the results are showed in **Supplementary Fig. 6k**. The results showed that relative protein levels decreased to 0.09-0.39 compared with siCtrl.

(2) We appreciate the reviewer to pointed out the different baseline responses of HR in A2780 DR-GFP experiments. We have also noticed this issue and optimized the experiments concerning the effect of transfection reagent Lipo2000 in A2780 cells. Therefore, we reduced the dosage of Lipo2000 from 10 μ L per well to 5 μ L per well and performed the experiments again. The new results showed that there was no significant difference in baseline responses of HR between the DMSO and siCtrl groups in the A2780 cells (**Fig. 4m-o**).

We have supplemented these new data in our revised manuscript. Changes in corresponding *Materials and Methods* (Lines 858), *Results* (Lines 347-354) descriptions and *Figure legends* (Lines 1423-1435, 1732-1734) are marked in red.

Reviewer #2 (Remarks to the Author):

The study provides new evidence supporting a role for the epigenetic regulator LSD1 HRD status and response of ovarian cancer cells to Parp inhibitors. The study carries **disease value** and expand the existing knowledge of the regulation of HRD status in cancer cells. The manuscript is **well written** and the data is for the most part **well presented**. However, there are few aspects of the study requiring additional experimentation to support the main conclusions of the manuscript and enhance the novelty and significance of the study.

Response: We highly appreciate this reviewer for the constructive comments and suggestions, which have led to a great improvement of our study. Now we have made extensive revisions to address each point as below.

1. First, the mechanisms of coupling LSD1 to HRD targets should be defined. Is it transcription factor based? Are the signaling regulating LSD1 targeting known oncogenic cascades in ovarian cancer? This is critical to define the disease value of the manuscript.

Response: We appreciate the reviewer's comments. In this manuscript, we demonstrated that BRCA1/2 and RAD51 are direct targets of LSD1, and LSD1 mainly regulates these gene transcription dependently of its canonical demethylase function. To support this conclusion, we have done a tremendous amount ChIP-qPCR of LSD1 along with H3K9me2 and H3K4me2 and rescue experiments including ChIP-qPCR, RT-qPCR, western blot analysis, assessment of HR ability, and

evaluation of cell sensitivity to PARPi (**Fig. 5f-l and Supplementary Fig. 8a-g**). The results showed that genetic depletion of LSD1 increased H3K9me2 level at BRCA1/2 and RAD51 loci, decreased BRCA1/2 and RAD51 expression in mRNA and protein level, suppressed HR repair, and increased sensitivity of OC cells to PARPi (olaparib), which could be rescued by wild-type (WT) LSD1, but not catalytically inactive LSD1.

We believe that direct transcriptional repression of HR genes after LSD1 inhibition is the dominant mechanism. Transcriptional regulation is a common mechanism through which many epigenetic regulators, such as BET (*Yang et al. Sci Transl Med. 2017, PMID: 28747513*; *Sun et al. Cancer Cell. 2018, PMID: 29533782*), HDAC (*Konstantinopoulos et al. Gynecol Oncol. 2014, PMID: 24631446*), and DNMT (*Abbotts et al. Proc Natl Acad Sci U S A. 2019, PMID: 31591209*), modulate HR. However, we cannot exclude other indirect mechanisms that may also cooperate or contribute to this synergistic effect. We added these issues in the **Discussion** section in the revised manuscript (**Lines 593-616**). The sentences are as follows:

"Our study was limited to focusing on LSD1i-targeted genes which are directly involved in DNA damage repair. Although we believe that direct transcriptional repression of HR genes is the dominant mechanism, we cannot exclude other indirect mechanisms that may also cooperate or contribute to this synergistic effect. For example, knockout of LSD1 or LSD1i (HCI-2509) decreases the expression of c-MYC protein^{58, 59}, and the dual class I HDAC and LSD1 inhibitor known as Domatinostat reduces expression of FOXM1⁶⁰. Given that both c-MYC and FOXM1 regulate the genes that control DSB repair⁶¹, c-MYC or FOXM1 reduced by LSD1 inhibition might also indirectly contribute to reduction of HR gene expression and subsequently result in HR deficiency. Moreover, previous studies have suggested that LSD1 plays at least an indirect role in DDR by demethylating p53, thereby inhibiting p53-mediated transcriptional activation and apoptosis⁶². Besides, LSD1i may also directly influence the DNA damage response by disrupting chromatin signaling and impairing HR factor recruitment. Sulkowski et al. have reported that oncometabolites suppress HR via direct inhibition of the lysine demethylase KDM4B, leading to global elevation of H3K9me3 chromatin marks⁵⁷, which impedes Tip60 recruitment to DSBs, causing HR

failure, persistence of DSBs, and ultimately PARPi sensitization. As H3K9me2 is the substrate of H3K9me3, we found that LSD1 inhibition resulted in increase of not only H3K9me2 but also H3K9me3 protein levels (Supplementary Fig. 3d), in agreement with previous observation⁵⁴. Therefore, we speculate that Tip60 might also be impaired at DNA breaks, with diminished recruitment of downstream repair factors and impeded HR activity, because of H3K9me3 accumulation caused by LSD1 inhibition. Therefore, it is still necessary to further characterize how other mechanisms contribute to the synergistic effect of LSD1i and PARPi in some cellular contexts."

2. Second, the use of ZY0511 is novel but the LSD1 inhibitor is not, the authors should try expanding the translational potential and test the inhibitor in human HRD and non-HRD ovarian cancer PDOs or PDXs.

Response: We thank the reviewer for the suggestion improve clinical implication of the current study. According to reviewer's comment, we tested the combination effects in two PDOs derived from OC patients, HR-proficient (KO-25127) and HR-deficient (KO-96412), respectively. The results showed that LSD1i (ZY0511) markedly potentiated the killing effects of olaparib in HR-proficient PDOs, but not HR-deficient, ovarian cancer PDOs (**Supplementary Fig. 10a, b**), which further support our results that repression of LSD1 potentiates HR-proficient OC to PARP inhibitors.

We have supplemented these new data in our revised manuscript. Changes in corresponding *Materials and Methods* (Lines 929-939), *Results* (Lines 494-503) descriptions and *Figure legends* (Lines 1785-1787) are marked in red.

3. Third, the data of invasion and proliferation are difficult to interpret as the migration effect could be explain by the effect on growth and cells death.

Response: We appreciated the reviewer's comment. In our original manuscript, we showed that genetic depletion or pharmacological inhibition of LSD1 suppressed the proliferation, migration, and invasion of OC cells. In our study on cell migration, we utilized transwell and wound healing assay which are classic and widely used (*Nature* (PMID: 31511693), *Nature Cell Biology* (PMID: 29058718), *Nature Communications* (PMID: 37443154 and PMID: 37524695) and *Cancer Cell* (PMID: 30107178)). In above two assays, cells were first starved for at least 12 h prior to eliminate the impact on cell growth and cell death.

However, to dispel the confusion, we have ultimately decided to remove the migration and invasion data in our revised manuscript because they are dispensable for our study on the enhanced sensitivity of OC to PARP inhibitors in combination with LSD1 inhibition.

4. And fourth, all the human correlations should include RAD51 and PALB2.

Response: In previous manuscript, we compared LSD1 mRNA levels in BRCA1/2 mutant and wild-type OC patients using the cBioPortal online platform (**Response Fig. 5a, b**). Based on the reviewer's suggestion, we have included the mutations of RAD51 and PALB2 as well. However, due to the low frequency of RAD51 and PALB2 mutations (~1%), only four mutated patients (one with RAD51 mutation and

three with PALB2 mutation) were included for analysis in the TCGA-Nature 2011 dataset, and only five mutated patients (one with RAD51 mutation and four with PALB2 mutation) were included for analysis in the TCGA-Firehose Legacy dataset. This low mutation frequency is consistent with the reported studies (*Konstantinopoulos et al. Cancer Discov. 2015, PMID: 26463832*). Due to the extremely limited sample size of PALB2 and RAD51 mutations, we were unable to directly compare the expression of LSD1 between groups with and without these mutations. The small sample size hinders our ability to draw statistically meaningful conclusions in this regard.

Our original results showed that LSD1 was expressed at higher levels in OC patients with wild-type BRCA1/2 compared with those with BRCA1/2 mutations (**Response Fig. 5a, b**). Therefore, we have included RAD51 and PALB2 mutations in the original grouping of HRD caused by BRCA1/2 mutation. Similarly, the expression of LSD1 was lower in OC patients with mutations in all four genes (BRCA1/2, RAD51 and PALB2) compared with wild-type patients, and these analyses showed statistical significance (**Response Fig. 5c, d**). In conclusion, these results demonstrate that LSD1 expression is higher in HR-proficient OC patients compared to those with HRD, suggesting that targeting LSD1 may hold greater clinical significance in HR-proficient OC patients.

Reviewer #3 (Remarks to the Author):

PARP inhibitors have represented a paradigm shift for the treatment of ovarian cancer, however the impact of PARP inhibitors has largely been restricted to patients with homologous recombination (HR) deficient tumors. Also, the authors mention the clinical problem of PARP inhibitor resistance and there are no approved strategies for treating resistant disease. Tao L and colleagues present a report that demonstrates the utility of targeting an epigenetic modifier, LSD1, to induce a HR deficient state in a HR proficient tumor and increase PARP inhibitor sensitivity. The investigators demonstrate that LSD1 directly regulates the transcription of critical HR components - BRCA1, BRCA2, and RAD51. Consistently, targeting LSD1 leads to the induction of DNA double-strand breaks and increased PARP inhibitor sensitivity. **The strengths** of the study include multiple cell line models, orthogonal approaches to targeting LSD1, biomarkers paired with functional assays, a patient-derived xenograft model, innovation with the LSD1 targeting agent, next-generation chromatin profiling, and **clearly presented data**. The findings will be of interest to investigators in the fields of ovarian cancer, DNA damage, and epigenetics. While the study is considered to be **methodologically rigorous and novel**, there are weaknesses that should be noted

prior to publication.

Response: We thank the reviewer for the positive comments on the novelty and significance of our study and the thoughtful suggestions that helped improve the manuscript. We have addressed this reviewer's major concerns as described below. We believe that the quality of our study is significantly improved by addressing these concerns.

1) Most high-grade serous ovarian carcinoma is appreciated to originate in the fallopian tube, thus the "normal" comparisons to ovarian surface epithelium while important are not optimal. The authors should address these concerns through the examination of already existing datasets or through additional comparisons.

Response: We thank and agree with the reviewer for the comments. As requested, we have performed immunohistochemistry (IHC) analysis using paraffin-embedded tissues and western blot analysis using fresh tissues to examine the protein level of LSD1 in OC samples compared with non-malignant normal tissues, including normal human ovarian surface epithelium (HOSE) and fallopian tube epithelium (FTE). The results showed that LSD1 was upregulated in OC tissues compared with normal HOSE and FTE tissues by IHC (**Fig. 1d, e**) and western blot analysis (**Supplementary Fig. 1i, j**), consistent with our findings for the publicly available datasets (**Supplementary Fig. 1a**).

We have supplemented these new data in our revised manuscript. Changes in corresponding *Results* (Lines 141-147, 166-169) descriptions and *Figure legends* (Lines 1292-1295, 1594-1596, 1611-1614) are marked in red.

2) The conclusion that LSD1 targeting had no apparent effect on NHEJ activity is not well supported based on the U2OS cell line functional assay or the 53BP1 data. Furthermore, LSD1 inhibition led to an apparent significant decrease in both Ku70/80 at the mRNA level.

Response: We appreciate the reviewer's comments. We have now excluded the original sentence: "However, LSD1 inhibition had no apparent effect on NHEJ efficiency."

As suggestion by **Reviewer #1-Minor issues 4**, I have excluded U2OS cell line functional assay and included DR-GFP and EJ5-GFP reporter assay using A2780 and ES2. From this result, we have rephrased the results section and conclusion sentences as follows (**lines 348-355**): "*LSD1i (ZY0511 or SP2577) treatment and LSD1 knockdown by siRNAs targeting LSD1 resulted in substantial suppression of HR in A2780 and ES2 DR-GFP cell models, and this suppression was similar in extent to that seen with siRNAs targeting two key HR genes, BRCA2 and RAD51. However, LSD1 inhibition slightly affected the NHEJ capacity in A2780 and ES2 EJ5-GFP cell models, while knockdown of Ku80 significantly reduced the NHEJ capacity in these models (Fig. 4m-o and Supplementary Fig. 6k). Together, these results demonstrate that LSD1 inhibition suppresses HR and increases DNA DSBs. Together, these results demonstrate that LSD1 inhibition suppresses HR and increases DNA DSBs*". In addition, the reason for the differential changes observed in U2OS and OC cells (A2780 and ES2) after LSD1 inhibition in the EJ5-GFP reporter experiment may be

attributed to tissue or cell specificity.

We have supplemented these new data in our revised manuscript. Changes in corresponding *Materials and Methods* (Lines 858), *Results* (Lines 345-355) descriptions and *Figure legends* (Lines 1423-1435, 1732-1734) are marked in red.

3) The downregulation of BRCA1, BRCA2, and RAD51 at the protein level is not recapitulated well at the mRNA level, which is counter to the proposed mechanism of LSD1-mediated transcriptional regulation. The authors should address this apparent disconnect.

Response: We appreciate the reviewer's valuable comments and apologize for confusion caused in our previous manuscript. To clarify, we have revised the figure and improved the statistical analysis in the revised manuscript to address these concerns and avoid any confusion. In fact, the trends and extent of downregulation of BRCA1/2 and RAD51 at protein levels were consistent with the changes observed at the RNA level. Please also refer to our response to **Reviewer #1- Major issues 4**.

We have supplemented these new data in our revised manuscript. Changes in corresponding *Results* (Lines 273-274, 280-286, 300-302) descriptions and *Figure legends* (Lines 1370-1372, 1377-1384, 1680-1686, 1690-1694) are marked in red.

4) The justification to focus the study on LSD1 is not well established as there are numerous regulators of histone H3 K4 and K9 methylation.

Response: We thank the reviewer for making this point. The DNA damage repair (DDR) pathways are closely associated with chromatin remodeling mediated by

histone modifications, providing a rationale for combining PARPi with epigenetic agents. As pointed out by the reviewer, several H3K4-specific and H3K9-specific lysine methyltransferases (KMTs) and lysine demethylases (KDMs), including SUV39H1 (KMT1A), SETDB1 (KMT1E), KDM4B and KDM5A, have been found to be involved in the DDR (*Gong et al. Mutat Res Rev Mutat Res. 2019, PMID: 31395347; Sulkowski et al. Nature. 2020, PMID: 32494005*). However, the development of inhibitors targeting these epigenetic targets is still in preclinical research, and there have been no inhibitors entering clinical oncology studies yet (*Davalos et al. CA Cancer J Clin. 2023, PMID: 36512337*).

While it is true that multiple regulators are involved, we have chosen to focus on LSD1 due to its yet-to-be-fully-understood role in DDR and the fact that there are LSD1i entered clinical studies. In fact, emerging studies implicate LSD1 in the regulation of DDR in a cell/tissue-specific manner. LSD1 is shown to be recruited to the DNA Damage sites in RNF168 dependent manner and promotes the recruitment of 53BP1 and BRCA1 in U2OS cells (*Mosammamparast et al. J Cell Biol. 2013, PMID: 24217620*). LSD1 depletion sensitizes HeLa cells to γ -irradiation, HEK293 cells to DNA damage agents (bleomycin and etoposide) and SKOV3 cells to cisplatin (*Peng et al. Nucleic Acids Res. 2015, PMID: 25999347; Shao et al. Oncol Lett. 2018, PMID: 29928330*). LSD1 inhibition also attenuates DNA double-strand break repair pathways and enhances temozolomide-mediated DNA damage in glioma stem cells (*Alejo et al. Neuro Oncol. 2023, PMID: 36652263*). However, the precise molecular mechanism of LSD1 underlying DNA damage repair and whether LSD1 inhibition sensitizes HR-proficient OC to PARPi remain unclear. Therefore, in our study, we have explored this and provided strong evidence that LSD1 knockdown, knockout or pharmacological inhibition decreases HR, induces HRD and sensitizes HR-proficient OC cells to PARPi *in vitro* and in multiple *in vivo* models.

According to the above discussion, we have rewritten the introduction by incorporating your concerns and the suggestions provided in the next fifth comment (**Review #3-Major 5**) and added discussion about this point (**Lines 582-592**). In the revised introduction, we have integrated these feedback points and provided a

comprehensive overview of the research topic.

5) The investigators should place their work in the context of other studies that target the epigenetic environment in the context of maintaining genomic stability and PARP inhibitor response, such as work highlighting HDAC, PRMT, and KMT.

Response: We thank the reviewer for the valuable suggestion. According to the advice of the reviewer, we have amended the introduction text and added a paragraph to the introduction (**lines 74-90**) accordingly as below.

"To date, no direct inhibitors specifically targeting the proteins catalyzing HR are available. Intriguingly, to expand the use of PARP inhibitors to a larger group of HR-proficient OC patients, recent studies have focused on new combination strategies using agents that can induce HRD^{7, 9}. Targeting actionable proteins can interfere with gene expression, nuclear localization, and/or the recruitment of HR proteins, ultimately resulting in the indirect inhibition of HR and thereby engendering PARPi sensitivity^{7, 9}. For example, prior studies have demonstrated that targeting PI3K/AKT, RAS/MEK, and vascular endothelial growth factor receptor (VEGFR) pathways has the potential to pharmacologically induce an HRD phenotype¹⁰⁻¹⁴. Importantly, the DNA damage repair pathways are closely associated with chromatin remodeling mediated by histone modifications, providing a rationale for combining PARPi with epigenetic agents such as DNA methyltransferase (DNMT) inhibitors, histone deacetylase (HDAC) inhibitors, bromodomain and extra-terminal domain (BET) inhibitors, enhancer of the zeste homolog 2 (EZH2) inhibitors and protein arginine methyltransferase 5 (PRMT5) inhibitors¹⁵⁻²⁰. In this regard, selectively impairing HR in OC cells has been demonstrated to sensitize HR-proficient cancer cells to PARPi in preclinical and early clinical trials⁷⁻²⁰."

6) The investigators should discuss the potential mechanism as the “normal” cells are not responsive to LSD1 targeting. Related, in tumor sections from LSD1 targeting in vivo models are the investigators able to appreciate differences in H3K4 and H3K9 methylation in non-tumor tissues?

Response: We thank the reviewer for the valuable comments. This may, in part, be due to the higher expression of LSD1 in cancer tissues than normal tissues. We detected LSD1 protein level on tumor sections as well as major normal organs including the liver, spleen, and kidneys. The IHC results demonstrated that the expression of LSD1 in the tumors was significantly higher than in the liver, spleen, and kidney (**Supplementary Fig. 10f**). This is consistent with the higher expression of LSD1 in OC cell lines compared with normal ovarian epithelial cells, and the lack of significant inhibitory effect of LSD1 inhibition in normal ovarian epithelial IOSE80 cells. In addition, it was observed that cells expressing shLSD1 were less sensitive to ZY0511 treatment compared to cells expressing the non-targeting control shCtrl (**Supplementary Fig. 1i-l, Supplementary Fig. 2b-e, Supplementary Fig. 3f, g, j-m, and Fig. 2a-c**).

Additionally, as requested, we have now performed IHC staining on tumor sections as well as major normal organs including the liver, spleen, and kidney after LSD1 inhibitor (ZY0511) treatment. Our results showed that ZY0511 treatment did not significantly alter the levels of H3K4me2 and H3K9me2 in the liver, spleen, and kidney of mice. However, ZY0511 treatment led to an increase in the amount of H3K4me2 and H3K9me2 specifically in the tumor of xenografts (**Supplementary Fig. 10f**). This may, in part, be due to the higher expression of LSD1 in cancer tissues than normal tissues, including the liver, spleen, and kidney.

We have supplemented these new data in our revised manuscript. Changes in corresponding *Materials and Methods* (Lines 782-787), *Results* (Lines 522-530) descriptions and *Figure legends* (Lines 1800-1801) are marked in red.

7) PARPi response can partially be attributed to enhanced immune cell activation, LSD1 is expressed in immune cells, and LSD1 inhibitor accumulates in the spleen, investigators should highlight they only use immune compromised models, and this is a limitation.

Response: We thank the reviewer for raising this insightful point. To address the similar concerns raised by both you and Reviewer #1, we additionally investigated combination effects of LSD1i and PARPi in ID8 syngeneic OC mouse model using immune competent C57BL/6 mice. Indeed, in the syngeneic mouse model, LSD1i (ZY0511) and PARPi (olaparib) showed minimal effect at 35.4% and 36.1% tumor growth inhibition (TGI), respectively and the combination treatment resulted in 69.8% TGI (**Supplementary Fig. 10 c, d**). These results indicate that the combination therapy of LSD1i and PARPi not only inhibits tumor growth in immunocompromised mice but also retains its efficacy in immune competent mice. Please also refer to our response to **Reviewer #1- Major issues 3**.

We have supplemented these new data in our revised manuscript. Changes in corresponding *Materials and Methods* (Lines 723-724, 913-916), *Results* (Lines 503-507) descriptions and *Figure legends* (Lines 1788-1795) are marked in red.

8) LSD1 targeting is proposed to regulate H3K4 and H3K9 methylation, but outside of CUT&Tag there are no immunoblots indicating the LSD1-dependent changes in these modifications or how quickly the LSDi is remodeling the epigenome environment.

Response: As suggested, we detected global changes of H3K4 and H3K9 methylation in OC cells by western blot analysis and observed LSD1 inhibition by ZY0511 treatment resulted in increase of H3K4me1/2 and H3K9me1/2 in a concentration- and time-dependent manner (**Supplementary Fig. 3d, e**). These observations further support the role of LSD1 in regulating histone methylation marks and suggest that LSD1 inhibition by ZY0511 treatment can effectively modulate these epigenetic modifications.

We have supplemented these new data in our revised manuscript. Changes in corresponding **Results** (Lines 212-215) descriptions and **Figure legends** (Lines 1653-1655) are marked in red.

9) MINOR – the cell lines used throughout the study ES-2, A2780, and SKOV3 are controversial in the ovarian cancer field and while the study is relatively cell line agnostic, the wildtype p53 status of A2780 may explain possible differences observed between this cell line compared to the others.

Response: We appreciate the insightful comment from the reviewer. Although there has been some controversy regarding these three cell lines (*Domcke et al. Nat Commun. 2013, PMID: 23839242; Papp et al. Cell Reports. 2018, PMID: 30485824*), we choose them in our study because they exhibited high levels of LSD1 and did not harbor BRCA1/2 mutations (<https://cancer.sanger.ac.uk/cosmic>), considered HR-proficient cell lines in the context of this manuscript. To further validate the results from cells line, we generated PDX and PDO models to support our results.

As the reviewer pointed out, the wild-type p53 status of A2780 may indeed explain the observed differences compared to the other cell lines. For example, the wild-type p53 status of A2780 may be the reason for its relatively higher sensitivity to PARPi compared to the other two cell lines, consistent with previous reports showing that colorectal cancer cells carrying wild-type p53 are more sensitive to PARPi (*Chen et al. Proc Natl Acad Sci U S A. 2021, PMID: 34266953; Jelinic et al. Mol Cancer Ther. 2014, PMID: 24694947*). We acknowledge the importance of selecting appropriate cell lines for interpreting research results. We will consider using other cell lines that have broader representativeness to validate and support our findings in the future. Again, we greatly appreciate the valuable feedback provided by the reviewer.

Reviewer #4 (Remarks to the Author):

Reviewer's comments on Tao et al @ Nature Comm.

Tao et al investigated the effect of suppressing LSD1 on sensitizing ovarian cancer (OC) to PARP inhibitors. Using the patient samples, cell lines and various mouse and PDX models they have shown that the histone demethylase LSD1 is overexpressed in OC and upon shRNA depletion or pharmacological inhibition with a LSD1 inhibitor

ZY0511, OC cells had reduced oncogenic properties *in vitro* and *in vivo*, via modulating the expression of genes involved in the homologous recombination DNA repair pathway. They have done a **tremendous amount** of experiments and presented the data in a **well organized and logical** way. Most of the experiments are well controlled and the results are **convincing and meet publication quality**. However, the reviewer found a few minor issues regarding the interpretation and explanation of data, and concern about the novelty of this work:

Response: We highly thank for this reviewer's positive comments and constructive suggestions.

Major concern:

The authors have published papers on the discovery of the ZY0511 LSD1 inhibitor by Li et al, 2019 and applied the inhibitor to cervical cancer (Hela cells), colorectal cancer (HCT115 cells) by Li et al, 2021 as well as B-cell lymphoma by Liu et al, 2021. In this study they further applied the ZY0511 LSD1 inhibitor on ovarian cancer and showed that inhibiting LSD1 by ZY0511 can repress the transcription of HR pathway genes and subsequently confer sensitivity to PARP inhibition. Though the mechanism is different from that in B-cell lymphoma in which inhibiting LSD1 induced apoptosis and autophagy, the molecular mechanism of LSD1 in demethylating H3K9 and H3K4 has been well reported. The results presented here are of immediate interest to cancer biologists and clinicians and are of great potential to be applied to OC patients, yet, the reviewer is not sure whether this work should be published in NC or other cancer specific journals.

Response: We thank this reviewer's positive comments regarding the significance of our study. In this paper, we demonstrate that LSD1 is an important regulator of OC. Importantly, genetic depletion or pharmacological inhibition of LSD1 induces HRD, through depletion of BRCA1/2 and RAD51, and sensitizes HR-proficient OC cells to PARPi *in vitro* and in multiple *in vivo* models, including patient-derived xenograft and patient-derived organoids models. *In vivo* data strongly support low toxicity of the combination of LSD1i and PARPi. Thus, our findings provide a strong rationale for

clinical application of PARPi in combination with LSD1i for OC patients with HR-proficient. Due to these notable observations of targeted and combination therapy, we believe that our research findings are of immediate interest to cancer biologists and clinicians and may be of particular interest to the readers of *Nature Communications*.

Minor concerns:

1. LSD1 CUT&Tag results showed that the LSD1 bound to a significant number of genes that are repressed upon LSD1 inhibition, likely through H3K9me2 dependent and independent mechanisms. What about the genes that are upregulated upon inhibiting/depleting LSD1 (Figure 5a). The authors did not mention about this part of the differentially expressed genes in results / discussion. Would any DNA damage repair genes/pathways being affect? It would be interesting to analyze and show the results.

Response: As suggested by the reviewer, we further analyzed the overlap genes both differentially expressed after LSD1 inhibition and bound by LSD1 (**Supplementary Fig. 7a, b**). Ingenuity Pathway Analysis of the total 1113 overlap genes that were both downregulated by LSD1 inhibition and directly bound by LSD1 revealed a significant enrichment in the DNA damage repair pathway, including the ATM signaling pathway and role of BRCA1 in DNA damage response pathway, while the upregulated genes were enriched in the pathways leading to cell death, such as apoptosis, autophagy, and ferroptosis signaling pathway (**Supplementary Fig. 7c, d**), consistent with previous reported studies (*Zhao et al. Pharmacol Res. 2021, PMID: 34246782; Chao et al. Oncotarget. 2017, PMID: 29088798; Lu et al. Sci Rep. 2022, Sci Rep. 2022*).

We have supplemented these new data in our revised manuscript. Changes in corresponding **Results** (Lines 368-375) descriptions and **Figure legends** (Lines 1736-1743) are marked in red.

2. Why the spleen has a very high concentration of ZY0511 (Figure 2m)? It is known that the spleen is important for the degradation of aged RBC and the immune system. Figure 2n showed no defect on the overall physiology of the spleen and other organs (e.g. liver, which has trapped the 2nd highest amount of ZY0511) is normal, and the cell counts of hematopoietic lineage did not change, suggesting that the functions of these organs are not affected / to the extent to show a phenotype detected by H&E staining. The reviewer noticed that the authors have monitored the body weight of mouse with different treatments (Fig 8d, extended data Fig 6). However, whether the histone modifications including H3K4 and H3K9 methylations are altered and the gene expression profiles of the spleen and liver are changed remain unknown.

Response: We thank the reviewer for raising this insightful point. Significantly, as requested, we have included IHC staining on tumor sections as well as major normal organs including the liver, spleen, and kidneys after LSD1 inhibitor (ZY0511) treatment. Our results showed that ZY0511 treatment did not significantly alter the levels of H3K4me2 and H3K9me2 in the liver, spleen, and kidney of mice. However, ZY0511 treatment led to an increase in the amount of H3K4me2 and H3K9me2 specifically in the tumor of xenografts (**Supplementary Fig. 10f**). Additionally, we assessed the impact of LSD1i (ZY0511) treatment on the expression of BRCA1/2 and RAD51 in normal tissues isolated from ZY0511-treated mice. However, no significant effect of ZY0511 treatment was observed in certain normal tissues, including the liver and spleen (**Supplementary Fig. 10e**). These data suggest that ZY0511 specifically

down-regulates BRCA1/2 and RAD51 in tumor cells but with no corresponding effect on these gene expression in nonmalignant and healthy tissue. This studies future support the tolerability of ZY0511 in combination with PARPi, warranting exploration in human clinical trials.

The high concentration of ZY0511 in the spleen (previously Figure 2m, now Fig.2k) could be due to several factors. One possibility is that the spleen has a higher affinity for ZY0511 compared with other organs, resulting in its accumulation in this particular tissue. In fact, we found that LSD1 was overexpressed in the spleen compared with some other normal organs from the platform of the human protein atlas (<https://www.proteinatlas.org/ENSG00000004487-KDM1A/tissue>). Moreover, IHC staining using LSD1 on the sections demonstrated that LSD1 was expressed in normal mouse spleen tissue sections, with levels higher than in normal liver tissue but noticeably lower than in tumor tissue sections (**Supplementary Fig. 10f**). However, no changes of hematological, blood biochemistry parameters and mean body weight of mice and no lesions of the spleen were detected after ZY0511 administration. These studies supported the tolerability of ZY0511 treatment *in vivo*. It is also important to consider the pharmacokinetics and distribution properties of ZY0511, which can influence its accumulation in specific tissues. We thank the reviewer for their expertise and for raising such interesting points of discussion.

We have supplemented these new data in our revised manuscript. Changes in corresponding *Materials and Methods* (Lines 740-741, 747-750, 782-787), *Results* (Lines 516-530) descriptions and *Figure legends* (Lines 1796-1801) are marked in red.

3. What account for the difference in ATAC-seq between ZY0511 inhibition and shLSD1 (Fig 5d)? The percent inhibition by ZY0511 and degree of knockdown by shRNA of this experiment is not known. Also, what is the molecular basis behind the ZY0511 inhibition and the binding of LSD1 to the chromatin (Fig 5c)?

Response: We thank the reviewer for the valuable comments.

(1) In our original manuscript, the changes observed in ATAC-seq after ZY0511 inhibition are greater than those caused by shLSD1. This could be because ZY0511 induces a greater degree of LSD1 inactivation compared with the extent of LSD1 loss caused by shLSD1. Our protein validation results indicate that the extent of LSD1 knockdown is approximately 70% in ES2 cells used in ATAC-seq (**Supplementary Fig. 2a**). Our previous study demonstrated that ZY0511 exhibited strong efficiency against LSD1 with an IC_{50} value of 1.7 nM and showed good binding affinity with calculated dissociation constants of 2.42 nM (*Sang et al. MedComm. 2023, PMID: 37250145*). In addition, the duration of action is different between the two groups. The shLSD1 cell line may have undergone long-term puromycin selection for constructing stable LSD1 knockdown cell lines, which could result in certain genetic compensation effects, whereas ZY0511 treatment was only administered for 24 h, and

this effect is transient. Importantly, we additionally performed ATAC-seq in LSD1 knockout (KO) cell and compared with the results with LSD1 inhibition by ZY0511. There was no difference between LSD1 pharmacological inhibition and LSD1 knockout (**Response Fig. 6**).

(2) The full length of LSD1 is comprised of 852 amino acids with three key structure domains: N-terminal Swi3-Rsc8-Moira domain (SWIRM domain, residues 172-270); C-terminal amine oxidase like domain (AOL domain, residues 271-417 and 523-833); and central tower-like domain (Tower domain, residues 418-522). Our previous study of the computer-based molecular docking showed that ZY0511 bound to LSD1 (PDB ID: 5LHH) at Gly358 of AOL domain via hydrogen bond (*Li et al. Cancer Lett. 2019, PMID: 30978443*). Moreover, the stability and binding mechanism of ZY0511 in the binding site of LSD1 were further evaluated by molecular docking, ligand-based pharmacophore, molecular dynamics (MD) simulations, molecular mechanics generalized born surface area (MM/GBSA) analysis, quantum mechanics/molecular mechanics (QM/MM) calculation and Hirshfeld surface analysis (*Zhang et al. Biochim Biophys Acta Gen Subj. 2021, PMID: 34390793*). The study showed that ZY0511 may be allosteric inhibitor that showed high binding affinity towards H3 binding pocket of LSD1 with the neighboring amino acids (Gly358, Cys360, Leu362,

Asp375, and Glu379). Although we attempted to experimentally validate the binding site of ZY0511 with LSD1 by co-crystallization, we can't success yet.

Besides, the molecular basis underlying the binding of LSD1 to chromatin can be summarized as follows: The AOL domain of LSD1 serves as the catalytic center for regulating enzymatic activity and targeting substrate proteins. It is divided by the Tower domain into two lobes: (1) One forms a noncovalent FAD-binding site, consisting of three fragments (residues 271–356, 559–657, and 770–833), similar to other amine oxidases; (2) The other forms a more open funnel-shaped active site for substrate binding and recognition, comprising three fragments (residues 357–417, 523–558, and 658–769), allowing LSD1 to pocket more surrounding residues near the target lysine. The crystal structure of LSD1 in complex with nucleosome (PDB code: 6VYP) revealed that the histone H3 tail protrudes from the nucleosome particle to insert into the large substrate cavity of AOL domain, thus properly positioning the methylated lysine residues in substrates (*Kim et al. Mol Cell. 2020, PMID: 32396821*). The lysine triad of residues Lys355, Lys357, and Lys359 in LSD1 gates the entrance of the H3 binding pocket. The N-terminal 1-3 residues of H3 are located at a negatively charged pocket composed of residues Ala539, Asn540, Trp552, Asp553, Asp555, and Asp556 and form hydrogen bond and electrostatic interactions with Asp553 and Asp556. The Gln5 and Thr6 of H3 form hydrogen bond interactions with Asn535 and His564 in LSD1, respectively. Furthermore, Arg8 of H3 also forms hydrogen bond interactions with Cys360, Asp375, and Glu379 in LSD1. The partially disordered side chains of Lys9 and Lys14 of H3 form favorable electrostatic interactions with several negatively charged residues on the LSD1 surface (*Song et al. J Med Chem. 2023, PMID: 36537915*).

4. typo: Line 387, but not catalytically inactive LSD1, restored cell HR....

Response: We thank the reviewer for spotting the typo that is now rephrased.

REVIEWERS' COMMENTS

Reviewer #1 (Remarks to the Author):

The authors have diligently utilized the time, to reasonably address most of my queries. The manuscript has improved substantially. I have some additional questions which can be addressed in the manuscript.

1. The authors have reported difficulties, in Cut and Run experiments with the cut and run experiments with the Crispr knockout cell line for LSD1 due to insufficient IP material for sequencing. This seems to be a little difficult to understand as even regular non specific pulldowns happen with species specific IgG antibodies, which are usable for preparing sequencing libraries. This is a critical experiment that needs to be included if possible.

2. The LSD1 mutants (K661 and DM) shows identical rescue phenotypes across different cell lines. Do they rescue transcriptionally as well? There should be a few lines in discussion as to how the enzymatically replete LSD1 (K661) is able to perform rescue as comparable to the complete enzymatic dead LSD1 (DM).

3. The authors should include some discussion as to how LSD1 potentially regulates H3K9Me2 (Direct versus indirect regulation through H3K9 modulators).

Reviewer #2 (Remarks to the Author):

The authors has been responsive to previous critiques. This reviewer does not have any further comments. Congratulations on the great work!

Reviewer #3 (Remarks to the Author):

The authors have addressed all of the concerns from the initial review. The authors have added clarification and numerous new data panels. Overall, the revised manuscript and data presented are improved, rigorous, and of high quality.

Notably, since reviewing the initial manuscript there have been updated guidelines in terms of PARP inhibitor use in the context of HR-status. In the context of a patient being treated with a PARP inhibitor, the results of the study would likely only apply to those being treated with Niraparib. Notably, Niraparib is considerably more toxic compared to other PARPi and thus the doses used throughout the manuscript are considered non-physiological. Note, that toxicity studies, while comprehensive, were conducted with olaparib. While these updates do somewhat diminish the impact of the findings, there would still be an interest in the reported findings. The authors may want to consider addressing these updated guidelines in the discussion.

Reviewer #4 (Remarks to the Author):

The authors have addressed all my questions and concerns in the rebuttal letter. I have no further comment on the revised manuscript. Thank you.

Point by Point Response to Reviewers' Comments

We appreciate the insightful and constructive comments provided by the reviewers and thank the Editorial team's clear instructions for revision. Below we provide a point-by-point response to the reviewers' concerns. Reviewer comments are shown in black, while our responses are in **blue**. Please note that the figure citations in our response below refer to the new (post-revision) figures. We have highlighted the changes within the manuscript in **red**. We hope the Reviewers and the Editors will find this manuscript to be much improved and suitable for publication.

REVIEWERS' COMMENTS

Reviewer #1 (Remarks to the Author):

The authors have diligently utilized the time, to reasonably address most of my queries. The manuscript has improved substantially. I have some additional questions which can be addressed in the manuscript.

Response: We thank the reviewer for the positive comments.

1. The authors have reported difficulties, in Cut and Run experiments with the cut and run experiments with the Crispr knockout cell line for LSD1 due to insufficient IP material for sequencing. This seems to be a little difficult to understand as even regular non specific pulldowns happen with species specific IgG antibodies, which are usable for preparing sequencing libraries. This is a critical experiment that needs to be included if possible.

Response: We appreciate the reviewer's comments. We previously attempted CUT&Tag-seq in LSD1 knockout cells as suggested. However, there was a very minimal amount of DNA bound by the LSD1 antibody after LSD1 knockout, which was insufficient sequencing. The library quality was assessed using the Agilent 5400 system (Agilent, USA) and quantified with Qubit3.0 (Invitrogen, USA). The quality control results are shown in **Response Fig. 1a**. In the WT group, there is a periodic trapezoidal band distribution, while the knockout group does not exhibit a clear main peak. Additionally, the concentration of WT sample detected by Qubit fluorescence is 8.07 nM, while the knockout group only has 0.27 nM, which does not meet the minimum sequencing concentration requirement of 1 nM for Illumina Novaseq 6000 platforms. This is at least partially due to the complete deletion of LSD1.

To address the reviewer's concerns, we performed ChIP-qPCR analyses using LSD1 antibody and its isotype control IgG antibody in both LSD1 knockout and knockdown cells. As shown in **Response Fig. 1b, c**, LSD1 was significantly enriched at the HR gene loci compared with IgG, and this enrichment was diminished in LSD1 knockout or knockdown conditions. Notably, the LSD1 knockdown group exhibited residual binding at HR gene loci compared with the LSD1 knockout group, which could be attributed to the remaining LSD1 protein that can still be bound by the LSD1 antibody after LSD1 knockdown. In fact, due to the higher signal-to-noise ratio of CUT&Tag-seq as a cutting-edge technology, the results obtained using only WT cells for CUT&Tag-seq are convincing in demonstrating the binding sites of LSD1.

We acknowledge the reviewers' suggestion of using LSD1 knockout cells for CUT&Tag-seq, which would provide a more rigorous and important investigation. Regrettably, due to the limited time for revision, we were unable to conduct this specific experiment. However, we intend to improve our experimental system in the future, such as using an alternative LSD1 antibody or employing LSD1 knockdown cells for CUT&Tag-seq or ChIP-seq experiments.

2. The LSD1 mutants (K661 and DM) shows identical rescue phenotypes across different cell lines. Do they rescue transcriptionally as well? There should be a few lines in discussion as to how the enzymatically replete LSD1 (K661) is able to perform rescue as comparable to the complete enzymatic dead LSD1 (DM).

Response: We appreciate the reviewer's comments. According to the previous suggestion, we have reconstituted LSD1 knockdown and LSD1 knockout OC cells with wild-type LSD1 (LSD1-WT) or catalytically inactive LSD1 mutants (LSD1-K661A and a double mutant LSD1-A539E/K661A (LSD1-DM)) and repeated all the rescue experiments, including ChIP-qPCR, RT-qPCR, western blot analysis, assessment of HR ability, and evaluation of cell sensitivity to PARPi (Fig. 5f-1 and Supplementary Fig. 8a-g). The results showed that, in addition to phenotypes, genetic depletion of LSD1 decreased BRCA1/2 and RAD51 expression in mRNA and protein level which could be rescued by wild-type (WT) LSD1, but not catalytically inactive LSD1.

Specifically, the mutants (LSD1-K661A and LSD1-DM) showed identical rescue phenotypes, as well as gene transcription and expression in two different A2780 and ES2 cells (Fig. 5h-j and Supplementary Fig. 8c-e). In addition, we have added a few sentences to discuss it in the *Discussion* section (Lines 681-690). The sentences are as follows:

"It is worth noting that the mutants (LSD1-K661A and LSD1(A539E/K661A) double mutation) showed identical rescue gene expression and phenotypes. However, a previous study has shown that the LSD1(K661A) mutant retains demethylase activity on nucleosome substrates to some extent, while the LSD1(A539E/K661A) double mutation completely abrogates LSD1 enzymatic activity⁴⁸. The difference in LSD1(K661A) demethylation activity may be attributed to different experimental conditions between biochemical and cellular assays. Our study suggests that the K661 residue is a critical catalytic site of LSD1 in OC cells. Moreover, further exploration of the key catalytic residues of LSD1 in vitro and in vivo is worth being examined in future studies."

3. The authors should include some discussion as to how LSD1 potentially regulates H3K9Me2 (Direct versus indirect regulation through H3K9 modulators).

Response: We appreciate the constructive recommendation. We have added these issues in the *Discussion* section in the revised manuscript (Lines 616-626). The sentences are as follows:

"Moreover, although our study identifies LSD1 selectively and directly regulates H3K9 demethylation at HR gene loci, we cannot exclude the indirect regulation

through H3K9 modulators. SUV39H1/2, SETDB2, G9a and GLP mainly target H3K9 for methylation, whereas KDM3, KDM4, PHF2 and PHF8 mainly target H3K9 for demethylation. In addition to its demethylation of histone lysine residues, LSD1 is able to demethylate nonhistone proteins and regulate protein stability through demethylase-independent activity^{27, 63}. LSD1 can also interact with various chromatin-modifying enzymes and transcription factors, forming complexes that regulate gene expression⁶⁴, suggesting that LSD1 has the potential to indirectly regulate H3K9me2 levels by modulating the activity or recruitment of these H3K9 modulators."

Reviewer #2 (Remarks to the Author):

The authors has been responsive to previous critiques. This reviewer does not have any further comments. Congratulations on the great work!

Response: We thank the reviewer for their appreciation of our study and the constructive comments.

Reviewer #3 (Remarks to the Author):

The authors have addressed all of the concerns from the initial review. The authors have added clarification and numerous new data panels. Overall, the revised manuscript and data presented are improved, rigorous, and of high quality.

Response: We thank the reviewer for the positive comments.

Notably, since reviewing the initial manuscript there have been updated guidelines in terms of PARP inhibitor use in the context of HR-status. In the context of a patient being treated with a PARP inhibitor, the results of the study would likely only apply to those being treated with Niraparib. Notably, Niraparib is considerably more toxic compared to other PARPi and thus the doses used throughout the manuscript are considered non-physiological. Note, that toxicity studies, while comprehensive, were conducted with olaparib. While these updates do somewhat diminish the impact of the findings, there would still be an interest in the reported findings. The authors may want to consider addressing these updated guidelines in the discussion.

Response: We thank the reviewer for the positive comments. We thank the reviewer for the positive comments. We have included more discussions in the *Discussion* section in the revised manuscript (Lines 673-680) as follows:

"Although our in vivo data strongly support the efficacy of LSD1i in combination

of either of three PARPi (olaparib, niraparib, or rucaparib), low toxicity of the combination were mainly conducted with olaparib. Besides, the usage of PARPi in the management of various OC patient populations are slightly different according to the newest EMSO guideline⁶⁷. we have not demonstrated the activity and tolerability of the combination strategy in patients. Thus, the safety profile and therapy of the PARPi/LSD1i combination in patients carefully warrant exploration in human clinical trials."

Reviewer #4 (Remarks to the Author):

The authors have addressed all my questions and concerns in the rebuttal letter. I have no further comment on the revised manuscript. Thank you.

Response: We thank the reviewer for their appreciation of our study and the constructive comments.